# ActiveCQ: Active Estimation of Causal Quantities

**Erdun Gao[1,2] & Dino Sejdinovic[1,2]**
[1]Responsible AI Research Centre, Australian Institute for Machine Learning
[2]School of Mathematical Sciences, Adelaide University
`{erdun.gao,dino.sejdinovic}@adelaide.edu.au`

## Abstract

Estimating causal quantities (CQs) typically requires large datasets, which can be expensive to obtain, especially when measuring individual outcomes is costly. This challenge highlights the importance of sample-efficient active learning strategies. To address the narrow focus of prior work on the conditional average treatment effect, we formalize the broader task of Active Estimation of Causal Quantities (ActiveCQ) and propose a unified framework for this general problem. Built upon the insight that many CQs are integrals of regression functions, our framework models the regression function with a Gaussian process. For the distribution component, we explore both a baseline using explicit density estimators and a more integrated method using conditional mean embeddings in a reproducing kernel Hilbert space. This latter approach offers key advantages: it bypasses explicit density estimation, operates within the same function space as the GP, and adaptively refines the distributional model after each update. Our framework enables the principled derivation of acquisition strategies from the CQ's posterior uncertainty; we instantiate this principle with two utility functions based on information gain and total variance reduction. A range of simulated and semi-synthetic experiments demonstrate that our proposed framework significantly outperforms relevant baselines, achieving substantial gains in sample efficiency across a variety of CQs.

## 1 Introduction

Causality (Pearl, 2009; Imbens & Rubin, 2015; Hernan & Robins, 2023; Ding, 2024) aims to understand how interventions influence outcomes by modeling data-generating processes. Within this domain, causal inference (Hernan & Robins, 2023; Ding, 2024) focuses on estimating treatment effects, which are vital for decision-making in fields like economics (Heckman, 2000) and healthcare (Foster et al., 2011). While randomized controlled trials (RCTs) are the gold standard, they are often impractical (Benson & Hartz, 2017), prompting reliance on observational data (Rubin, 2005). In particular, we are often interested in how causal effects differ across various subpopulations, and there are several distinct causal quantities (CQs) such as the (conditional) average treatment effect ((C)ATE) and the average treatment effect on the treated (ATT) (Singh et al., 2024). Even in observational studies, measuring individual outcomes for causal estimation can be costly and burdensome (Nwaimo et al., 2024; Kallus & Mao, 2024). In personalized medicine, this may involve invasive procedures or expensive tests (Bi et al., 2019; Turk, 2002; Nwankwo et al., 2025), while in economics, tracking outcomes like income changes often requires labor-intensive follow-ups (McKenzie, 2012). These challenges highlight the need for efficient methods to identify which instances (e.g., patients or respondents) to prioritize for outcome measurement, enabling accurate estimation of CQs while minimizing data collection costs.

Active learning (AL) offers a promising approach to addressing this challenge by improving accuracy with fewer labeled samples, specifically by strategically selecting the most informative data points (Lindley, 1956; Chaloner & Verdinelli, 1995; Settles, 1994; Rainforth et al., 2024). Prior research on active causal inference has primarily focused on two fronts: developing active strategies for learning generalizable CATE estimators, often conditioning on all available covariates (Jesson et al., 2021; Qin et al., 2021; Wen et al., 2025a), and designing adaptable, loss-targeting acquisition functions (Connolly et al., 2023). Beyond these, many scientific fields emphasize identifying causal

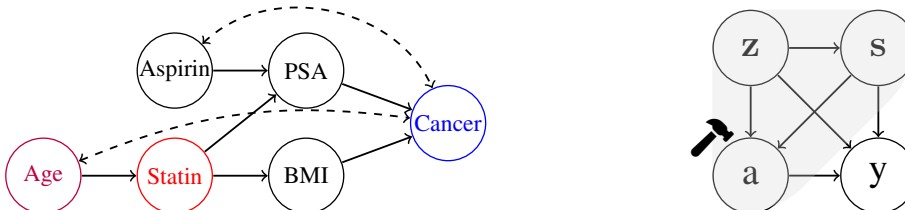

Figure 1: Causal graphs illustrating (Left) a motivating health example (Aglietti et al., 2020) and (Right) the general data generation model used in our framework, where the shaded area indicates variables observable in the pool.

effects within specific population subgroups (Jaber et al., 2019). For example, in a healthcare context like the one illustrated in Fig. 1 (left), researchers might investigate whether statin benefits older or younger patients, a question that is directly tied to CATE, which examines how causal effects differ across subgroups defined by age (Abrevaya et al., 2015). Similarly, in social services, providers may need to select which unstructured case notes to annotate to evaluate if outreach interventions effectively lead to housing placement for specific vulnerable populations (Nwankwo et al., 2025). In business analytics, companies often seek to quantify whether sales campaigns yield higher long-term value for distinct customer segments, requiring targeted verification of outcomes (Wen et al., 2025b;a; Tran et al., 2024). Another common focus is estimating the counterfactual outcomes for individuals currently receiving a fixed dosage of statin, such as what would happen had their dosage been increased, a scenario that pertains to the ATT. Furthermore, estimating treatment effects for a target population using observations collected from a different source population presents a significant challenge, commonly referred to as average treatment effect under distribution shift (ATEDS) (Singh et al., 2024; Gao et al., 2025; 2026). Clearly, for each distinct CQ, the choice of what constitutes the most informative data differs, and thus necessitates a tailored active learning strategy. This observation leads us to our key research question:

> 📌 **Key Question**: What are the guiding principles for strategically selecting items from a pool dataset in order to build accurate estimators for a given causal quantity of interest?

**Challenges.** Traditional information-theoretic AL approaches typically aim to maximize the information gained about the model parameters or minimize posterior uncertainty *over the unlabeled pool of data*. However, when estimating different CQs, the focus often shifts to the interventional distribution for some specific subpopulations. This shift naturally induces a distribution mismatch, where data samples are drawn from one distribution, but the outcome regression function needs to generalize to another. Consequently, conventional AL methods, such as Bayesian Active Learning by Disagreement (BALD) (Houlsby et al., 2011) and total variance reduction (TVR) (Cohn et al., 1996), frequently fail to align with the goal of constructing accurate regressors for the intended target distribution. These limitations highlight the necessity for CQ-aware acquisition strategies that explicitly account for the target interventional objective.

**Contributions.** Our primary contribution is a unified framework that both formalizes the task of *Active Estimation of Causal Quantities (ActiveCQ)* (Sec. 3) and provides a principled solution for it. This addresses a broad class of CQs often overlooked by the literature's focus on CATE. The cornerstone of our framework is a novel integral representation that unifies these disparate CQs. We model the required regression function with a Gaussian Process (GP) (Sec. 4.1) and demonstrate that using Conditional Mean Embeddings (CMEs) for the target distribution provides a powerful synergy, sidestepping the difficult challenge of direct density estimation (Sec. 4.3). This unified representation subsequently enables the principled derivation of bespoke utility functions. We show how classic strategies like information gain and total variance reduction (Sec. 4.5) can be instantiated in elegant, closed-form expressions, automatically and analytically tailored to the CQ of interest. We support our framework with convergence guarantees for this class of CQs (Sec. 4.6). Finally, we evaluate our method on the ActiveCQ task through several simulations and semi-synthetic datasets, demonstrating that our approach significantly outperforms baseline methods (Sec. 5).

## 2 PRELIMINARIES

**Causal DAG.** We represent a directed acyclic graph (DAG) as $\mathcal{G} = (\mathbf{v}, \mathcal{E})$, where $\mathbf{v}$ is a set of nodes corresponding to random variables, and $\mathcal{E}$ is a set of directed edges. A DAG is termed a *causal DAG* if each edge $v_i \to v_j$ indicates a direct causal effect. For a node $a \in \mathbf{v}$, an intervention, denoted by the **do-operator** $do(a = a)$ or simply $do(a)$, corresponds to an external action that sets $a$ to a fixed value $a$. This intervention modifies the data-generating process, leading to a post-intervention distribution $\mathbb{P}^*_{\mathbf{v}|do(a)}$ with density $p(\boldsymbol{v}|do(a)) = \prod_{v_i \in \mathbf{v} \setminus a} p(v_i|pa(v_i, \mathcal{G}))\mathbb{1}(a = a)$, where $pa(v_i, \mathcal{G})$ denotes the parents of $v_i$ in $\mathcal{G}$. Our analysis is based on the causal graph in Fig. 1 (right), which considers a treatment $a \in \mathcal{A}$, adjustment variables/confounders $\mathbf{s} \in \mathcal{S}$, optional effect modifiers $\mathbf{z} \in \mathcal{Z}$, and an outcome $\mathbf{y} \in \mathbb{R}$. For simplicity, we restrict attention to scalar treatment and outcome variables; the framework extends straightforwardly to the multivariate setting. The corresponding observational and interventional distributions are:

$$
\begin{aligned}
\textbf{(Observational)} \quad & p(a, \boldsymbol{z}, \boldsymbol{s}, y) = p(\boldsymbol{z})p(\boldsymbol{s}|\boldsymbol{z})p(a|\boldsymbol{z}, \boldsymbol{s})p(y|a, \boldsymbol{z}, \boldsymbol{s}), \\
\textbf{(Interventional)} \quad & p^*(a, \boldsymbol{z}, \boldsymbol{s}, y) = p(\boldsymbol{z})p(\boldsymbol{s}|\boldsymbol{z})p^*(a)p(y|a, \boldsymbol{z}, \boldsymbol{s}),
\end{aligned}
\tag{1}
$$

where $p^*(a)$ is the interventional treatment distribution. We occasionally denote the full set of input covariates as $\mathbf{x} = (a, \mathbf{z}, \mathbf{s})$. Further details on DAGs are provided in App. C.1.

**Causal Quantities Estimation.** In the causal inference literature, CQs typically refer to the expected effect of a do-operation, expressed as $\mathbb{E}[y|do(a)]$ over specific subpopulations. In what follows, we outline some commonly studied CQs, focusing on their estimation throughout this paper.

**Definition 1.** *(1) ATE: $\tau_{\text{ATE}}(a) := \mathbb{E}[y|do(a)]$ represents the average effect over the whole population under the intervention $a$. (2) CATE: $\tau_{\text{CATE}}(a, \boldsymbol{z}) := \mathbb{E}[y|do(a), \mathbf{z} = \boldsymbol{z}]$ represents the average effect over the subpopulation with $\mathbf{z} = \boldsymbol{z}$ under the intervention $a$. (3) ATT: $\tau_{\text{ATT}}(a, \tilde{a}) := \mathbb{E}[y|do(a), a = \tilde{a}]$ represents the average effect over the subpopulation who received treatment $\tilde{a}$ had they instead received the intervention $a$. (4) ATE with distribution shift (DS): $\tau_{\text{DS}}(a, \tilde{\mathbb{P}}) := \mathbb{E}[y \mid do(a), \tilde{\mathbb{P}}]$ denotes the average treatment effect under intervention $a$, evaluated over a target population whose covariate distribution $\tilde{p}(\boldsymbol{z}, \boldsymbol{s})$ differs from the observational distribution $p(\boldsymbol{z}, \boldsymbol{s})$ used in Eq. 1.*

To estimate these CQs, certain assumptions are necessary for identifiability. In this paper, we focus on scenarios where no unmeasured confounders exist between the cause and effect, and where the Stable Unit Treatment Value Assumption (SUTVA) and the positivity condition hold. More detailed discussions can be found in App. C.1.

**Lemma 1.** *Under these assumptions of selection on observables and covariate shift, we have*

$$
\tau_{\text{ATE}}(a) = \int_{\mathcal{S}} \mathbb{E}[y|a = a, \mathbf{s} = \boldsymbol{s}] \, \mathbb{P}_{\mathbf{s}}(d\boldsymbol{s}) \qquad \tau_{\text{ATT}}(a, \tilde{a}) = \int_{\mathcal{S}} \mathbb{E}[y|a = a, \mathbf{s} = \boldsymbol{s}] \, \mathbb{P}_{\mathbf{s}|a}(d\boldsymbol{s}|\tilde{a})
$$

$$
\tau_{\text{CATE}}(a, \boldsymbol{z}) = \int_{\mathcal{S}} \mathbb{E}[y|a = a, \mathbf{s} = \boldsymbol{s}, \mathbf{z} = \boldsymbol{z}] \, \mathbb{P}_{\mathbf{s}|\mathbf{z}}(d\boldsymbol{s}|\boldsymbol{z}) \qquad \tau_{\text{DS}}(a, \tilde{\mathbb{P}}) = \int_{\mathcal{S}} \mathbb{E}[y|a = a, \mathbf{s} = \boldsymbol{s}] \, \tilde{\mathbb{P}}_{\mathbf{s}}(d\boldsymbol{s})
$$

**Remark 1.** *To unify the estimation of various CQs, we can express all of them as $\tau_{\text{CQ}} = \int_{\overline{\mathcal{S}}} \mathbb{E}[y \mid a = a, \overline{\mathbf{s}} = \overline{\boldsymbol{s}}] \, \mathbb{P}^*_{\text{CQ}}(d\overline{\boldsymbol{s}})$. For CATE, where the effect modifier $\mathbf{z}$ is fixed, the integration variable is $\overline{\mathbf{s}} = \mathbf{s}$ and the integral is over the conditional distribution $\mathbb{P}_{\mathbf{s}|\mathbf{z}}$. For global quantities (ATE, ATT, and DS), the distinction between effect modifiers $\mathbf{z}$ and other confounders $\mathbf{s}$ is unnecessary for identification. Thus, for notational simplicity in these cases, we overload $\mathbf{s}$ to represent the full set of adjustment variables (effectively merging $\mathbf{z}$ and $\mathbf{s}$) and integrate over the entire joint distribution $\mathbb{P}^*_{\text{CQ}}$.*

## 3 THE ACTIVECQ PROBLEM FORMULATION

**Datasets.** In traditional CQ estimation tasks (Singh et al., 2024), it is typically assumed that an observational dataset $\{(\boldsymbol{x}^{(i)}, y^{(i)})\}_{i=1}^{n}$ is accessible, where the observations are drawn from the joint distribution $\mathbb{P}_{\mathbf{xy}}$. In the AL setup, as illustrated in Fig. 1 (right), we have access to two datasets: a labeled training dataset[1] $\mathcal{D}_T = \{(\boldsymbol{x}^{(i)}, y^{(i)})\}_{i=1}^{n_T}$ and an unlabeled pool dataset $\mathcal{D}_P = \{\boldsymbol{x}^{(i)}\}_{i=1}^{n_P}$ (Qin et al., 2021; Jesson et al., 2021). For convenience, we use symbols like $\boldsymbol{X}_T$ and $\boldsymbol{X}_P$ to represent

---

[1]"Labeled" indicates that the outcome is observable.

the corresponding sets of input observations in $\mathcal{D}_T$ and $\mathcal{D}_P$, respectively. Within this setup, a budget is allocated to select $n_B$ observations $\boldsymbol{X}_B = \{\boldsymbol{x}^{(i)}\}_{i=1}^{n_B}$ from $\mathcal{D}_P$. The corresponding labels, $\boldsymbol{y}_B = \{y^{(i)}\}_{i=1}^{n_B}$, can then be requested and added to $\mathcal{D}_T$ to update the training set. The updated $\mathcal{D}_T$ is subsequently used to estimate the desired CQs.

**Adaptive and batch modes.** The standard "adaptive" setting follows the *Label-One-Then-Train* loop, where a single label is acquired per round, repeated for $n_B$ rounds to acquire all labels (Holzmüller et al., 2023). In this process, the selection of $\boldsymbol{x}^{(i)}$ at each step depends on the labels of previously selected data points, $\{\boldsymbol{x}^{(j)}, y^{(j)}\}_{j<i}$ (Hübotter et al., 2024). Batch mode active learning involves requesting labels in smaller batches of size $n_b$ over $\lceil n_B/n_b \rceil$ rounds, rather than requesting all $n_B$ labels at once or one by one (Kirsch et al., 2019). Let $\mathcal{D} = \{\boldsymbol{x}^{(i)}\}_{i=1}^{n}$ (where $n = n_T + n_P$) denote the union of the observations of $\mathbf{x}$ from both the labeled dataset $\mathcal{D}_T$ and the unlabeled dataset $\mathcal{D}_P$. This union corresponds to the gray shaded area in Fig. 1 (right) and remains fully accessible throughout the process. After each training round, we request labels for a batch of $n_b$ data points, $\{\boldsymbol{x}^{(i)}\}_{i=1}^{n_b}$, from $\mathcal{D}_P$ and incorporate them into $\mathcal{D}_T$. Notably, batch-mode active learning reduces to the adaptive setting when $n_b = 1$. In this paper, we focus on the batch-mode setup.

> Note that our setting is purely observational. We can only query an individual's pre-existing, factual outcome rather than intervene to assign a new treatment and observe a counterfactual. This fundamentally distinguishes our work from active experimental design, which requires the ability to perform interventions (Toth et al., 2022; Kato et al., 2024; Klein et al., 2025).

# 4 UNCERTAINTY QUANTIFICATION OF CQS

To efficiently estimate CQs with minimal labeling cost, we propose a Bayesian active learning framework (Chaloner & Verdinelli, 1995). Our approach is guided by an information-theoretic principle: at each round, we select data points that are expected to maximally reduce the posterior uncertainty over the target causal quantity. We develop and illustrate this methodology for the CATE, $\hat{\tau}_{\text{CATE}}(a, \boldsymbol{z})$, abbreviated as $\hat{\tau}(a, \boldsymbol{z})$ for clarity, as it represents the most general and complex case among the causal quantities identified in Lemma 1. While our main exposition focuses on CATE, the framework is general, and its detailed application to other quantities is provided in App. D.

## 4.1 REGRESSION FUNCTION MODELING

We assume that the outcome follows the model $\mathbf{y} = \mathbb{E}[\mathbf{y}|\mathbf{a}, \mathbf{z}, \mathbf{s}] + \varepsilon$, where $\varepsilon \overset{\text{i.i.d.}}{\sim} \mathcal{N}(0, \sigma^2)$ (Singh et al., 2024). The observations $\{(a_i, \boldsymbol{z}_i, \boldsymbol{s}_i, y_i)\}_{i=1}^{n_T}$ are independently and identically sampled from $\mathbb{P}_{\mathbf{a}\mathbf{z}\mathbf{s}\mathbf{y}}$. To quantify the epistemic uncertainty of the causal effect, we model the outcome using a Gaussian Process (GP), where $\mathrm{f}(a, \boldsymbol{z}, \boldsymbol{s}) = \mathbb{E}[\mathbf{y}|\mathbf{a}, \mathbf{z}, \mathbf{s}]$ (Williams & Rasmussen, 2006; Kanagawa et al., 2025). The function $\mathrm{f}$ is assigned a GP prior, $\mathrm{f} \sim \mathcal{GP}(0, k)$, with a zero mean function and covariance kernel $k$, which is constructed as a product kernel to handle multiple inputs. Given the training dataset $\mathcal{D}_T = \{\boldsymbol{a}_T, \boldsymbol{Z}_T, \boldsymbol{S}_T, \boldsymbol{y}_T\}$, we compute the posterior GP. For new inputs $\boldsymbol{x}$ and $\boldsymbol{x}'$, the posterior mean and covariance function of $\mathrm{f}$ can be calculated as follows:

$$m(\boldsymbol{x}) = \boldsymbol{k}_{\boldsymbol{x}\boldsymbol{X}_T}(\boldsymbol{K}_{\boldsymbol{X}_T\boldsymbol{X}_T} + \sigma^2\boldsymbol{I})^{-1}\boldsymbol{y}_T, \quad k_{\text{post}}(\boldsymbol{x}, \boldsymbol{x}') = k_{\boldsymbol{x}\boldsymbol{x}'} - \boldsymbol{k}_{\boldsymbol{x}\boldsymbol{X}_T}(\boldsymbol{K}_{\boldsymbol{X}_T\boldsymbol{X}_T} + \sigma^2\boldsymbol{I})^{-1}\boldsymbol{k}_{\boldsymbol{X}_T\boldsymbol{x}'}. \quad (2)$$

where $k_{\boldsymbol{x}\boldsymbol{x}'} = k_{aa'}k_{\boldsymbol{z}\boldsymbol{z}'}k_{\boldsymbol{s}\boldsymbol{s}'}$, $\boldsymbol{k}_{\boldsymbol{x}\boldsymbol{X}_T} = \boldsymbol{k}_{a\boldsymbol{a}_T} \odot \boldsymbol{k}_{\boldsymbol{z}\boldsymbol{Z}_T} \odot \boldsymbol{k}_{\boldsymbol{s}\boldsymbol{S}_T}$, and $\boldsymbol{K}_{\boldsymbol{X}_T\boldsymbol{X}_T} = \boldsymbol{K}_{\boldsymbol{a}_T\boldsymbol{a}_T} \odot \boldsymbol{K}_{\boldsymbol{Z}_T\boldsymbol{Z}_T} \odot \boldsymbol{K}_{\boldsymbol{S}_T\boldsymbol{S}_T}$. For brevity, we use $k(\cdot, \cdot)$ as a general kernel notation, where specific kernels follow from their arguments (e.g., $k(\boldsymbol{x}, \boldsymbol{x}')$ denotes the kernel on $\mathcal{X}$). Additionally, let $\phi(\boldsymbol{x})$ denote the corresponding feature map. We define $k_{\boldsymbol{x}\boldsymbol{x}'} := k(\boldsymbol{x}, \boldsymbol{x}')$, $\boldsymbol{k}_{\boldsymbol{x}\boldsymbol{X}_T} := [k(\boldsymbol{x}, \boldsymbol{X}_1), \ldots, k(\boldsymbol{x}, \boldsymbol{X}_{n_T})]$, and $\boldsymbol{K}_{\boldsymbol{X}_T\boldsymbol{X}_T} = [k(\boldsymbol{X}_i, \boldsymbol{X}_j)]_{i,j=1}^{n_T}$.

**Remark 2** (On Scalability and Complexity). *While the standard GP formulation in Eq. 2 has $O(n_T^3)$ complexity, our framework's core contribution is orthogonal to this specific implementation. The proposed framework is modular and readily accommodates various scalable approximations, such as sparse variational GPs, Random Fourier Features, and Nyström methods, to handle larger datasets.*

## 4.2 CONDITIONAL DENSITY ESTIMATION

Next, we turn our attention to modeling the conditional distribution $\mathbb{P}_{\mathbf{s}|\mathbf{z}}$ and explore two distinct approaches: a conditional density estimator (CDE) and a conditional kernel mean embedding (CME).

**Conditional Density Estimator.** A natural approach to estimate the conditional distribution is to use a CDE, such as a mixture density network (Bishop & Nasrabadi, 2006), a least squares density ratio estimator (Sugiyama et al., 2010), a conditional normalizing flow (Trippe & Turner, 2018), or a squared neural family (Tsuchida et al., 2023; 2025). Importantly, estimating $\mathbb{P}_{\mathbf{s}|\mathbf{z}}$ requires only paired observations $(\mathbf{s}, \mathbf{z})$, and does not rely on outcome labels, allowing it to be done using both training and pool datasets before label acquisition. Since $(\boldsymbol{S}, \boldsymbol{Z})$ is always available, incorporating the pool significantly enhances estimation accuracy with minimal overhead.

**Conditional Embeddings in RKHS.** We introduce an alternative representation of the conditional distribution in a reproducing kernel Hilbert space (RKHS). As we model the regression function using a GP, the input features a, $\mathbf{z}$, and $\mathbf{s}$ are mapped into a joint feature space. The product kernel used in GP regression corresponds to the tensor product RKHS $\mathcal{H}_{\mathcal{A}\mathcal{Z}\mathcal{S}} := \mathcal{H}_{\mathcal{A}} \otimes \mathcal{H}_{\mathcal{Z}} \otimes \mathcal{H}_{\mathcal{S}}$. We can then represent the conditional distribution $\mathbb{P}_{\mathbf{s}|\mathbf{z}=\boldsymbol{z}}$ in $\mathcal{H}_{\mathcal{S}}$ using the CME, which is defined as:

$$\mu_{\mathbf{s}|\mathbf{z}=\boldsymbol{z}} := \mathbb{E}_{\mathbf{s}|\mathbf{z}=\boldsymbol{z}}[\phi(\boldsymbol{s})] = \int_{\mathcal{S}} \phi(\boldsymbol{s})\mathbb{P}_{\mathbf{s}|\mathbf{z}}(d\boldsymbol{s}|\boldsymbol{z}). \tag{3}$$

The CME serves as the embedding of $\mathbb{P}_{\mathbf{s}|\mathbf{z}=\boldsymbol{z}}$ in $\mathcal{H}_{\mathcal{S}}$. As discussed in (Song et al., 2013; Muandet et al., 2017), the CME can be associated with a Hilbert-Schmidt operator $C_{\mathbf{s}|\mathbf{z}} : \mathcal{H}_{\mathcal{Z}} \to \mathcal{H}_{\mathcal{S}}$, referred to as the conditional mean embedding operator (CMO). This operator satisfies $\mu_{\mathbf{s}|\mathbf{z}=\boldsymbol{z}} = C_{\mathbf{s}|\mathbf{z}}\phi(\boldsymbol{z})$, and can be understood as $C_{\mathbf{s}|\mathbf{z}} = C_{\mathbf{sz}}C_{\mathbf{zz}}^{-1}$ with $C_{\mathbf{sz}} := \mathbb{E}_{\mathbf{s},\mathbf{z}}[\phi(\mathbf{s}) \otimes \phi(\mathbf{z})]$ and $C_{\mathbf{zz}} := \mathbb{E}_{\mathbf{z}}[\phi(\mathbf{z}) \otimes \phi(\mathbf{z})]$ representing the covariance operators. Similar to the CDE approach, we can use all paired observations $(\boldsymbol{Z}, \boldsymbol{S})$ in $\mathcal{D}$ to empirically estimate $C_{\mathbf{s}|\mathbf{z}}$ as $\hat{C}_{\mathbf{s}|\mathbf{z}} = \Phi_{\boldsymbol{S}}(\boldsymbol{K}_{\boldsymbol{ZZ}} + \lambda\boldsymbol{I})^{-1}\Phi_{\boldsymbol{Z}}^T$, where $\lambda > 0$ is a regularization parameter. The kernel for $\mathbf{z}$ here need not be the same as the one employed in the GP, and the kernel for $\mathbf{s}$ will be discussed later.

## 4.3 INTEGRAL OVER CONDITIONAL DISTRIBUTION

Leveraging these two representations of $\mathbb{P}_{\mathbf{s}|\mathbf{z}=\boldsymbol{z}}$, we then proceed to derive the final estimator for the CQ $\hat{\tau}(a, \boldsymbol{z})$. As a linear functional of the regression function f, $\hat{\tau}(a, \boldsymbol{z})$ follows a distribution determined by the underlying GP, which is also a GP with mean and covariance functions derived from the conditional structure as in (Chau et al., 2021a). For all $a, a' \in \mathcal{A}$ and $\boldsymbol{z}, \boldsymbol{z}' \in \mathcal{Z}$, we have:

$$\nu(a, \boldsymbol{z}) = \mathbb{E}_{\mathbf{s}\sim\mathbb{P}_{\mathbf{s}|\mathbf{z}}}\left[m(a, \boldsymbol{z}, \mathbf{s})\right],$$
$$q\left((a, \boldsymbol{z}), (a', \boldsymbol{z}')\right) = \mathbb{E}_{\mathbf{s}\sim\mathbb{P}_{\mathbf{s}|\mathbf{z}}, \mathbf{s}'\sim\mathbb{P}_{\mathbf{s}|\mathbf{z}'}}\left[k_{\text{post}}\left((a, \boldsymbol{z}, \mathbf{s}), (a', \boldsymbol{z}', \mathbf{s}')\right)\right]. \tag{4}$$

**CDE-based method.** To approximate the posterior mean and covariance, we employ a sampling-based method. Specifically, we first draw $n_{\mathbf{s}}$ samples $\{\boldsymbol{s}^{(1)}, \cdots, \boldsymbol{s}^{(n_{\mathbf{s}})}\}$ from $\mathbb{P}_{\mathbf{s}|\mathbf{z}=\boldsymbol{z}}$, and similarly $\{\boldsymbol{s}'^{(1)}, \cdots, \boldsymbol{s}'^{(n_{\mathbf{s}})}\}$ from $\mathbb{P}_{\mathbf{s}|\mathbf{z}=\boldsymbol{z}'}$. The posterior mean $\nu(a, \boldsymbol{z})$ is then approximated by averaging the function $m(a, \boldsymbol{z}, \boldsymbol{s})$ over these samples, while the posterior covariance is estimated by averaging the kernel values over all pairwise combinations:

$$\nu(a, \boldsymbol{z}) = \frac{1}{n_{\mathbf{s}}}\sum_{i=1}^{n_{\mathbf{s}}} m(a, \boldsymbol{z}, \boldsymbol{s}^{(i)}), \quad q\left((a, \boldsymbol{z}), (a', \boldsymbol{z}')\right) = \frac{1}{n_{\mathbf{s}}^2}\sum_{i=1}^{n_{\mathbf{s}}}\sum_{j=1}^{n_{\mathbf{s}}} k_{\text{post}}\left((a, \boldsymbol{z}, \boldsymbol{s}^{(i)}), (a', \boldsymbol{z}', \boldsymbol{s}'^{(j)})\right). \tag{5}$$

The expressions for $m$ and $k_{\text{post}}$ are provided in Eq. 2. This sampling approach marginalizes out $\mathbf{s}$, approximating the posterior distribution and thereby enabling CATE estimation with uncertainty. The estimator will be used for sample-efficient estimation in the next section.

**CME-based method.** Leveraging the CME $\mu_{\mathbf{s}|\boldsymbol{z}}$, we can evaluate the integrals defining the posterior mean $\nu(a, \boldsymbol{z})$ and covariance $q((a, \boldsymbol{z}), (a', \boldsymbol{z}'))$ in the previous equation analytically. By representing the conditional distribution in the RKHS via $\hat{\mu}_{\mathbf{s}|\boldsymbol{z}}$, the integration of the GP posterior mean $m$ and covariance $k_{\text{post}}$ can be reduced to closed-form kernel operations. This allows us to derive the CATE estimator directly from the conditional structure of the GP.

**Proposition 1.** *Given the dataset $\mathcal{D}_T = \{a_T, Z_T, S_T, y_T\}$ and $\mathcal{D} = \{A, Z, S\}$, if f is the posterior GP learned from $\mathcal{D}_T$, then $\hat{\tau}_{\mathrm{CATE}}$ is a functional of f defined on $(a, z)$ with the following mean and covariance estimated using $\phi_{\bar{x}} := \phi_a \otimes \phi_z \otimes \hat{\mu}_{s|z=z}$ and $\phi_{\bar{x}'} := \phi_{a'} \otimes \phi_{z'} \otimes \hat{\mu}_{s|z=z'}$,*

$$
\nu(a, z) = \langle \phi_{\bar{x}}, m_{\mathrm{f}} \rangle_{\mathcal{H}_{\mathcal{AZS}}} = k_{\bar{x} X_T} (K_{X_T X_T} + \sigma^2 I)^{-1} y_T,
$$
$$
q((a, z), (a', z')) = k_{\bar{x}\bar{x}'} - k_{\bar{x} X_T} (K_{X_T X_T} + \sigma^2 I)^{-1} k_{X_T \bar{x}'}.
$$
(6)

*where the effective kernel terms incorporating the CME are defined as: $k_{\bar{x}\bar{x}'} = k_{aa'} k_{zz'} (k_{zZ} (K_{ZZ} + \lambda I)^{-1} K_{SS} (K_{ZZ} + \lambda I)^{-1} k_{Zz'})$, $k_{\bar{x} X_T} = k_{aa_T} \odot k_{zZ_T} \odot (k_{zZ} (K_{ZZ} + \lambda I)^{-1} K_{SS_T})$, $K_{X_T X_T} = K_{a_T a_T} \odot K_{Z_T Z_T} \odot K_{S_T S_T}$, and $k_{X_T \bar{x}'} = k_{a_T a'} \odot k_{Z_T z'} \odot (K_{S_T S} (K_{ZZ} + \lambda I)^{-1} k_{Zz'})$. Here, $\lambda > 0$ is the regularization parameter for the CME, and $\sigma^2 > 0$ is the GP observation noise variance.*

To ensure that the CME corresponds to the RKHSs that appropriately match the kernels learned in the GP regression, the features $\phi(a), \phi(z), \phi(s)$ need to be updated after each batch acquisition round. By leveraging the CME, this approach effectively obviates the need for explicit conditional distribution estimation. It shifts the CATE estimation task from the computationally intensive numerical integration of a regression function to directly manipulating a distribution embedding, thereby streamlining computation and improving efficiency. Additionally, uncertainty from the CME itself can be introduced and propagated alongside the GP's inherent uncertainty, as explored in methods like BayesIMP (Chau et al., 2021b) and IMPspec (Dance et al., 2024), although detailed exploration lies beyond the scope of the current paper. Further advantages that make CME particularly well-suited to our AL process are detailed in App. G.2.

### 4.4 SPECIFYING THE SUBPOPULATION OF INTEREST

To evaluate the estimator $\hat{\tau}(a, z)$, it is crucial to specify the subpopulation and treatments of interest, as they determine where estimation accuracy matters most. We consider a set of treatment–effect modifier pairs $(a_I, Z_I) = \{(a_i, z_i)\}_{i=1}^{n_I}$ over which performance is assessed. For instance, if the goal is to understand the response of a specific subpopulation defined by $z = z$ to varying treatments, one may fix each $z_i = z$ and draw $a_i$ uniformly from a finite treatment set $\mathcal{A}$, i.e., $a_i \sim \mathrm{Uniform}(\mathcal{A})$ and $(a_i, z_i) = (a_i, z)$. Alternatively, if the interest lies in evaluating responses to a fixed treatment $a$ across different effect modifiers, each $a_i$ is fixed to $a$ and $z_i$ is sampled uniformly from a set $\mathcal{Z}$, i.e., $z_i \sim \mathrm{Uniform}(\mathcal{Z})$ and $(a_i, z_i) = (a, z_i)$. These scenarios reflect different inferential goals and ensure that the assessment of $\hat{\tau}(a, z)$ is aligned with the intended application. The resulting sample $\{(a_i, z_i)\}_{i=1}^{n_I}$ forms the basis for measuring the estimator's accuracy in the region of interest.

### 4.5 UNCERTAINTY REDUCTION

We are now ready to address the **Key Question**. To this end, we propose the following guiding principle for selecting individuals from the pool dataset $\mathcal{D}_P$ whose outcomes should be acquired.

> 🔑 **Key Principle**: Select a subset of samples $X_B$ from the pool $\mathcal{D}_P$ in a manner that minimizes the posterior uncertainty of the estimator $\hat{\tau}(a_I, Z_I)$.

The distinction between active CQ estimation and traditional AL tasks is evident. Traditional AL methods, such as BALD and TVR, focus on minimizing the uncertainty of the regression function f over a reference input distribution, often represented by the unlabeled pool (Smith et al., 2023). In contrast, active estimation of CQ aims to reduce the uncertainty of the CQ of interest directly, with the regression function serving only as a means to an end. This principle guides data selection by quantifying uncertainty or label utility using methods like entropy and variance of the target estimator. We explore two strategies: information gain (IG) and TVR, and discuss their connection in App. D.5.

**Data Acquisitions via IG.** To evaluate the uncertainty of the target estimator $\hat{\tau}(a_I, Z_I)$, we utilize its differential entropy, denoted as $\mathrm{H}[\hat{\tau}(a_I, Z_I)]$. To obtain the labels for observations $X_B$ from $\mathcal{D}_P$, we quantify posterior uncertainty using conditional entropy after observing $y_{X_B}$. This is expressed as $\mathbb{E}_{y_{X_B} \sim p(\cdot|X_B, \mathcal{D}_T)}[\mathrm{H}(\hat{\tau}(a_I, Z_I) \mid \mathcal{D}_T, y_{X_B})]$, which we simplify as $\mathrm{H}(\hat{\tau}(a_I, Z_I) \mid y_{X_B}, \mathcal{D}_T)$. Here, we use $y$ to show its inherent randomness, which will be marginalized further. The IG, defined as $\mathrm{I}(\hat{\tau}(a_I, Z_I); y_{X_B} | \mathcal{D}_T) = \mathrm{H}[\hat{\tau}(a_I, Z_I)|\mathcal{D}_T] - \mathrm{H}[\hat{\tau}(a_I, Z_I)|y_{X_B}, \mathcal{D}_T]$, captures the reduction in uncertainty about $\hat{\tau}(a_I, Z_I)$ after observing $y_{X_B}$. For brevity, we omit the constraints $X_B \subseteq \mathcal{D}_P$

and $|\boldsymbol{X}_B| = n_b$ in the following optimization expressions, assuming all selections are from the pool set with the prescribed batch size unless specified otherwise. Based on the IG criterion, the acquisition rule is

$$\boldsymbol{X}_B^* = \underset{\substack{\boldsymbol{X}_B \subseteq \mathcal{D}_P \\ |\boldsymbol{X}_B| = n_b}}{\arg\min} \mathrm{H}\left(\hat{\tau}(\boldsymbol{a}_I, \boldsymbol{Z}_I) \mid \mathbf{y}_{\boldsymbol{X}_B}, \mathcal{D}_T\right) = \underset{\substack{\boldsymbol{X}_B \subseteq \mathcal{D}_P \\ |\boldsymbol{X}_B| = n_b}}{\arg\max} \mathrm{I}(\hat{\tau}(\boldsymbol{a}_I, \boldsymbol{Z}_I); \mathbf{y}_{\boldsymbol{X}_B} | \mathcal{D}_T). \tag{7}$$

Since we employ a GP framework, this quantity can be expressed in closed form using the posterior covariance matrix of the predictive distribution. Namely, for Gaussian-distributed quantities, entropy is given by: $\mathrm{H}(\mathcal{N}(0, \boldsymbol{\Sigma})) = \frac{1}{2} \log |(2\pi e)\boldsymbol{\Sigma}|$. Thus, the mutual information simplifies to:

$$\boldsymbol{X}_B^* = \underset{\substack{\boldsymbol{X}_B \subseteq \mathcal{D}_P \\ |\boldsymbol{X}_B| = n_b}}{\arg\max} \frac{1}{2} \log \left( \frac{\det\left(\mathbb{V}\mathrm{ar}[\hat{\tau}(\boldsymbol{a}_I, \boldsymbol{Z}_I)|\mathcal{D}_T]\right)}{\det\left(\mathbb{V}\mathrm{ar}[\hat{\tau}(\boldsymbol{a}_I, \boldsymbol{Z}_I)|\mathcal{D}_T, \mathbf{y}_{\boldsymbol{X}_B}]\right)} \right) = \underset{\substack{\boldsymbol{X}_B \subseteq \mathcal{D}_P \\ |\boldsymbol{X}_B| = n_b}}{\arg\min} \det\left(\mathbb{V}\mathrm{ar}[\hat{\tau}(\boldsymbol{a}_I, \boldsymbol{Z}_I)|\mathcal{D}_T, \mathbf{y}_{\boldsymbol{X}_B}]\right). \tag{8}$$

where $\det(\cdot)$ represents the determinant of a matrix. Note that although the notation includes $\mathbf{y}_{\boldsymbol{X}_B}$, the denominator's covariance matrix does not actually depend on the specific values of outputs, only on the input locations $\boldsymbol{X}_B$. This is a standard property of the Gaussian process.

**Data Acquisitions via TVR.** The other measure of prediction uncertainty is the total variance, which is defined as the sum of the marginal variances over the target set $\sum_{(a,\boldsymbol{z}) \in (\boldsymbol{a}_I, \boldsymbol{Z}_I)} \mathbb{V}\mathrm{ar}[\hat{\tau}(a, \boldsymbol{z})]$. Therefore, we can have the TVR strategy as our data acquisition function as follows.

$$\boldsymbol{X}_B^* = \underset{\substack{\boldsymbol{X}_B \subseteq \mathcal{D}_P \\ |\boldsymbol{X}_B| = n_b}}{\arg\min} \mathrm{Tr}\left(\mathbb{V}\mathrm{ar}[\hat{\tau}(\boldsymbol{a}_I, \boldsymbol{Z}_I)|\mathcal{D}_T, \mathbf{y}_{\boldsymbol{X}_B}]\right). \tag{9}$$

Here, we define $U_{\mathrm{IG}}(\boldsymbol{X}_B) = \mathrm{I}(\hat{\tau}(\boldsymbol{a}_I, \boldsymbol{Z}_I); \mathbf{y}_{\boldsymbol{X}_B} | \mathcal{D}_T)$ as the IG-based label utility function of $\boldsymbol{X}_B$. Similarly, the corresponding utility function for the TVR-based approach is given by $U_{\mathrm{TVR}}(\boldsymbol{X}_B) = -\mathrm{Tr}\left(\mathbb{V}\mathrm{ar}[\hat{\tau}(\boldsymbol{a}_I, \boldsymbol{Z}_I) | \mathcal{D}_T, \mathbf{y}_{\boldsymbol{X}_B}]\right)$, representing the TVR-based label utility function of $\boldsymbol{X}_B$. Based on these utility functions, we can express the unified acquisition rule as $\boldsymbol{X}_B = \arg\max U(\boldsymbol{X}_B)$.

**Batch Selection.** To enable efficient data acquisition, we adopt batch-wise selection, choosing $n_b$ data points at a time. A simple method involves using the utility function to rank all data points in $\mathcal{D}_P$ and selecting the top $n_b$. Recent studies (Gentile et al., 2024) emphasize the importance of ensuring diversity within batches for greater efficiency. This can be achieved through a greedy approximation, where the selection of each data point $\boldsymbol{x}_i$ considers the previously selected points $\{\boldsymbol{x}_j\}_{j=1}^{i-1}$. This can be formalized as maximizing a batch utility function $U(\boldsymbol{X}_b)$, which values the joint contribution of points in the batch. As finding the optimal batch $\boldsymbol{X}_b^* = \arg\max_{|\boldsymbol{X}_b| = n_b} U(\boldsymbol{X}_b)$ is computationally intractable, we employ a greedy approximation. This approach sequentially constructs the batch by iteratively adding the point that provides the highest marginal utility gain. Specifically, given the set of already selected points $\boldsymbol{X}_{i-1}^*$, the next point is chosen as:

$$\boldsymbol{x}_i^* = \underset{\boldsymbol{x} \in \mathcal{D}_P \setminus \boldsymbol{X}_{i-1}^*}{\arg\max} U(\boldsymbol{X}_{i-1}^* \cup \{\boldsymbol{x}\}) \tag{10}$$

By employing this greedy selection strategy, we can enforce the diversity and informativeness of data points in one batch. The overall procedure for active CATE estimation is shown in Alg. 1. Moreover, softmax-BALD, introduced in (Kirsch et al., 2023) through importance-weighted sampling over the pool dataset, was also adopted in CausalBALD; more details and results are provided in App. F.6.3.

## 4.6 Uncertainty Decay Analysis

In this subsection, we analyze the convergence of posterior uncertainty when using the proposed data acquisition function with the CQ estimator. We begin by introducing the following assumption.

**Assumption 1.** *The utility function $U$ is submodular over the pool dataset $\boldsymbol{X}_P$.*

Note that differential entropy often violates this assumption, while the information gain utility satisfies it under mild conditions in the GP framework (Krause et al., 2008; Srinivas et al., 2012). Further justification of this assumption is provided in App. E.1. We then define the following key quantities.

**Definition 2.** *Let $\boldsymbol{f}_{\boldsymbol{X}_P}$ denote the latent GP values at the pool inputs $\boldsymbol{X}_P$. We define the information capacities*

$$\gamma_{n_B}^{\mathrm{CQ}} \overset{\text{def}}{=} \max_{\substack{\boldsymbol{X} \subseteq \boldsymbol{X}_P \\ |\boldsymbol{X}| \leq n_B}} \mathrm{I}(\hat{\tau}(\boldsymbol{a}_I, \boldsymbol{Z}_I); \mathbf{y}_{\boldsymbol{X}} | \mathcal{D}_T), \quad \gamma_{n_B}^P \overset{\text{def}}{=} \max_{\substack{\boldsymbol{X} \subseteq \boldsymbol{X}_P \\ |\boldsymbol{X}| \leq n_B}} \mathrm{I}(\boldsymbol{f}_{\boldsymbol{X}_P}; \mathbf{y}_{\boldsymbol{X}} | \mathcal{D}_T), \tag{11}$$

Figure 2: Comparison of $\sqrt{\text{AMSE}}$ on simulation datasets (shaded: standard error). Baselines: Random, $\mu$-BALD, Coresets and TVR. Ours: "G" for greedy, others for top-$b$ acquisition.

and let $\gamma_{n_B} \overset{\text{def}}{=} \max\{\gamma_{n_B}^{\text{CQ}}, \gamma_{n_B}^{P}\}$ denote the dominating information capacity. We also define the irreducible uncertainty as

$$\eta_{\mathcal{D}_P}^2(a, \boldsymbol{z}) \overset{\text{def}}{=} \mathbb{V}\text{ar}\big[\hat{\tau}(a, \boldsymbol{z}) \mid \mathcal{D}_T, \boldsymbol{f}_{\boldsymbol{X}_P}\big], \tag{12}$$

which represents the remaining uncertainty of $\hat{\tau}(a, \boldsymbol{z})$ after the latent regressor values over the pool are known.

Finally, we bound the marginal variance of the treatment effect estimator under a given acquisition strategy, showing it is controlled by both the irreducible uncertainty and the information capacity:

**Theorem 2.** *Suppose Assumption 1 holds, and the data acquisition follows a utility function $U$ satisfying this assumption. Let $\boldsymbol{X}_B^{\star}$ denote the set of $n_B$ individuals acquired from $\mathcal{D}_P$. Under these conditions, there exists a constant $C > 0$ such that for any $n_B \geq 1$ and for each pair $(a, \boldsymbol{z}) \in (\boldsymbol{a}_I, \boldsymbol{Z}_I)$, the marginal posterior variance is bounded as:*

$$\mathbb{V}\text{ar}\big[\hat{\tau}(a, \boldsymbol{z}) \mid \mathcal{D}_T, \mathbf{y}_{\boldsymbol{X}_B^{\star}}\big] \leq \eta_{\mathcal{D}_P}^2(a, \boldsymbol{z}) + C\left(\gamma_{n_B}/\sqrt{n_B}\right). \tag{13}$$

The convergence analysis proof of our proposed acquisition strategy is presented in App. E, mainly building on the transductive active learning framework introduced by Hübotter et al. (2024).

## 5 EXPERIMENTAL RESULTS

We validate our approaches for active CQ estimation on multiple simulations as well as the semi-synthetic IHDP (Louizos et al., 2017) and Lalonde (LaLonde, 1986) datasets.

**Baselines, Implementations, and Metrics.** To evaluate the effectiveness of our proposed framework, we compare it with various baseline acquisition strategies, including random selection, total variance reduction, $\mu$-BALD, and QHTE (which is based on the core-set method). To ensure a fair comparison across all settings and methods, we use a GP to approximate the regression function, and either a Mixture Density Network (MDN) (Bishop & Nasrabadi, 2006) or CME for the conditional distribution estimation. Detailed implementations as well as the hyper-parameters of the baseline methods and our proposed methods are provided in App. F.2. We evaluate the performance of methods using the Average Mean Squared Error (AMSE), which quantifies the accuracy of the estimated CQ compared to the true CQ. Note that the uncertainty reduction criterion is not well-suited for batch active learning. A detailed explanation is provided in App. F.4.1. All results are reported as the mean ± standard deviation, computed over 20 independent random test set runs for each configuration.

**Synthetic Data Analysis.** Limited access to counterfactual data often necessitates the use of synthetic or semi-synthetic datasets for evaluating treatment effect estimation methods (Bica et al., 2020; Gao et al., 2024). We design two simulation settings, each involving a single conditioning variable. The first includes two adjustment variables to facilitate visualization, while the second uses four adjustment variables for numerical evaluation. For tasks beyond CATE estimation, the conditioning variable is treated as part of the adjustment set and is not explicitly illustrated. All simulation datasets are generated using a predefined process, where treatment assignment is influenced by covariates, following a data generation process similar to that in (Abrevaya et al., 2015; Singh et al., 2024). We define two types of target treatments for all CQs: one with a fixed treatment value and another considering all possible treatment values, as described in Sec. 4.4. For clarity, we present results for the second case in the main paper, while comprehensive results for all combinations of these settings, both for binary and continuous treatments, are available in App. F.6.3.

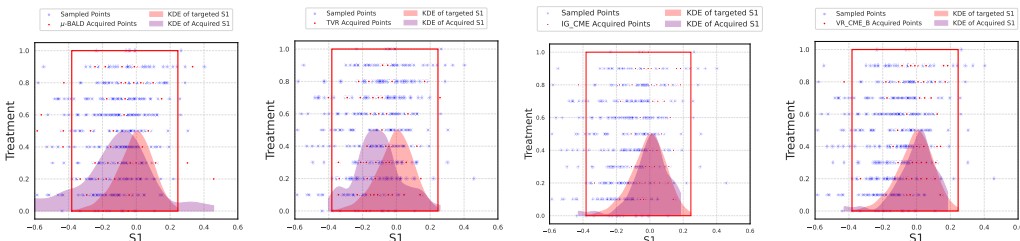

Figure 3: Comparison of acquisition patterns for four methods (left to right): $\mu$-BALD, TVR, IG-CME, and TVR-CME. Representative acquired points are shown for selected methods; complete visualizations are provided in App. F.6.1.

**Results.** Fig. 2 presents the results for CATE, ATE, ATT, and DS on the simulation dataset. Our proposed methods consistently achieve the best performance by prioritizing data that aligns with the target distribution of interest. The corresponding sampling results are shown in Fig. 3, demonstrating that our proposed method can acquire observations well-aligned with the target distribution. Moreover, for CATE, we can see that TVR with CME consistently outperforms MC sampling-based methods in both cases, as CME directly operates on features relevant for the GP regression task, making it more prediction-oriented and efficient compared to estimating conditional densities. More detailed explanations on the advantages of CME in our AL setup are provided in App. G.2. For ATE, all methods, including the baselines, sample from the entire population, leading to similar performance among the uncertainty-aware methods, all of which outperform random acquisition. However, IG-based methods may suffer from numerical instability when computing the determinant of large covariance matrices, potentially leading to suboptimal performance. We also observe that in the ATE with DS case, all our proposed methods significantly outperform the baseline methods, due to the distribution shift between the target and sampling distributions. Additional ablation results on the stability of our methods, considering factors such as different starting points, pool dataset sizes, batch sizes, and kernel choices, are provided in App. F.6.2.

**Running Time Analysis.** In Fig. 4, we present the running time for all methods on the CATE estimation task. While baseline methods that use naive top-$n_b$ selection per acquisition round are slightly faster, our approaches incur additional cost due to learning conditional distributions. The overall runtime is influenced by three main factors: (1) greedy acquisition, which selects points sequentially and triggers frequent posterior updates; (2) the pool size, which affects the scale of utility evaluation; and (3) the entropy computation in IG-based strategies. As shown in Fig. 4(b), smaller batch sizes intensify the cost of greedy selection due to repeated posterior recalculations. For GP-based methods, covariate dimensionality has limited impact, as runtime is mainly driven by distance computations. Despite these overheads, all methods remain computationally feasible in our experiments. Further details on computational complexity are provided in App. F.5.

**Semi-synthetic Data Analysis.** For real-world case evaluation, we compare all algorithms on the widely used IHDP benchmark, which includes real covariates paired with simulated outcomes. This benchmark poses significant challenges due to its small size, imbalance, and limited overlap in covariate distributions. Notably, the two potential outcome functions, while supported on the same covariates, have distinct functional forms, making the estimation of their difference particularly difficult. Our results show that for CATE and DS scenarios, where substantial distributional shifts exist between the target distribution and the pool data distribution, our proposed methods—especially those leveraging the CME framework—consistently outperform other approaches. A comprehensive description of the dataset, additional experimental results under various setups, and detailed discussions are provided in App. F.6.4. We additionally evaluate on the Lalonde dataset; further results and implementation details appear in App. F.6.5.

## 6 RELATED WORK

Previous work on active causal learning has mainly emphasized two aspects: (1) patient recruitment (Deng et al., 2011; Song et al., 2023) and (2) selective acquisition of costly outcomes (Jesson et al., 2021; Wen et al., 2025b) to reduce uncertainty in CATE estimation given all covariates. Our work follows the latter, assuming treatments are assigned but outcomes are expensive to obtain. In

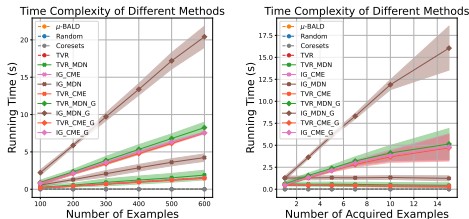

Figure 4: Running time comparisons.

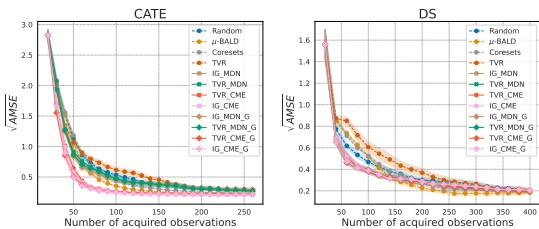

Figure 5: Performance comparison on IHDP.

contrast to prior studies, we aim to efficiently estimate a wider range of causal quantities beyond CATE. A related line of research is active causal inference, exemplified by ABCI (Toth et al., 2022), which leverages GPs and information-theoretic criteria. However, it requires interventional data and is therefore not applicable to the observational setting we consider. This direction has since been extended in follow-up works (Annadani et al., 2024; Zhou et al., 2024). Another relevant stream employs kernel-based approaches to causal inference (Sejdinovic, 2025), where conditional distributions are embedded in an RKHS rather than explicitly estimated, yielding closed-form solutions for multiple causal quantities (Singh et al., 2024; Mastouri et al., 2021). Relatedly, BayesIMP (Chau et al., 2021b) addresses uncertainty quantification with a focus on data fusion. Further discussion of transductive active learning, active GP methods, and comparisons with ABCI is provided in App. B.

## 7   CONCLUSION

In this paper, we introduced and formalized the task of *Active Estimation of Causal Quantities (ActiveCQ)*, broadening the field's focus beyond the commonly studied CATE. We proposed a unified Bayesian framework for this new research direction that represents CQs as integrals, pairing a Gaussian process for the regression function with conditional mean embeddings for the target distribution. This powerful combination allows for the model's adaptive refinement and enables the systematic derivation of uncertainty-aware utility functions. We demonstrated this by instantiating two strategies based on information gain and total variance reduction. Through simulations and semi-synthetic datasets, we showed that our principled approach significantly outperforms traditional active learning baselines, including BALD and vanilla TVR. Our work lays a foundation for numerous future directions. While our GP-based instantiation inherits cubic complexity, the framework's modularity is a key strength, inviting future work that integrates scalable approximations like sparse GPs or Nyström methods. Furthermore, our proposed framework can be extended to handle more complex causal settings involving hidden confounders or instrumental variables.

## 8   ACKNOWLEDGMENTS

This work is supported by the Responsible AI Research Centre (RAIR). EG is also supported by the Centre for Augmented Reasoning (CAR).

## ETHICS STATEMENT

This work utilizes standard public datasets and involves no human subjects, animals, or sensitive data. The research raises no specific ethical, privacy, or bias concerns and strictly adheres to the ICLR Code of Ethics.

## REPRODUCIBILITY STATEMENT

We have taken several measures to ensure the reproducibility of our work. A detailed description of the datasets and preprocessing steps is provided in App. F.1. The implementation details of our method, including model architectures, training procedures, baselines, as well as comprehensive experimental settings and hyperparameter choices, are documented in App. F.2 and App. D. An in-depth analysis of the experimental results is included in App. F.

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

# Appendix

## Table of Contents

## A  ACKNOWLEDGMENT OF LLM USAGE

In preparing this manuscript, LLMs were employed solely as general-purpose writing aids. Their use was limited to word- and sentence-level polishing, including correcting typos, improving grammar, and refining phrasing. LLMs were not used for research ideation, scientific analysis, generation of results, or interpretation of findings. All scientific concepts, methods, analyses, and conclusions presented in this work are entirely the responsibility of the authors.

## B  MORE RELATED WORKS

### B.1  RELATED WORKS (CONTINUING)

Building on the idea of learning causal quantities in RKHS, several works have extended this approach to more complex settings. For instance, Singh et al. (2019) and Mastouri et al. (2021), as well as Ren et al. (2024), address scenarios involving unmeasured confounders by leveraging instrumental variables and proxy variables, respectively. Additionally, Zhu et al. (2022) focus on settings with measurement error in treatments, while Singh et al. (2021) explore applications in sequential data. Other advancements include kernel-based methods for causal hypothesis testing (Hu et al., 2024) and modeling distributional treatment effects (Park et al., 2021). BayesIMP (Chau et al., 2021b) introduced the mean process technique to incorporate uncertainty in causal estimation, particularly to handle dataset fusion problems. This work also investigated the uncertainty arising from learning the conditional distribution and proposed a method to integrate uncertainties across multiple stages. Subsequently, IMPspec (Dance et al., 2024) introduced the spectral representation of Hilbert spaces to address the limitations of restricted nuclear-dominant kernels identified in BayesIMP. Sejdinovic (2025) provided an overview of using kernel methods to address causal inference problems. Our framework has the potential to integrate these advanced techniques, enabling perhaps further improvements in performance. Technically, our proposed method for modeling different CQs using CME and GP is closely related to the conditional mean process (Chau et al., 2021a) and conditional Bayesian quadrature (Chen et al., 2024).

### B.2  TRANSDUCTIVE ACTIVE LEARNING

Traditional active learning (TAL) implicitly assumes that the observational data, including both the training dataset and the pool dataset, represent the overall target distribution of interest (Holzmüller et al., 2023; Kirsch, 2023). Under this assumption, information gain-based methods naturally select the most informative points from the pool dataset, aligning effectively with the learning objective. Recent research, however, has begun addressing scenarios where the target distribution differs from the observational distribution, framing this challenge as a transductive active learning problem (Hübotter et al., 2024; Smith et al., 2023; Kothawade et al., 2021). Our problem naturally aligns with TAL, enabling us to draw upon similar insights to address the challenges we face. The key distinction lies in the target quantities: while TAL typically focuses on optimizing the average performance across all data points within a distribution, our work emphasizes average metrics over specific subpopulations. However, if we shift our focus to point-specific performance metrics, such as the conditional individual treatment effect or the individual treatment effect, our framework can directly benefit from TAL insights. Furthermore, unlike previous approaches, our framework requires the estimation of conditional distributions. By representing and learning these distributions in an RKHS, we provide a more efficient and task-oriented solution. This feature-driven, task-specific approach to conditional distribution estimation not only enhances our framework but also introduces new perspectives for advancing transductive active learning research.

### B.3  IN-DEPTH DISCUSSION OF ABCI

Here, we provide a detailed comparison with the Active Bayesian Causal Inference (ABCI) method proposed by Toth et al. (2022), an active learning framework for integrated causal discovery and reasoning. ABCI is designed for settings where the causal structure is unknown and must be learned from interventional data, using information gain to select experiments and GPs to model causal mechanisms. While our work shares high-level themes with ABCI, such as the use of GPs and

information-theoretic acquisition functions, the two frameworks diverge fundamentally in their problem settings, methodological approaches, and core contributions.

**Divergent Problem Settings.** The most critical distinction lies in the problem formulation. ABCI operates in a setting where the causal graph is unknown; its primary goal is to actively select interventions to jointly learn this structure and its underlying mechanisms. The uncertainty it seeks to resolve pertains to both the causal graph and the functional relationships, which it addresses by acquiring new interventional data. In contrast, our framework assumes the causal structure is known and concentrates on the efficient estimation of causal quantities from observational data. Our approach does not generate new data through interventions. Instead, it strategically selects the most informative samples from a fixed, pre-existing pool of individuals with assigned treatments. This difference situates the two methods in parallel yet distinct lines of research. ABCI is concerned with causal discovery via intervention, whereas our work is tailored for causal estimation from observation. Consequently, ABCI is not directly applicable as a baseline in our setting. Even if adapted to assume a known graph, its core mechanism of acquiring data through new interventions is incompatible with a fixed observational context.

**Distinct Methodological Contributions.** The methodological foundations of the two frameworks are also markedly different. Our work introduces a unified Bayesian framework for actively estimating a broad spectrum of CQs, a scope that extends beyond that of ABCI. A key technical innovation in our approach is the integration of Conditional Mean Embeddings with Gaussian Processes. This combination yields flexible and principled estimators capable of targeting diverse causal queries. This modeling strategy is entirely different from that of ABCI, which uses GPs to capture individual causal mechanisms within a DAG. Furthermore, we provide a formal performance guarantee by deriving a theoretical uncertainty decay rate for our acquisition function, a contribution not present in ABCI. Therefore, although both methods leverage GPs and information-theoretic ideas, these are superficial similarities. In our framework, these tools are applied to regression-based estimation from observational data to answer specific causal queries. In ABCI, they are used to facilitate structural learning. Ultimately, the methods pursue different objectives under incompatible assumptions and are not methodologically comparable.

## C  FURTHER PRELIMINARIES AND DEFINITIONS

In this paper, upright Roman letters (e.g., v) represent graph nodes and their associated random variables; calligraphic letters (e.g., $\mathcal{V}$) denote measurable spaces; and italic letters (e.g., v $= v$) indicate specific realizations. Bold letters (e.g., $\mathbf{v}$) refer to sets of nodes or random vectors, while bold italic letters (e.g., $\mathbf{v} = \boldsymbol{v}$) are used for their realizations.

### C.1  DAG AND CAUSAL GRAPHS

**Directed Acyclic Graph (DAG).** Let $\mathcal{G} = (\mathbf{v}, \mathcal{E})$ represent a graph consisting of a set of nodes (variables) $\mathbf{v} = \{v_1, \cdots, v_p\}$, and a set of edges $\mathcal{E}$. $\mathcal{G}$ is called a DAG if it contains only directed edges ($\rightarrow$) and no directed cycles in $\mathcal{G}$. Let $p(\boldsymbol{v})$ be an observational density over $\mathbf{v}$, which is said to be *Markov compatible* with a DAG $\mathcal{G} = (\mathbf{v}, \mathbf{e})$ if it factorizes as $p(\boldsymbol{v}) = \prod_{v_i \in \mathbf{v}} p(v_i | \mathrm{pa}(v_i, \mathcal{G}))$, where $\mathrm{pa}(v_i, \mathcal{G})$ includes the values of the parent nodes of $v_i$ in $\mathcal{G}$. We also assume the *positivity* condition, meaning we restrict our attention to distributions where $p(\boldsymbol{v}) > 0$ for all valid values of $\mathbf{v}$.

**Causal DAGs.** Causality is designed to describe physical processes in the real world, and DAGs serve as a powerful framework for formally representing this concept (Pearl, 2009). When $\mathcal{G}$ is a DAG, it is considered a *causal DAG* if every directed edge $v_i \rightarrow v_j$ represents a direct causal effect of $v_i$ on $v_j$. Let $\mathbf{a} \subseteq \mathbf{v}$ be a node set in a causal DAG $\mathcal{G}$. The intervention $\mathrm{do}(\mathbf{a} = \boldsymbol{a})$ [2], or $\mathrm{do}(\boldsymbol{a})$ for short, denotes an outside intervention that sets $\mathbf{a}$ to fixed values $\boldsymbol{a}$. An interventional density $p(\boldsymbol{v} | \mathrm{do}(\boldsymbol{a}))$ is a density resulting from such an intervention. Let $\mathcal{P}^*$ represent the set of all interventional densities $p(\boldsymbol{v} | \mathrm{do}(\boldsymbol{a}))$. We say that a graph $\mathcal{G} = (\mathbf{v}, \mathcal{E})$ is compatible with $\mathcal{P}^*$ if and

---

[2]In the main paper, we restrict the treatment variable to a single dimension, whereas in the appendix, this constraint is relaxed to multiple variables.

only if, for every $p(\boldsymbol{v}|\mathrm{do}(\boldsymbol{a})) \in \mathcal{P}^*$, the following condition holds:

$$p(\boldsymbol{v}|\mathrm{do}(\boldsymbol{a})) = \prod_{\mathrm{v}_i \in \mathbf{v} \setminus \mathbf{a}} p(v_i|\mathrm{pa}(v_i, \mathcal{G}))\mathbb{1}(\mathbf{a} = \boldsymbol{a}). \tag{14}$$

We say that an interventional density $p(\boldsymbol{v}|\mathrm{do}(\boldsymbol{a}))$ is consistent with $\mathcal{G}$ if it belongs to a set of interventional densities $\mathcal{P}^*$, where $\mathcal{G}$ is compatible with $\mathcal{P}^*$ (Pearl, 2009). Moreover, any observational density that is Markov compatible with $\mathcal{G}$ is also consistent with $\mathcal{G}$ (LaPlante & Perkovic, 2024).

We then present a set of standard assumptions to establish the identifiability of the causal quantities discussed in the main paper, including the ATE, CATE, ATT, and ATEDS. It is important to note that the assumptions provided here are sufficient but not necessary conditions for identifying these causal quantities, with a primary focus on scenarios where unmeasured confounding is absent and a valid adjustment set is available. Beyond these assumptions, alternative frameworks such as instrumental variable assumptions (Xu et al., 2021), front-door criteria (Chau et al., 2021b), and proxy variable approaches (Mastouri et al., 2021) can also enable identifiability in certain cases. However, these are beyond the current scope of our work. Extending our framework to accommodate such conditions remains an interesting direction for future research. Throughout this paper, we make the following assumptions:

**Assumption 2** (Stable Unit Treatment Value Assumption (SUTVA)). *The relationship between the observed outcome* $\mathbf{y}$ *and the potential outcomes is stable. This entails two key conditions:*

1. ***No Interference:*** *The potential outcome of any individual unit depends only on the treatment assigned to that unit and is unaffected by the treatment assignments of other units.*

2. ***Consistency:*** *For any unit, if the treatment* $\mathbf{a}$ *is set to a value* $\boldsymbol{a}$*, the observed outcome* $\mathbf{y}$ *equals the potential outcome* $\mathbf{y}(\boldsymbol{a})$*. Formally,* $\mathbf{y} = \mathbf{y}(\boldsymbol{a})$ *if* $\mathbf{a} = \boldsymbol{a}$*.*

*This assumption ensures that the mapping from treatment to outcome is well-defined and allows us to link observed data to theoretical causal quantities.*

**Assumption 3** (Causal Markov Compatibility). *The joint distribution of the variables is consistent with a causal DAG* $\mathcal{G}$*, meaning:*

$$p(\boldsymbol{v}|do(\boldsymbol{a})) = \prod_{\mathrm{v}_i \in \mathbf{v} \setminus \mathbf{a}} p(v_i|pa(v_i, \mathcal{G}))\mathbb{1}(\mathbf{a} = \boldsymbol{a}). \tag{15}$$

*Here,* $pa(v_i, \mathcal{G})$ *denotes the set of parent variables of* $\mathrm{v}_i$ *in the graph* $\mathcal{G}$*.*

**Assumption 4** (Backdoor Criterion). *There exists a set of variables* $\mathbf{s} \subseteq \mathbf{v}$ *(the backdoor adjustment set) such that no element of* $\mathbf{s}$ *is a descendant of* $\mathbf{a}$*, and* $\mathbf{s}$ *blocks every path from* $\mathbf{a}$ *to* $\mathbf{y}$ *that begins with an arrow into* $\mathbf{a}$ *in the graph* $\mathcal{G}$*. This ensures that adjusting for* $\mathbf{s}$ *removes confounding bias in the causal effect of* $\mathbf{a}$ *on* $\mathbf{y}$*.*

**Assumption 5** (Positivity). *The treatment assignment is well-defined for all levels of the adjustment set* $\mathbf{s}$*. Formally:*

$$p(\mathbf{a} = \boldsymbol{a}|\mathbf{s} = \boldsymbol{s}) > 0 \quad \forall \boldsymbol{a}, \boldsymbol{s}, \tag{16}$$

*where* $\boldsymbol{s}$ *is a realization of the adjustment set* $\mathbf{s}$*. This assumption ensures that every treatment level* $\boldsymbol{a}$ *has a non-zero probability of being observed for all values of* $\mathbf{s}$*. Notice that for the CATE case, the adjustment set effectively encompasses both* $\mathbf{s}$ *and* $\mathbf{z}$*.*

For the case of ATE with distribution shift, it is essential to impose constraints on the shifted distribution $\tilde{\mathbb{P}}$ to ensure the identifiability of this causal quantity. In addition to the assumptions outlined previously, we further introduce the following assumption:

**Assumption 6** (Distribution shift (Singh et al., 2024)). *Assume the following:*

1. *The joint distribution of* $\mathbf{y}$ *and* $\mathbf{s}$ *under the shifted distribution is given by:*

$$\tilde{\mathbb{P}}(\mathbf{y}, \mathbf{a}, \mathbf{s}) = \mathbb{P}(\mathbf{y}|\mathbf{a}, \mathbf{s})\tilde{\mathbb{P}}(\mathbf{a}, \mathbf{s}). \tag{17}$$

2. *The shifted marginal distribution* $\tilde{\mathbb{P}}(\mathbf{a}, \mathbf{s})$ *is absolutely continuous with respect to the original distribution* $\mathbb{P}(\mathbf{a}, \mathbf{s})$*.*

Under the above assumption, populations $\mathbb{P}$ and $\tilde{\mathbb{P}}$ differ only in the distribution of the treatment variable and covariates. Furthermore, the support of $\mathbb{P}$ encompasses the support of $\tilde{\mathbb{P}}$, ensuring that the treatment effect is well-defined across both populations. As a direct implication, the regression function remains invariant between $\mathbb{P}$ and $\tilde{\mathbb{P}}$, regardless of the distributional shift.

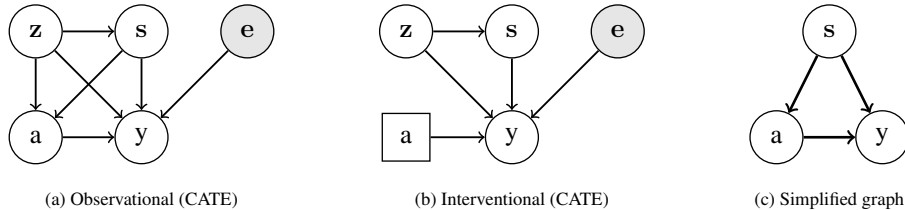

(a) Observational (CATE)          (b) Interventional (CATE)          (c) Simplified graph

Figure 6: Data generation processes depicted by DAGs. (a) and (b) illustrate the full causal graph for the CATE setting under observational and interventional distributions, respectively. (c) shows the simplified graph used for ATE, ATT, and ATEDS settings.

**Difference from the PO Framework.** The Potential Outcomes (PO) and causal graph frameworks provide distinct approaches for defining causal effects, differing in their foundational principles and particularly in their handling of continuous treatments. The PO framework defines causal effects via hypothetical counterfactuals, namely the potential outcomes under different treatment scenarios (Ding, 2024). In contrast, the CG framework defines them directly from the structural relationships encoded in a graph, representing explicit causal assumptions (Pearl, 2009). While both frameworks align conceptually for binary treatments, treatment contrasts such as $\mathbb{E}[\mathbf{y}(1) - \mathbf{y}(0)]$ can be obtained by taking differences of interventional mean quantities. The CG framework generalizes more naturally. Causal quantities like the ATE and CATE are consistently defined through interventions on the graph, corresponding directly to the Average Dose-Response Function (ADRF) and its conditional version, respectively (Hernan & Robins, 2023). The PO framework, however, must extend its definition to a function of treatment doses, often requiring additional modeling assumptions like smoothness or monotonicity to estimate the dose-response curve. In essence, the primary distinction is the representation of causality: PO models counterfactual outcomes explicitly, while CG models the causal mechanisms that generate them, offering a more unified approach for both binary and continuous interventions.

**ITE, CATE, and Our CATE.** We distinguish our definition of the CATE from two related concepts in the literature: the Individual Treatment Effect (ITE) and the commonly used definition of CATE. Our entire analysis is grounded in the structural causal model framework, using the do-operator to define causal quantities. A key aspect of our approach is that we focus on defining the conditional potential outcome under a specific treatment $\mathbf{a} = \boldsymbol{a}$, rather than the contrast between two different treatments. To formalize these concepts, we consider a model (e.g., Fig. 6) that includes observed covariates $(\mathbf{s}, \mathbf{z})$ and an unobserved random vector $\mathbf{e}$. This vector $\mathbf{e}$ represents all remaining individual-level latent factors that are not confounders, ensuring that conditioning on $(\mathbf{s}, \mathbf{z}, \mathbf{e})$ fully captures an individual's characteristics. The ITE is a theoretical construct representing the causal effect for a single individual $i$. It is defined by conditioning on all their characteristics, both observed and unobserved:

$$\tau_{\text{ITE}}(\boldsymbol{a}; \text{individual } i) := \mathbb{E}[\mathbf{y} \mid \text{do}(\mathbf{a} = \boldsymbol{a}), \mathbf{s} = \boldsymbol{s}_i, \mathbf{z} = \boldsymbol{z}_i, \mathbf{e} = \boldsymbol{e}_i]. \tag{18}$$

The ITE is impossible to estimate in practice because the latent vector $\mathbf{e}_i$ is unknown. To achieve practical estimation, the literature typically focuses on the CATE, which averages over the unobserved factors $\mathbf{e}$. A common definition conditions on the full set of *observed* covariates:

$$\tau_{\text{CATE}}(\boldsymbol{a}; \boldsymbol{s}, \boldsymbol{z}) := \mathbb{E}[\mathbf{y} \mid \text{do}(\mathbf{a} = \boldsymbol{a}), \mathbf{s} = \boldsymbol{s}, \mathbf{z} = \boldsymbol{z}]. \tag{19}$$

This quantity represents the average treatment effect for the subpopulation of individuals sharing the same observed characteristics $(\boldsymbol{s}, \boldsymbol{z})$. However, this definition can be restrictive, as it compels the analysis to be conditioned on the entire set of observed covariates. In this work, we take a more flexible and practical definition of CATE. Our framework allows conditioning on any chosen subset of observed covariates, which we denote by $\mathbf{z}$. The standard definition above thus becomes a special case of our framework where $\mathbf{z}$ includes all observed covariates. This flexibility allows for the estimation of treatment effects for broader or more specific subpopulations based on the context of the analysis.

## C.2 KERNELS AND RKHS

**Reproducing Kernel Hilbert Spaces (RKHS).** Consider any spaces $\mathcal{U}, \mathcal{V} \in \{\mathcal{A}, \mathcal{Z}, \mathcal{S}, \mathcal{X}, \mathcal{Y}\}$, and let $k : \mathcal{U} \times \mathcal{U} \rightarrow \mathbb{R}$ be a positive semi-definite kernel. The canonical feature map associated with this kernel

is $\phi_{\boldsymbol{u}} := k(\boldsymbol{u}, \cdot)$ for any $\boldsymbol{u} \in \mathcal{U}$. The features matrix $\Phi_{\boldsymbol{A}}$ is constructed by stacking the feature maps of each row of $\boldsymbol{A}$ as the columns, i.e., $\Phi_{\boldsymbol{A}} := [\phi_{\boldsymbol{A}_{1,:}}, \cdots, \phi_{\boldsymbol{A}_{n,:}}]$, where $\boldsymbol{A} := [\boldsymbol{a}_1, \cdots, \boldsymbol{a}_n]^T$. We denote the Gram matrix as $\boldsymbol{K}_{\boldsymbol{A}} := \Phi_{\boldsymbol{A}}^T \Phi_{\boldsymbol{A}}$ and the vector of evaluations $\boldsymbol{k}_{\boldsymbol{a}\boldsymbol{A}}$ as $[k(\boldsymbol{a}, \boldsymbol{a}_1), \cdots, k(\boldsymbol{a}, \boldsymbol{a}_n)]$, and $\boldsymbol{k}_{\boldsymbol{A}\boldsymbol{a}} = \boldsymbol{k}_{\boldsymbol{a}\boldsymbol{A}}^T$. The RKHS spanned by this kernel is denoted by $\mathcal{H}_{\mathcal{A}}$. It consists of real-valued functions defined on $\mathcal{A}$ and is endowed with the inner product $\langle \cdot, \cdot \rangle_{\mathcal{H}_{\mathcal{A}}}$. We use $\otimes$ and $\odot$ to represent tensor product and Hadamard product respectively and use $\mathcal{H}_{\mathcal{A}\mathcal{B}}$ to represent the tensor product space $\mathcal{H}_{\mathcal{A}} \otimes \mathcal{H}_{\mathcal{B}}$. For any distribution $\mathbb{P}_{\mathbf{a}}$ on $\mathcal{A}$, we define the kernel mean embedding of $\mathbb{P}_{\mathbf{a}}$ as $\mu_{\mathbf{a}} := \int_{\mathcal{A}} k(\boldsymbol{a}, \cdot) \mathbb{P}_{\mathbf{a}}(d\boldsymbol{a}) \in \mathcal{H}_{\mathcal{A}}$, which also corresponds to the Riesz representer of expectational functional $f \mapsto \mathbb{E}[f(\mathbf{a})] = \langle f, \mu_{\mathbf{a}} \rangle_{\mathcal{H}_{\mathcal{A}}}$, due to the reproducing property that $\langle f, k(\boldsymbol{a}, \cdot) \rangle_{\mathcal{H}_{\mathcal{A}}} = f(\boldsymbol{a})$ for all $f \in \mathcal{H}_{\mathcal{A}}$. Extending to the conditional distribution, for example $\mathbb{P}_{\mathbf{a}|\mathbf{b}=\boldsymbol{b}}$, we define $\mu_{\mathbf{a}|\boldsymbol{b}} := \int_{\mathcal{A}} k(\boldsymbol{a}, \cdot) \mathbb{P}_{\mathbf{a}|\mathbf{b}}(d\boldsymbol{a}|\boldsymbol{b}) \in \mathcal{H}_{\mathcal{A}}$ as the conditional mean embedding of $\mathbb{P}_{\mathbf{a}|\mathbf{b}=\boldsymbol{b}}$.

**The tensor product RKHS** expands the traditional RKHS framework to effectively manage functions with multiple arguments. Given two RKHSs, $\mathcal{H}_{\mathcal{A}}$ and $\mathcal{H}_{\mathcal{B}}$, associated with kernels $k(\boldsymbol{a}, \cdot)$ and $k(\boldsymbol{b}, \cdot)$, the tensor product RKHS $\mathcal{H} = \mathcal{H}_{\mathcal{A}} \otimes \mathcal{H}_{\mathcal{B}}$ creates a space of functions defined over the Cartesian product $\mathcal{A} \times \mathcal{B}$, where $\mathcal{A}$ and $\mathcal{B}$ represent the input spaces of the respective RKHSs. The kernel of this tensor product space is the product of the individual kernels:

$$k_{\mathcal{A} \otimes \mathcal{B}}((\boldsymbol{a}, \boldsymbol{b}), (\boldsymbol{a}', \boldsymbol{b}')) = k_{\mathcal{A}}(\boldsymbol{a}, \boldsymbol{a}') \cdot k_{\mathcal{B}}(\boldsymbol{b}, \boldsymbol{b}'), \tag{20}$$

where $(\boldsymbol{a}, \boldsymbol{b}), (\boldsymbol{a}', \boldsymbol{b}') \in \mathcal{A} \times \mathcal{B}$. The feature map for this space is the tensor product of the individual feature maps, $\phi(\boldsymbol{a}, \boldsymbol{b}) = \phi_{\mathcal{A}}(\boldsymbol{a}) \otimes \phi_{\mathcal{B}}(\boldsymbol{b})$. The norm is multiplicative for elementary tensors: $\|\phi_{\mathcal{A}}(\boldsymbol{a}) \otimes \phi_{\mathcal{B}}(\boldsymbol{b})\|_{\mathcal{H}_{\mathcal{A}} \otimes \mathcal{H}_{\mathcal{B}}} = \|\phi_{\mathcal{A}}(\boldsymbol{a})\|_{\mathcal{H}_{\mathcal{A}}} \cdot \|\phi_{\mathcal{B}}(\boldsymbol{b})\|_{\mathcal{H}_{\mathcal{B}}}$. This framework is essential for modeling functions $f : \mathcal{A} \times \mathcal{B} \to \mathbb{R}$ by capturing both the main effects of the variables and their interactions.

## C.3 INFORMATION-THEORETIC MEASURES

This section provides concise definitions for the information-theoretic quantities used in our work.

**Differential Entropy.** For a continuous random variable $\mathbf{a}$ with probability density function (PDF) $p(\boldsymbol{a})$ over its support $\mathcal{A}$, the differential entropy quantifies its uncertainty:

$$\mathrm{H}(\mathbf{a}) := \int_{\mathcal{A}} -\log p(\boldsymbol{a}) \, d\mathbb{P}_{\mathbf{a}}(\boldsymbol{a}). \tag{21}$$

Unlike discrete entropy, differential entropy can be negative and depends on the coordinate system.

**Conditional Entropy.** The conditional entropy $\mathrm{H}(\mathbf{a} \mid \mathbf{b})$ measures the remaining uncertainty in $\mathbf{a}$ given $\mathbf{b}$:

$$\mathrm{H}(\mathbf{a} \mid \mathbf{b}) := \int_{\mathcal{A} \times \mathcal{B}} -\log p(\boldsymbol{a} \mid \boldsymbol{b}) \, \mathbb{P}_{\mathbf{a}\mathbf{b}}(d\boldsymbol{a}, d\boldsymbol{b}). \tag{22}$$

**Mutual Information.** The mutual information $\mathrm{I}(\mathbf{a}; \mathbf{b})$ quantifies the reduction in uncertainty about one variable from observing the other. It is defined as:

$$\mathrm{I}(\mathbf{a}; \mathbf{b}) := \mathrm{H}(\mathbf{a}) - \mathrm{H}(\mathbf{a} \mid \mathbf{b}), \tag{23}$$

which is symmetric and can also be expressed using the joint entropy $\mathrm{H}(\mathbf{a}, \mathbf{b})$:

$$\mathrm{I}(\mathbf{a}; \mathbf{b}) = \mathrm{H}(\mathbf{a}) + \mathrm{H}(\mathbf{b}) - \mathrm{H}(\mathbf{a}, \mathbf{b}). \tag{24}$$

These measures are fundamental to acquisition functions like BALD (Houlsby et al., 2011), which seeks to maximize the mutual information between model parameters and unknown labels.

## D MODELING FOR VARIOUS CAUSAL QUANTITIES

This section expands on the modeling of various CQs. As established in the main paper, our framework for the CATE integrates several key components. We began by modeling the outcome regression surface with a GP. The central challenge, evaluating the integral in the CATE definition, was addressed by leveraging CMEs to represent the conditional distribution $\mathbb{P}_{\mathbf{s}|\mathbf{z}}$. This technique provides an analytic, closed-form posterior distribution for the CATE estimator, which is crucial for quantifying uncertainty and enabling effective active learning. Building on this foundation, we now extend the methodology to other key causal quantities. We will detail the specific modeling adaptations and estimator formulations for the ATE, ATT, and ATEDS.

## D.1 Derivation for the CATE

Here, we provide a detailed, step-by-step derivation for the posterior mean $\nu(a, z)$ and covariance $q((a, z), (a', z'))$ of the CATE estimator, as presented in Prop. 1. The entire derivation is based on expressing the standard GP posterior formulas in terms of inner products in the RKHS. The key insight is the use of an *effective feature map* $\phi_{\bar{x}} := \phi_a \otimes \phi_z \otimes \hat{\mu}_{s|z=z}$, which allows the CATE estimation (an expectation over $s$) to be treated as a standard prediction problem.

Let $\Phi_{X_T} := [\phi_{a_i} \otimes \phi_{z_i} \otimes \phi_{s_i}]_{i=1}^{n_T}$ denote the feature matrix for the paired training inputs $X_T = \{(a_i, z_i, s_i)\}_{i=1}^{n_T}$. The following equations show the expansion of these inner products.

- The derivation for $\nu(a, z)$ begins with the abstract inner product between the effective feature map and the GP posterior mean function, $\langle \phi_{\bar{x}}, m_f \rangle_{\mathcal{H}_{\mathcal{AZS}}}$.
- Similarly, the derivation for $q((a, z), (a', z'))$ starts with the abstract formula involving the prior kernel, $\langle \phi_{\bar{x}}, \phi_{\bar{x}'} \rangle$, and the update term, which depends on the cross-covariance inner products $\langle \phi_{\bar{x}}, \Phi_{X_T} \rangle$.

The subsequent steps in the equations expand these abstract inner products into concrete matrix-vector operations. The notation $(\dots)^T(\dots)$ is used to represent these operations in feature space. The underbrace annotations explicitly identify how these expanded expressions correspond to the final, compact terms like the effective cross-covariance vector $k_{\bar{x} X_T}$, the effective kernel $k_{\bar{x}\bar{x}'}$, and the full training Gram matrix $K_{X_T X_T}$.

$$
\begin{aligned}
\nu(a, z) &= \langle \phi_{\bar{x}}, m_f \rangle_{\mathcal{H}_{\mathcal{AZS}}} \\
&= \phi_{\bar{x}}^T \Phi_{X_T} (K_{X_T X_T} + \sigma^2 I)^{-1} y_T \\
&= \underbrace{\left( k_{a a_T} \odot k_{z Z_T} \odot \left( k_{zZ} (K_{ZZ} + \lambda I)^{-1} K_{SS_T} \right) \right)}_{k_{\bar{x} X_T}} \\
&\quad \times \underbrace{\left( (K_{a_T a_T} \odot K_{Z_T Z_T} \odot K_{S_T S_T}) + \sigma^2 I \right)^{-1}}_{(K_{X_T X_T} + \sigma^2 I)^{-1}} y_T \\
&= k_{\bar{x} X_T} (K_{X_T X_T} + \sigma^2 I)^{-1} y_T,
\end{aligned}
\tag{25}
$$

$$
\begin{aligned}
q((a, z), (a', z')) &= \langle \phi_{\bar{x}}, \phi_{\bar{x}'} \rangle - \langle \phi_{\bar{x}}, \Phi_{X_T} \rangle (K_{X_T X_T} + \sigma^2 I)^{-1} \langle \Phi_{X_T}, \phi_{\bar{x}'} \rangle \\
&= \underbrace{\left( k_{aa'} k_{zz'} \left( k_{zZ} (K_{ZZ} + \lambda I)^{-1} K_{SS} (K_{ZZ} + \lambda I)^{-1} k_{Zz'} \right) \right)}_{k_{\bar{x}\bar{x}'}} \\
&\quad - \underbrace{\left( k_{a a_T} \odot k_{z Z_T} \odot \left( k_{zZ} (K_{ZZ} + \lambda I)^{-1} K_{SS_T} \right) \right)}_{k_{\bar{x} X_T}} \\
&\quad \times \underbrace{\left( (K_{a_T a_T} \odot K_{Z_T Z_T} \odot K_{S_T S_T}) + \sigma^2 I \right)^{-1}}_{(K_{X_T X_T} + \sigma^2 I)^{-1}} \\
&\quad \times \underbrace{\left( k_{a_T a'} \odot k_{Z_T z'} \odot \left( K_{S_T S} (K_{ZZ} + \lambda I)^{-1} k_{Zz'} \right) \right)}_{k_{X_T \bar{x}'}} \\
&= k_{\bar{x}\bar{x}'} - k_{\bar{x} X_T} (K_{X_T X_T} + \sigma^2 I)^{-1} k_{X_T \bar{x}'}.
\end{aligned}
\tag{26}
$$

The parameter $\lambda > 0$ is the regularization parameter for the CME, and $\sigma^2 > 0$ is the GP observation noise variance.

## D.2 Derivation for ATE

**Definition and Estimation Strategy.** In this subsection, we detail the modeling and posterior derivation for the ATE. The ATE for a specific treatment $a$ is defined as the expected outcome,

marginalized over the entire population distribution of the covariates $\mathbf{s}$:

$$\tau_{\text{ATE}}(a) := \int_{\mathcal{S}} \mathbb{E}[\mathbf{y} \mid \mathbf{a} = a, \mathbf{s} = \boldsymbol{s}] \, \mathbb{P}_{\mathbf{s}}(d\boldsymbol{s}). \tag{27}$$

We model the response surface $\mathbb{E}[\mathbf{y} \mid \mathbf{a} = a, \mathbf{s} = \boldsymbol{s}]$ with a GP $f(a, \boldsymbol{s})$. The primary challenge is to evaluate the integral over the marginal distribution $\mathbb{P}_{\mathbf{s}}$. Unlike CATE, which requires a conditional distribution, we can directly estimate $\mathbb{P}_{\mathbf{s}}$ from all available samples in our dataset $\mathcal{D} = \mathcal{D}_T \cup \mathcal{D}_P$. Our strategy is to represent the distribution $\mathbb{P}_{\mathbf{s}}$ in the RKHS using its Mean Embedding (ME), $\mu_{\mathbf{s}} = \mathbb{E}_{\mathbf{s}}[\phi(\boldsymbol{s})]$. We form an empirical estimate of the ME from all $n$ individuals in $\mathcal{D}$:

$$\hat{\mu}_{\mathbf{s}} = \frac{1}{n} \sum_{i=1}^{n} \phi(\boldsymbol{s}_i). \tag{28}$$

This allows the ATE estimator to be elegantly expressed as an inner product in the tensor product RKHS $\mathcal{H}_{\mathcal{AS}}$:

$$\hat{\tau}_{\text{ATE}}(a) = \langle \mathrm{f}, \phi(a) \otimes \hat{\mu}_{\mathbf{s}} \rangle_{\mathcal{H}_{\mathcal{AS}}}. \tag{29}$$

This formulation is powerful because it converts the integration problem into a standard GP prediction problem at an "effective" input point.

**Posterior Distribution.** Based on the inner product formulation, the posterior distribution of the ATE estimator is also a GP. The following proposition summarizes its posterior mean and covariance.

**Proposition 2** (ATE Estimator Posterior). *Given the training dataset $\mathcal{D}_T = \{\boldsymbol{a}_T, \boldsymbol{S}_T, \boldsymbol{y}_T\}$ and the full set of inputs $\mathcal{D} = \{\boldsymbol{a}, \boldsymbol{S}\}$, let $n = |\mathcal{D}|$. If $f(a, \boldsymbol{s})$ is the posterior GP learned from $\mathcal{D}_T$, then the ATE estimator $\hat{\tau}_{\text{ATE}}(a)$ follows a GP, $\hat{\tau}_{\text{ATE}} \sim \mathcal{GP}(\nu(a), q(a, a'))$. Its posterior mean and covariance are derived using an effective feature map $\phi_{\bar{a}} := \phi(a) \otimes \hat{\mu}_{\mathbf{s}}$.*

**Detailed Derivation.** The following equations show the step-by-step expansion of the GP posterior formulas.

$$\nu(a) = \langle \phi_a \otimes \hat{\mu}_{\mathbf{s}}, m_{\mathrm{f}} \rangle_{\mathcal{H}_{\mathcal{AS}}}$$
$$= \underbrace{\left( \boldsymbol{k}_{a\boldsymbol{a}_T} \odot \frac{1}{n} \mathbf{1}_n^T \boldsymbol{K}_{\boldsymbol{S}\boldsymbol{S}_T} \right)}_{\boldsymbol{k}_{\bar{a}\boldsymbol{X}_T}} \underbrace{\left( (\boldsymbol{K}_{\boldsymbol{a}_T\boldsymbol{a}_T} \odot \boldsymbol{K}_{\boldsymbol{S}_T\boldsymbol{S}_T}) + \sigma^2 \boldsymbol{I} \right)^{-1}}_{(\boldsymbol{K}_{\boldsymbol{X}_T\boldsymbol{X}_T} + \sigma^2\boldsymbol{I})^{-1}} \boldsymbol{y}_T \tag{30}$$
$$= \boldsymbol{k}_{\bar{a}\boldsymbol{X}_T} (\boldsymbol{K}_{\boldsymbol{X}_T\boldsymbol{X}_T} + \sigma^2 \boldsymbol{I})^{-1} \boldsymbol{y}_T,$$

$$q(a, a') = k_{\bar{a}\bar{a}'} - \boldsymbol{k}_{\bar{a}\boldsymbol{X}_T} (\boldsymbol{K}_{\boldsymbol{X}_T\boldsymbol{X}_T} + \sigma^2 \boldsymbol{I})^{-1} \boldsymbol{k}_{\boldsymbol{X}_T\bar{a}'}$$
$$= \underbrace{k_{aa'} \left( \frac{1}{n^2} \mathbf{1}_n^T \boldsymbol{K}_{\boldsymbol{S}\boldsymbol{S}} \mathbf{1}_n \right)}_{k_{\bar{a}\bar{a}'}}$$
$$- \underbrace{\left( \boldsymbol{k}_{a\boldsymbol{a}_T} \odot \frac{1}{n} \mathbf{1}_n^T \boldsymbol{K}_{\boldsymbol{S}\boldsymbol{S}_T} \right)}_{\boldsymbol{k}_{\bar{a}\boldsymbol{X}_T}} \left( (\boldsymbol{K}_{\boldsymbol{a}_T\boldsymbol{a}_T} \odot \boldsymbol{K}_{\boldsymbol{S}_T\boldsymbol{S}_T}) + \sigma^2 \boldsymbol{I} \right)^{-1} \underbrace{\left( \boldsymbol{k}_{\boldsymbol{a}_T a'} \odot \frac{1}{n} \boldsymbol{K}_{\boldsymbol{S}_T\boldsymbol{S}} \mathbf{1}_n \right)}_{\boldsymbol{k}_{\boldsymbol{X}_T\bar{a}'}}. \tag{31}$$

Here, $\boldsymbol{K}_{\boldsymbol{X}_T\boldsymbol{X}_T} = \boldsymbol{K}_{\boldsymbol{a}_T\boldsymbol{a}_T} \odot \boldsymbol{K}_{\boldsymbol{S}_T\boldsymbol{S}_T}$, $\mathbf{1}_n$ is a column vector of $n$ ones, and $\sigma^2 > 0$ is the GP observation noise variance. For simplicity in notation, we use $\mathbf{1}^T$ where the dimension is clear from context.

## D.3  DERIVATION FOR ATT

**Definition and Estimation Strategy.** This subsection details the modeling and posterior derivation for the ATT. The ATT evaluates the effect of a new treatment $a$ on the specific subpopulation that had previously received a treatment $\tilde{a}$. For simplicity, we omit the context variable $\mathbf{z}$ in this derivation. The ATT is formally defined as:

$$\tau_{\text{ATT}}(a, \tilde{a}) := \int_{\mathcal{S}} \mathbb{E}[\mathbf{y} \mid \mathbf{a} = a, \mathbf{s} = \boldsymbol{s}] \, \mathbb{P}_{\mathbf{s}|\mathbf{a}}(d\boldsymbol{s}|\tilde{a}). \tag{32}$$

We model the response surface $\mathbb{E}[\mathbf{y} \mid a = a, \mathbf{s} = s]$ with a GP $f(a, s)$. The central challenge is to handle the integral over the conditional distribution $\mathbb{P}_{\mathbf{s}|a=\tilde{a}}$. Analogous to the CATE case, we represent this conditional distribution in the RKHS using its CME, $\mu_{\mathbf{s}|\tilde{a}} = \mathbb{E}_{\mathbf{s}|a=\tilde{a}}[\phi(s)]$.

This allows the ATT estimator to be expressed as an inner product in the tensor product RKHS $\mathcal{H}_{\mathcal{AS}}$:

$$\hat{\tau}_{\text{ATT}}(a, \tilde{a}) = \langle \text{f}, \phi(a) \otimes \hat{\mu}_{\mathbf{s}|\tilde{a}} \rangle_{\mathcal{H}_{\mathcal{AS}}}, \tag{33}$$

where $\hat{\mu}_{\mathbf{s}|\tilde{a}}$ is the empirical estimate of the CME. This formulation converts the integration problem into a standard GP prediction at an "effective" input point.

**Posterior Distribution.** Based on this formulation, the posterior distribution of the ATT estimator is also a Gaussian Process. We summarize the result in the following proposition.

**Proposition 3** (ATT Estimator Posterior). *Given the training dataset $\mathcal{D}_T = \{\boldsymbol{a}_T, \boldsymbol{S}_T, \boldsymbol{y}_T\}$ and the full input set $\mathcal{D} = \{\boldsymbol{a}, \boldsymbol{S}\}$, if $f(a, s)$ is the posterior GP learned from $\mathcal{D}_T$, then the ATT estimator $\hat{\tau}_{\text{ATT}}(a, \tilde{a})$ follows a GP, $\hat{\tau}_{\text{ATT}} \sim \mathcal{GP}(\nu(a, \tilde{a}), q((a, \tilde{a}), (a', \tilde{a}')))$. Its posterior mean and covariance are derived using an effective feature map $\phi_{\bar{a}} \coloneqq \phi(a) \otimes \hat{\mu}_{\mathbf{s}|\tilde{a}}$.*

**Detailed Derivation.** The following equations show the step-by-step expansion of the GP posterior formulas.

$$\begin{aligned}
\nu(a, \tilde{a}) &= \langle \phi_a \otimes \hat{\mu}_{\mathbf{s}|\tilde{a}}, m_{\text{f}} \rangle_{\mathcal{H}_{\mathcal{AS}}} \\
&= \underbrace{\left( \boldsymbol{k}_{a\boldsymbol{a}_T} \odot \left( \boldsymbol{k}_{\tilde{a}\boldsymbol{a}}(\boldsymbol{K}_{\boldsymbol{a}\boldsymbol{a}} + \lambda \boldsymbol{I})^{-1} \boldsymbol{K}_{\boldsymbol{S}\boldsymbol{S}_T} \right) \right)}_{\boldsymbol{k}_{\bar{a}\boldsymbol{X}_T}} \\
&\quad \times \underbrace{\left( (\boldsymbol{K}_{\boldsymbol{a}_T\boldsymbol{a}_T} \odot \boldsymbol{K}_{\boldsymbol{S}_T\boldsymbol{S}_T}) + \sigma^2 \boldsymbol{I} \right)^{-1}}_{(\boldsymbol{K}_{\boldsymbol{X}_T\boldsymbol{X}_T} + \sigma^2 \boldsymbol{I})^{-1}} \boldsymbol{y}_T \\
&= \boldsymbol{k}_{\bar{a}\boldsymbol{X}_T} (\boldsymbol{K}_{\boldsymbol{X}_T\boldsymbol{X}_T} + \sigma^2 \boldsymbol{I})^{-1} \boldsymbol{y}_T,
\end{aligned} \tag{34}$$

$$\begin{aligned}
q\left((a, \tilde{a}), (a', \tilde{a}')\right) &= k_{\bar{a}\bar{a}'} - \boldsymbol{k}_{\bar{a}\boldsymbol{X}_T}(\boldsymbol{K}_{\boldsymbol{X}_T\boldsymbol{X}_T} + \sigma^2 \boldsymbol{I})^{-1} \boldsymbol{k}_{\boldsymbol{X}_T\bar{a}'} \\
&= k_{aa'} \underbrace{\left( \boldsymbol{k}_{\tilde{a}\boldsymbol{a}}(\boldsymbol{K}_{\boldsymbol{a}\boldsymbol{a}} + \lambda \boldsymbol{I})^{-1} \boldsymbol{K}_{\boldsymbol{S}\boldsymbol{S}}(\boldsymbol{K}_{\boldsymbol{a}\boldsymbol{a}} + \lambda \boldsymbol{I})^{-1} \boldsymbol{k}_{\boldsymbol{a}\tilde{a}'} \right)}_{k_{\bar{a}\bar{a}'}} \\
&\quad - \underbrace{\left( \boldsymbol{k}_{a\boldsymbol{a}_T} \odot \left( \boldsymbol{k}_{\tilde{a}\boldsymbol{a}}(\boldsymbol{K}_{\boldsymbol{a}\boldsymbol{a}} + \lambda \boldsymbol{I})^{-1} \boldsymbol{K}_{\boldsymbol{S}\boldsymbol{S}_T} \right) \right)}_{\boldsymbol{k}_{\bar{a}\boldsymbol{X}_T}} \\
&\quad \times \left( (\boldsymbol{K}_{\boldsymbol{a}_T\boldsymbol{a}_T} \odot \boldsymbol{K}_{\boldsymbol{S}_T\boldsymbol{S}_T}) + \sigma^2 \boldsymbol{I} \right)^{-1} \\
&\quad \times \underbrace{\left( \boldsymbol{k}_{\boldsymbol{a}_T a'} \odot \left( \boldsymbol{K}_{\boldsymbol{S}_T\boldsymbol{S}}(\boldsymbol{K}_{\boldsymbol{a}\boldsymbol{a}} + \lambda \boldsymbol{I})^{-1} \boldsymbol{k}_{\boldsymbol{a}\tilde{a}'} \right) \right)}_{\boldsymbol{k}_{\boldsymbol{X}_T\bar{a}'}}.
\end{aligned} \tag{35}$$

Here, $\boldsymbol{K}_{\boldsymbol{X}_T\boldsymbol{X}_T} = \boldsymbol{K}_{\boldsymbol{a}_T\boldsymbol{a}_T} \odot \boldsymbol{K}_{\boldsymbol{S}_T\boldsymbol{S}_T}$. The parameter $\lambda > 0$ is the regularization parameter for the CME, and $\sigma^2 > 0$ is the GP observation noise variance. Note that for the CME, the conditioning is on the treatment variable a, hence kernel matrices like $\boldsymbol{K}_{\boldsymbol{a}\boldsymbol{a}}$ are used for its estimation.

## D.4 Derivation for ATEDS

**Definition and Estimation Strategy.** This subsection details the modeling and posterior derivation for the ATEDS. This quantity evaluates the effect of a treatment $a$ over a new target population, whose covariate distribution $\tilde{\mathbb{P}}_{\mathbf{s}}$ may differ from the source population $\mathbb{P}_{\mathbf{s}}$ where the model is trained. The ATEDS is formally defined as:

$$\tau_{\text{ATEDS}}(a) \coloneqq \int_{\mathcal{S}} \mathbb{E}[\mathbf{y} \mid a = a, \mathbf{s} = s] \, \tilde{\mathbb{P}}_{\mathbf{s}}(ds). \tag{36}$$

The estimation challenge is that the regression function $f(a, s) = \mathbb{E}[\mathbf{y} \mid a = a, \mathbf{s} = s]$ is learned from the source population data $\mathcal{D}_T$, but the integration is performed over the target population distribution

$\tilde{\mathbb{P}}_{\mathbf{s}}$. We assume we have access to a set of i.i.d. samples $\{\tilde{s}_i\}_{i=1}^{n_{\tilde{s}}}$ from the target population, which we denote as $\tilde{\mathcal{D}} = \{\tilde{S}\}$.

Instead of learning the density $\tilde{p}(s)$, we can directly use these samples to estimate the integral. We represent the target distribution $\tilde{\mathbb{P}}_{\mathbf{s}}$ via its empirical ME, estimated from the target samples:

$$\hat{\tilde{\mu}}_{\mathbf{s}} = \frac{1}{n_{\tilde{s}}} \sum_{i=1}^{n_{\tilde{s}}} \phi(\tilde{s}_i). \tag{37}$$

The ATEDS estimator can then be expressed as an inner product in the RKHS:

$$\hat{\tau}_{\text{ATEDS}}(a) = \langle \text{f}, \phi(a) \otimes \hat{\tilde{\mu}}_{\mathbf{s}} \rangle_{\mathcal{H}_{\mathcal{AS}}}. \tag{38}$$

This formulation elegantly handles the distribution shift by converting the problem into a standard GP prediction at an "effective" input point that encodes the target distribution.

**Posterior Distribution.** Based on this formulation, the posterior distribution of the ATEDS estimator is a Gaussian Process. The following proposition summarizes its posterior mean and covariance.

**Proposition 4** (ATEDS Estimator Posterior). *Given the source training set $\mathcal{D}_T = \{a_T, S_T, y_T\}$ and a set of samples from the target population $\tilde{\mathcal{D}} = \{\tilde{S}\}$, let $n_{\tilde{s}} = |\tilde{\mathcal{D}}|$. If $\text{f}(a, s)$ is the posterior GP learned from $\mathcal{D}_T$, then the ATEDS estimator $\hat{\tau}_{\text{ATEDS}}(a)$ follows a GP, $\hat{\tau}_{\text{ATEDS}} \sim \mathcal{GP}(\nu(a), q(a, a'))$. Its posterior mean and covariance are derived using an effective feature map $\phi_{\bar{a}} := \phi(a) \otimes \hat{\tilde{\mu}}_{\mathbf{s}}$.*

**Detailed Derivation.** The following equations show the step-by-step expansion of the GP posterior formulas.

$$\nu(a) = \langle \phi_a \otimes \hat{\tilde{\mu}}_{\mathbf{s}}, m_{\text{f}} \rangle_{\mathcal{H}_{\mathcal{AS}}}$$

$$= \underbrace{\left( k_{aa_T} \odot \frac{1}{n_{\tilde{s}}} \mathbf{1}_{n_{\tilde{s}}}^T K_{\tilde{S}S_T} \right)}_{k_{\bar{a}X_T}} \underbrace{\left( (K_{a_T a_T} \odot K_{S_T S_T}) + \sigma^2 I \right)^{-1}}_{(K_{X_T X_T} + \sigma^2 I)^{-1}} y_T \tag{39}$$

$$= k_{\bar{a}X_T} (K_{X_T X_T} + \sigma^2 I)^{-1} y_T,$$

$$q(a, a') = k_{\bar{a}\bar{a}'} - k_{\bar{a}X_T} (K_{X_T X_T} + \sigma^2 I)^{-1} k_{X_T \bar{a}'}$$

$$= \underbrace{k_{aa'} \left( \frac{1}{n_{\tilde{s}}^2} \mathbf{1}_{n_{\tilde{s}}}^T K_{\tilde{S}\tilde{S}} \mathbf{1}_{n_{\tilde{s}}} \right)}_{k_{\bar{a}\bar{a}'}}$$

$$- \underbrace{\left( k_{aa_T} \odot \frac{1}{n_{\tilde{s}}} \mathbf{1}_{n_{\tilde{s}}}^T K_{\tilde{S}S_T} \right)}_{k_{\bar{a}X_T}} \left( (K_{a_T a_T} \odot K_{S_T S_T}) + \sigma^2 I \right)^{-1} \underbrace{\left( k_{a_T a'} \odot \frac{1}{n_{\tilde{s}}} K_{S_T \tilde{S}} \mathbf{1}_{n_{\tilde{s}}} \right)}_{k_{X_T \bar{a}'}}. \tag{40}$$

Here, $K_{X_T X_T} = K_{a_T a_T} \odot K_{S_T S_T}$, $\mathbf{1}_{n_{\tilde{s}}}$ is a column vector of $n_{\tilde{s}}$ ones, and $\sigma^2 > 0$ is the GP observation noise variance.

## D.5 CONNECTION BETWEEN IG AND TVR

We have instantiated our key principle of uncertainty reduction using two prominent strategies derived from information theory and optimal design: IG and TVR. While these two approaches are often presented as distinct heuristics, they share a deep connection, both aiming to shrink the posterior uncertainty of the CQ estimator. They differ, however, in how they quantify this uncertainty, focusing on different geometric properties of the posterior covariance matrix. The following remark clarifies this relationship by analyzing how each strategy operates on the eigenvalues of this matrix.

**Remark 3.** *IG and TVR can be understood through the eigenvalues, $\{\lambda_i\}$, of the posterior covariance matrix $Q_{post} = \mathbb{Var}[\hat{\tau}(a_I, Z_I)|\mathcal{D}_T, y_{X_B}]$. The **TVR** strategy aims to minimize the trace of the posterior covariance. Since the trace is the sum of the diagonal elements (the individual variances), and also the sum of the eigenvalues, the objective is:*

$$X_B^* = \underset{\substack{X_B \subseteq X_P \\ |X_B| = n_b}}{\arg\min} \text{Tr}(Q_{post}) = \underset{\substack{X_B \subseteq X_P \\ |X_B| = n_b}}{\arg\min} \sum_i \lambda_i. \tag{41}$$

*This corresponds to minimizing the arithmetic mean of the eigenvalues, effectively shrinking the average uncertainty across all dimensions. This is also known as A-optimality. The **IG** strategy is equivalent, up to terms independent of $\boldsymbol{X}_B$, to minimizing the determinant of the posterior covariance. Since the determinant is the product of the eigenvalues, the objective is:*

$$\boldsymbol{X}_B^* = \underset{\substack{\boldsymbol{X}_B \subseteq \boldsymbol{X}_P \\ |\boldsymbol{X}_B| = n_b}}{\arg\min} \det(\boldsymbol{Q}_{post}) = \underset{\substack{\boldsymbol{X}_B \subseteq \boldsymbol{X}_P \\ |\boldsymbol{X}_B| = n_b}}{\arg\min} \prod_i \lambda_i. \tag{42}$$

*This is equivalent to minimizing $\log(\det(\boldsymbol{Q}_{post})) = \sum_i \log(\lambda_i)$, which corresponds to minimizing the volume of the uncertainty ellipsoid, or the geometric mean of the eigenvalues. This is also known as D-optimality.*

## E  CONVERGENCE ANALYSIS

In this section, we analyze the uncertainty decay of the CQ estimator under our proposed method. Unlike inductive active learning, which reduces uncertainty in predictions over the empirical distribution represented by the pool dataset $p_{\text{pool}}(\boldsymbol{x})$, our problem is inherently transductive: we aim to evaluate the regression function over a distribution that typically differs from $p_{\text{pool}}(\boldsymbol{x})$, as it is specific to the target CQ. Hence, our approach can be viewed as a form of transductive active learning (TAL). The key distinction is in the objective: TAL seeks to optimize a function f over a target dataset, focusing on pointwise performance. In contrast, our method estimates an average over a distribution, shifting the emphasis from individual points to distributional estimation. Nevertheless, our theoretical analysis leverages the TAL framework introduced by Hübotter et al. (2024).

### E.1  JUSTIFICATION OF ASSUMPTION 1

To leverage the theoretical framework of Hübotter et al. (2024), we view the CQ target as the vector $\boldsymbol{v}_{\bar{\boldsymbol{X}}_I}$ and the TAL sample set as $\boldsymbol{X}_P$, the subset of the pool $\mathcal{D}_P$ for which outcomes can be acquired. When the numerical evaluation of a CQ induces a finite set of regressor inputs, denoted by $\boldsymbol{X}_{\text{CQ}}^{\text{eval}} \subseteq \mathcal{A} \times \mathcal{Z} \times \mathcal{S}$, the standard submodularity condition is naturally satisfied in cases where $\boldsymbol{X}_P \subseteq \boldsymbol{X}_{\text{CQ}}^{\text{eval}}$, as supported by Lemma C.9 in Hübotter et al. (2024). This situation arises for quantities such as ATE and DS under the positivity assumption, since the evaluation set can include counterfactual treatment assignments for individuals in the pool.

For CATE and ATT, the situation is more nuanced because the induced evaluation set and the sample set may overlap without a strict subset relationship. In these cases, the assumption may not strictly hold. To address this, we introduce a criterion called the Information Ratio later, which relaxes the assumption to a weak submodular condition. The resulting bound is derived under the weak submodularity assumption and covers the standard submodular case, affecting only a coefficient while maintaining the same convergence rate.

### E.2  PRELIMINARY DEFINITIONS AND CONCEPTS

In the following, we focus on the analysis of CATE, which is the most complex case. We use the same notations as in the main paper, where f is defined as a function $\mathcal{A} \times \mathcal{Z} \times \mathcal{S} \to \mathbb{R}$. Before proceeding with the theoretical analysis, we clarify the notations and define key preliminary concepts used in this section. Instead of using $\hat{\tau}_{\text{CATE}}(a, \boldsymbol{z})$ as defined in the main paper, we simplify the notation by introducing $\bar{\boldsymbol{x}} = (a, \boldsymbol{z})$ and $\bar{\boldsymbol{x}} \in \bar{\mathcal{X}} = \mathcal{A} \times \mathcal{Z}$, with $\bar{\boldsymbol{x}}_I = (\boldsymbol{a}_I, \boldsymbol{z}_I)$ and $\bar{\boldsymbol{X}}_I = (\boldsymbol{a}_I, \boldsymbol{Z}_I)$. Note that $\boldsymbol{x} = (a, \boldsymbol{z}, \boldsymbol{s})$ and should not be confused with $\bar{\boldsymbol{x}}$. Additionally, we define an operator $\upsilon[\text{f}] := \int_{\mathcal{S}} \text{f}(a, \boldsymbol{z}, \boldsymbol{s}) \mathbb{P}_{\text{s}|\boldsymbol{z}}(d\boldsymbol{s}|\boldsymbol{z})$. For brevity, we denote $\upsilon_{\bar{\boldsymbol{x}}} = \upsilon[\text{f}](a, \boldsymbol{z})$. Since f is a GP, $\upsilon_{\bar{\boldsymbol{x}}}$ follows a Gaussian distribution as it is a linear functional of f. Our proposed method can be seen as acquiring outcomes to reduce the posterior uncertainty of the vector $\boldsymbol{v}_{\bar{\boldsymbol{X}}_I}$, where the posterior mean and variance of any element $\upsilon_{\bar{\boldsymbol{x}}_I}$ are given in Prop. 1.

Throughout this section, all GP quantities are understood conditionally on the initial dataset $\mathcal{D}_0$. For notational simplicity, we write $\boldsymbol{K}_{\boldsymbol{X}_P \boldsymbol{X}_P}$ for the posterior covariance matrix $\text{Cov}(\boldsymbol{f}_{\boldsymbol{X}_P} \mid \mathcal{D}_0)$. Let $\mathcal{D}_n = \mathcal{D}_0 \cup \{(\boldsymbol{x}_i, y_i)\}_{i=1}^n$ denote the labeled dataset after $n$ acquisitions from $\mathcal{D}_P$. We define $\sigma_I^2 \overset{\text{def}}{=} \max_{\bar{\boldsymbol{x}}_I \in \bar{\boldsymbol{X}}_I} \mathbb{V}\text{ar}[\upsilon_{\bar{\boldsymbol{x}}_I} \mid \mathcal{D}_0]$, and $\sigma_P^2 \overset{\text{def}}{=} \max_{\boldsymbol{x} \in \boldsymbol{X}_P} \mathbb{V}\text{ar}[\text{f}_{\boldsymbol{x}} \mid \mathcal{D}_0]$, $\tilde{\sigma}_P^2 \overset{\text{def}}{=} \sigma_P^2 + \sigma^2$, where $\sigma^2$ is the

noise variance. The irreducible uncertainty is explicitly defined as $\eta^2_{\mathcal{D}_P}(\bar{x}) \overset{\text{def}}{=} \mathbb{V}\text{ar}[v_{\bar{x}} \mid \mathcal{D}_0, f_{X_P}]$. Furthermore, we define the information capacities $\gamma^v_n \overset{\text{def}}{=} \max_{X \subseteq X_P, |X| \leq n} I(v_{\bar{X}_I}; y_X \mid \mathcal{D}_0)$ and $\gamma^P_n \overset{\text{def}}{=} \max_{X \subseteq X_P, |X| \leq n} I(f_{X_P}; y_X \mid \mathcal{D}_0)$, and let their dominating capacity be $\bar{\gamma}_n \overset{\text{def}}{=} \max\{\gamma^v_n, \gamma^P_n\}$. To simplify notation in the main paper, we define the main text quantity $\gamma_n$ to equal this dominating capacity $\bar{\gamma}_n$.

**Definition 3** (Marginal Gain and Maximal Marginal Gain). *The marginal gain of $x \in X_P$ given $X \subseteq X_P$ is defined as*

$$\Delta_{\bar{X}_I}(x \mid X) \overset{\text{def}}{=} U_{\bar{X}_I}(X \cup \{x\}) - U_{\bar{X}_I}(X), \tag{43}$$

*which corresponds to the IG or TVR objective. The maximal marginal gain after $i$ greedy selections is defined as*

$$\Gamma_i \overset{\text{def}}{=} \max_{x \in X_P} \Delta_{\bar{X}_I}(x \mid x_{1:i}). \tag{44}$$

Marginal gain quantifies the information contribution of a single point $x$, while maximal marginal gain measures the peak possible contribution in a greedy sequence. Next, we define the information ratio.

**Definition 4** (Information Ratio). *The information ratio of $X \subseteq X_P$ given $D \subseteq X_P$, with $|X|, |D| < \infty$, is defined as*

$$\bar{\kappa}(X \mid D) \overset{\text{def}}{=} \frac{\sum_{x \in X} \Delta_{\bar{X}_I}(x \mid D)}{\Delta_{\bar{X}_I}(X \mid D)} \in [0, \infty). \tag{45}$$

**Remark 4.** *The insight behind these definitions is to leverage the classic performance guarantee for greedy optimization of submodular functions. For a set of acquired points $x_{1:n_A}$, this is given by $U(x_{1:n_A}) \geq (1 - e^{-c_U}) \max_{X \subseteq X_P, |X| \leq n_A} U(X)$, where $c_U$ is a parameter. This implies the utility obtained by our strategy is bounded by a fraction of the best achievable utility. Assuming Ass. 1 holds, $c_U = 1$ for adaptive acquisition, while $c_U$ depends on the submodularity ratio for batch acquisition.*

With these definitions in place, we proceed to a concept key to our convergence analysis: the Approximate Markov Boundary.

### E.3 APPROXIMATE MARKOV BOUNDARY

**Intuition 1.** *In inductive AL uncertainty decay analysis, the maximum information gain is typically used to bound the information gain achieved by different methods. Acquiring all observation outcomes in $\mathcal{D}_P$ would naturally lead to convergence to this maximum information gain. However, in our setting, even if all outcomes in $\mathcal{D}_P$ were acquired, the uncertainty over the target distribution of our estimator would not be fully eliminated. To account for this, we follow a similar approach to Hübotter et al. (2024) to define the irreducible uncertainty $\eta^2_{\mathcal{D}_P}(\bar{x})$ as the lowest uncertainty attainable when the latent regressor values over $X_P$ are known. However, a key challenge lies in distinguishing between the irreducible uncertainty and the reducible uncertainty, as analyzing the uncertainty decay rate requires isolating the reducible component. An idea inspired by the Markov Blanket can be used to derive a rough formulation: $v_{\bar{x}} \perp\!\!\!\perp f_{X_P} \mid f_{\text{MB}(B)}$ where $B \subseteq X_P$. This suggests that the uncertainty at $\bar{x}$ is conditionally independent of $f_{X_P}$ given the function values at the Markov Blanket $\text{MB}(B)$, providing a potential way to separate the irreducible and reducible uncertainties.*

Following the above Intuition, we introduce a new definition named *Approximate Markov Boundary* of $\bar{x}$ in $\mathcal{D}_P$, which is defined as:

**Definition 5** (Approximate Markov Boundary (AMB)). *For any $\epsilon > 0$, $n \geq 0$, and $\bar{x} \in \bar{X}_I$, we define $B_{n,\epsilon}(\bar{x})$ as the smallest (multi-)set of $X_P$ satisfying*

$$\mathbb{V}\text{ar}\left[v_{\bar{x}} \mid \mathcal{D}_n, y_{B_{n,\epsilon}(\bar{x})}\right] \leq \eta^2_{\mathcal{D}_P}(\bar{x}) + \epsilon. \tag{46}$$

*We refer to $B_{n,\epsilon}(\bar{x})$ as the $\epsilon$-approximate Markov boundary of $\bar{x}$ in $X_P$.*

The key idea behind constructing an AMB is that it acts as a bridge, linking the posterior uncertainty of $v_{\bar{x}}$ to two components: the fundamental irreducible uncertainty and a controllable approximation term $\epsilon$. The rough idea, then, is to analyze the uncertainty decay when observing the outcomes of

points in $\boldsymbol{B}_{n,\epsilon}(\bar{\boldsymbol{x}})$. Since Eq. 46 provides an upper bound on the remaining uncertainty, it can be leveraged to constrain the decay rate.

Before utilizing the AMB, we first establish its existence through a formal proof.

**Lemma 3.** *Assume that* $\boldsymbol{K}_{\boldsymbol{X}_P\boldsymbol{X}_P}$ *is positive definite and that* $\gamma_r^P/r \to 0$ *as* $r \to \infty$. *Let* $\epsilon > 0$ *and define* $r$ *as the smallest integer satisfying*

$$\frac{\gamma_r^P}{r} \le \frac{\epsilon\lambda_{\min}(\boldsymbol{K}_{\boldsymbol{X}_P\boldsymbol{X}_P})}{2n_P\sigma_I^2\tilde{\sigma}_P^2}. \tag{47}$$

*For any* $n \ge 0$ *and* $\bar{\boldsymbol{x}} \in \bar{\boldsymbol{X}}_I$, *there exists an* $\epsilon$-*approximate Markov boundary* $\boldsymbol{B}_{n,\epsilon}(\bar{\boldsymbol{x}})$ *for* $\bar{\boldsymbol{x}}$ *within* $\boldsymbol{X}_P$, *with a size bounded by* $r$.

*Proof.* By Lemma C.19 of Hübotter et al. (2024), applied to the GP posterior after $\mathcal{D}_0$, for the integer $r$ satisfying Eq. 47, there exists a set $\boldsymbol{B} \subseteq \boldsymbol{X}_P$ with $|\boldsymbol{B}| \le r$ such that, for all $\boldsymbol{x}' \in \boldsymbol{X}_P$, $\mathbb{V}\mathrm{ar}[\mathrm{f}_{\boldsymbol{x}'} \mid \mathcal{D}_0, \boldsymbol{y}_{\boldsymbol{B}}] \le \frac{\epsilon\lambda_{\min}(\boldsymbol{K}_{\boldsymbol{X}_P\boldsymbol{X}_P})}{n_P\sigma_I^2}$. Applying Lem. 4 yields $\mathbb{V}\mathrm{ar}[v_{\bar{\boldsymbol{x}}} \mid \mathcal{D}_0, \boldsymbol{y}_{\boldsymbol{B}}] \le \eta^2_{\mathcal{D}_P}(\bar{\boldsymbol{x}}) + \epsilon$. Since $\mathcal{D}_n$ contains $\mathcal{D}_0$, monotonicity of conditional variance gives $\mathbb{V}\mathrm{ar}[v_{\bar{\boldsymbol{x}}} \mid \mathcal{D}_n, \boldsymbol{y}_{\boldsymbol{B}}] \le \mathbb{V}\mathrm{ar}[v_{\bar{\boldsymbol{x}}} \mid \mathcal{D}_0, \boldsymbol{y}_{\boldsymbol{B}}] \le \eta^2_{\mathcal{D}_P}(\bar{\boldsymbol{x}}) + \epsilon$. Hence $\boldsymbol{B}$ satisfies Def. 5, and the smallest such set has size at most $r$. $\qquad\square$

Then, we move to examine a key property of any set satisfying a similar condition.

**Lemma 4.** *Let* $\epsilon > 0$ *and* $\boldsymbol{B} \subseteq \boldsymbol{X}_P$ *such that, for all* $\boldsymbol{x}' \in \boldsymbol{X}_P$,

$$\mathbb{V}\mathrm{ar}[\mathrm{f}_{\boldsymbol{x}'} \mid \mathcal{D}_0, \boldsymbol{y}_{\boldsymbol{B}}] \le \frac{\epsilon\lambda_{\min}(\boldsymbol{K}_{\boldsymbol{X}_P\boldsymbol{X}_P})}{n_P\sigma_I^2}. \tag{48}$$

*Then for any* $\bar{\boldsymbol{x}} \in \bar{\boldsymbol{X}}_I$, *we have*

$$\mathbb{V}\mathrm{ar}[v_{\bar{\boldsymbol{x}}} \mid \mathcal{D}_0, \boldsymbol{y}_{\boldsymbol{B}}] \le \mathbb{V}\mathrm{ar}[v_{\bar{\boldsymbol{x}}} \mid \mathcal{D}_0, \boldsymbol{f}_{\boldsymbol{X}_P}] + \epsilon. \tag{49}$$

*Proof.* Let the right-hand side of Eq. 48 be denoted by $\epsilon'$. From the given condition, we can derive that

$$\begin{aligned}
\mathbb{V}\mathrm{ar}[v_{\bar{\boldsymbol{x}}} \mid \mathcal{D}_0, \boldsymbol{y}_{\boldsymbol{B}}] &= \mathbb{E}_{\boldsymbol{f}_{\boldsymbol{X}_P}\mid\mathcal{D}_0,\boldsymbol{y}_{\boldsymbol{B}}}[\mathbb{V}\mathrm{ar}_{v_{\bar{\boldsymbol{x}}}}[v_{\bar{\boldsymbol{x}}} \mid \mathcal{D}_0, \boldsymbol{f}_{\boldsymbol{X}_P}, \boldsymbol{y}_{\boldsymbol{B}}]] + \mathbb{V}\mathrm{ar}_{\boldsymbol{f}_{\boldsymbol{X}_P}\mid\mathcal{D}_0,\boldsymbol{y}_{\boldsymbol{B}}}[\mathbb{E}_{v_{\bar{\boldsymbol{x}}}}[v_{\bar{\boldsymbol{x}}} \mid \mathcal{D}_0, \boldsymbol{f}_{\boldsymbol{X}_P}, \boldsymbol{y}_{\boldsymbol{B}}]] \\
&= \mathbb{V}\mathrm{ar}_{v_{\bar{\boldsymbol{x}}}}[v_{\bar{\boldsymbol{x}}} \mid \mathcal{D}_0, \boldsymbol{f}_{\boldsymbol{X}_P}, \boldsymbol{y}_{\boldsymbol{B}}] + \mathbb{V}\mathrm{ar}_{\boldsymbol{f}_{\boldsymbol{X}_P}\mid\mathcal{D}_0,\boldsymbol{y}_{\boldsymbol{B}}}[\mathbb{E}_{v_{\bar{\boldsymbol{x}}}}[v_{\bar{\boldsymbol{x}}} \mid \mathcal{D}_0, \boldsymbol{f}_{\boldsymbol{X}_P}, \boldsymbol{y}_{\boldsymbol{B}}]] \\
&= \mathbb{V}\mathrm{ar}_{v_{\bar{\boldsymbol{x}}}}[v_{\bar{\boldsymbol{x}}} \mid \mathcal{D}_0, \boldsymbol{f}_{\boldsymbol{X}_P}] + \mathbb{V}\mathrm{ar}_{\boldsymbol{f}_{\boldsymbol{X}_P}\mid\mathcal{D}_0,\boldsymbol{y}_{\boldsymbol{B}}}[\mathbb{E}_{v_{\bar{\boldsymbol{x}}}}[v_{\bar{\boldsymbol{x}}} \mid \mathcal{D}_0, \boldsymbol{f}_{\boldsymbol{X}_P}]].
\end{aligned} \tag{50}$$

The first step follows from the law of total variance. The second step utilizes the fact that the conditional variance of a GP depends only on observation locations, not their values. The third step results from the conditional independence $v_{\bar{\boldsymbol{x}}} \perp\!\!\!\perp \boldsymbol{y}_{\boldsymbol{B}} \mid \boldsymbol{f}_{\boldsymbol{X}_P}, \mathcal{D}_0$, as $\boldsymbol{B} \subseteq \boldsymbol{X}_P$. The remaining task is to bound the reducible uncertainty. We define a function $h_{\bar{\boldsymbol{x}}} : \mathbb{R}^{n_P} \to \mathbb{R}$, $\boldsymbol{f}_{\boldsymbol{X}_P} \mapsto \mathbb{E}[v_{\bar{\boldsymbol{x}}} \mid \mathcal{D}_0, \boldsymbol{f}_{\boldsymbol{X}_P}]$. Using the formula for the GP posterior mean, we have $h_{\bar{\boldsymbol{x}}}(\boldsymbol{f}_{\boldsymbol{X}_P}) = \mathbb{E}[v_{\bar{\boldsymbol{x}}} \mid \mathcal{D}_0] + \boldsymbol{l}^\top(\boldsymbol{f}_{\boldsymbol{X}_P} - \mathbb{E}[\boldsymbol{f}_{\boldsymbol{X}_P} \mid \mathcal{D}_0])$, where $\boldsymbol{l} = (\boldsymbol{K}_{\boldsymbol{X}_P\boldsymbol{X}_P})^{-1}\boldsymbol{\Upsilon}$ and $\boldsymbol{\Upsilon} = \mathrm{Cov}(\boldsymbol{f}_{\boldsymbol{X}_P}, v_{\bar{\boldsymbol{x}}} \mid \mathcal{D}_0)$. In the CME case, $\boldsymbol{\Upsilon}$ is obtained from the corresponding posterior effective kernel between $\boldsymbol{f}_{\boldsymbol{X}_P}$ and $v_{\bar{\boldsymbol{x}}}$; its explicit form is not needed for the argument. Since $h_{\bar{\boldsymbol{x}}}$ is a linear function in $\boldsymbol{f}_{\boldsymbol{X}_P}$, we have for the reducible uncertainty that

$$\mathbb{V}\mathrm{ar}[h_{\bar{\boldsymbol{x}}}(\boldsymbol{f}_{\boldsymbol{X}_P}) \mid \mathcal{D}_0, \boldsymbol{y}_{\boldsymbol{B}}] = \boldsymbol{l}^\top \mathbb{V}\mathrm{ar}[\boldsymbol{f}_{\boldsymbol{X}_P} \mid \mathcal{D}_0, \boldsymbol{y}_{\boldsymbol{B}}]\boldsymbol{l} \le \frac{\epsilon' n_P \sigma_I^2}{\lambda_{\min}(\boldsymbol{K}_{\boldsymbol{X}_P\boldsymbol{X}_P})}. \tag{51}$$

This inequality can be derived in a similar manner to Lemma C.20 in Hübotter et al. (2024), using analogous steps. By the definition of $\epsilon'$, this final term is equal to $\epsilon$. $\qquad\square$

### E.4 Proof of Theorem 2

To prove Thm. 2, we follow the analytical framework of Hübotter et al. (2024). The proof proceeds in three main steps: (1) we leverage the property of the AMB to relate the current variance to the information gain of the AMB set; (2) we bound this information gain using properties of submodular functions; and (3) we select a decaying approximation error $\epsilon$ to derive the final convergence rate.

First, from Lemma C.17 in Hübotter et al. (2024) and our AMB definition (Def. 5), we can bound the current variance of the estimator for any $\bar{x} \in \bar{X}_I$ as:

$$\mathbb{Var}[v_{\bar{x}} \mid \mathcal{D}_n] \le 2\sigma_I^2 \mathrm{I}(v_{\bar{x}}; y_{B_{n,\epsilon}(\bar{x})} \mid \mathcal{D}_n) + \eta_{\mathcal{D}_P}^2(\bar{x}) + \epsilon. \tag{52}$$

Next, we bound the mutual information term. By the submodularity of information gain (Assumption 1), the total gain from a set is bounded by the sum of marginal gains. This leads to a bound involving the maximal marginal gain $\Gamma$ (similar to the logic in the proof of Thm 3.3 in the cited work):

$$\mathrm{I}(v_{\bar{x}}; y_{B_{n,\epsilon}(\bar{x})} \mid \mathcal{D}_n) \le C_1 |B_{n,\epsilon}(\bar{x})| \cdot \Gamma_n, \tag{53}$$

where $C_1$ is a constant related to the submodularity/information ratio $\bar{\kappa}$. Let $b_\epsilon = |B_{n,\epsilon}(\bar{x})|$ be the size of the AMB. This gives:

$$\mathbb{Var}[v_{\bar{x}} \mid \mathcal{D}_n] \le 2\sigma_I^2 C_1 b_\epsilon \Gamma_n + \eta_{\mathcal{D}_P}^2(\bar{x}) + \epsilon. \tag{54}$$

Now, we select a specific value for $\epsilon$ that decays with $n$. Let $\epsilon = c\frac{\gamma_{\lceil\sqrt{n}\rceil}^P}{\lceil\sqrt{n}\rceil}$, with $c = 2n_P\sigma_I^2\tilde{\sigma}_P^2/\lambda_{\min}(K_{X_P X_P})$. From Eq. 47, this choice of $\epsilon$ ensures that an AMB exists with a size $b_\epsilon$ bounded by $r \approx \lceil\sqrt{n}\rceil$.

Substituting $b_\epsilon \le \lceil\sqrt{n}\rceil$ and the expression for $\epsilon$ into Eq. 54, we get:

$$\mathbb{Var}[v_{\bar{x}} \mid \mathcal{D}_n] \le 2\sigma_I^2 C_1 \lceil\sqrt{n}\rceil \Gamma_n + \eta_{\mathcal{D}_P}^2(\bar{x}) + c\frac{\gamma_{\lceil\sqrt{n}\rceil}^P}{\lceil\sqrt{n}\rceil}. \tag{55}$$

The term $\Gamma_n$ itself decays. Building on Theorem C.13 from Hübotter et al. (2024), the maximal marginal gain is bounded by a term that decays with $n$. A simplified consequence for our analysis is that we can find a constant $C_2$ such that:

$$\Gamma_n \le C_2 \frac{\bar{\gamma}_n}{n}. \tag{56}$$

Substituting this bound for $\Gamma_n$ into Eq. 55 resolves the reducible term:

$$2\sigma_I^2 C_1 \lceil\sqrt{n}\rceil \Gamma_n \le 2\sigma_I^2 C_1 \lceil\sqrt{n}\rceil \left(C_2 \frac{\bar{\gamma}_n}{n}\right) \le (4\sigma_I^2 C_1 C_2)\frac{\bar{\gamma}_n}{\sqrt{n}}. \tag{57}$$

Combining all the pieces, the total variance after $n_B$ active acquisitions is bounded by:

$$\begin{aligned}
\mathbb{Var}[v_{\bar{x}} \mid \mathcal{D}_{n_B}] &\le (4\sigma_I^2 C_1 C_2)\frac{\bar{\gamma}_{n_B}}{\sqrt{n_B}} + \eta_{\mathcal{D}_P}^2(\bar{x}) + c\frac{\bar{\gamma}_{\lceil\sqrt{n_B}\rceil}}{\sqrt{n_B}} \\
&\le \eta_{\mathcal{D}_P}^2(\bar{x}) + C\frac{\bar{\gamma}_{n_B}}{\sqrt{n_B}},
\end{aligned} \tag{58}$$

where the final step combines all constants into a single constant $C$, uses $\gamma_{\lceil\sqrt{n_B}\rceil}^P \le \bar{\gamma}_{\lceil\sqrt{n_B}\rceil} \le \bar{\gamma}_{n_B}$ by monotonicity, and uses $\lceil\sqrt{n_B}\rceil \ge \sqrt{n_B}$. In the main paper, we use $n_B$ for the total number of acquired samples, which corresponds to $n$ here. The results for TVR can be derived similarly. It is worth noting that while our convergence analysis builds upon the framework for TAL, our contribution lies in successfully generalizing this framework from the simpler task of pointwise prediction to the more complex task of distributional integral estimation. Our analysis reveals that, despite the added complexity of our problem, the optimal convergence rate retains a similar form, demonstrating the robustness and power of principled active learning for this new and important class of problems.

# F    EXPERIMENTS DETAILS

In this supplementary section, we present a detailed account of the experimental results discussed in the main text. This includes a comprehensive description of the data generation process, as well as the implementation details for both the baseline methods and our proposed methods. Specifically, we provide information on the visualization dataset, the simulation dataset, and the the semi-synthetic dataset: IHDP (Hill, 2011) and Lalonde (LaLonde, 1986).

### F.1 DATASETS

**General Setup**    For each experimental configuration, we report the **mean and standard error over 20 independent trials**, each initiated with a unique random seed.

- **Datasets and Evaluation:** All simulation datasets consist of 500 training, 200 validation, and 500 testing samples. We evaluate all methods on both the in-distribution training set and the testing set. The main paper presents the results on the testing dataset, while the Appendix here contains comprehensive results for both.

- **Variable Definitions:** Across all experiments, the treatment a and outcome y are one-dimensional. We denote a single observation as a quadruplet $(z, s, a, y)$, representing the conditioning covariate, adjustment covariates, treatment, and outcome, respectively. The dimensionality of the adjustment covariates **s** varies by setting.

- **Intervention Strategies:** We evaluate two types of interventions for all causal quantities:
    - **Hard Intervention:** The target treatment is a single value, $a$, randomly selected for each trial. In settings with continuous treatments, we discretize the treatment space for evaluation on the training set, as exact continuous values may not be present in the data.
    - **Soft Intervention:** The target is a treatment distribution, specified as $p^*(a) \sim$ Uniform$[\min(\boldsymbol{a}_T), \max(\boldsymbol{a}_T)]$. This requires the estimator to average over a range of treatments.

- **CATE-Specific Setups:** For CATE estimation, which depends on the conditioning variable z, we test two distinct scenarios:
    - **Fixed $z$:** A single value of $z$ is randomly drawn from its training set range and held constant across all 20 trials for a given configuration.
    - **Random $z$:** A new value of $z$ is randomly and independently drawn for each of the 20 trials.

#### F.1.1 VISUALIZATION

In the visualization case, where the goal is to examine the match between the acquired data points and the target distribution, we set the dimensionality of the adjustment to 2. One is controlled by the conditioning variable, and the other is a noisy variable that does not influence the treatment decision or the outcome variable. The process for generating a single observation is as follows:

1. Set the conditioning variable:

$$z \sim \text{Uniform}(-2, 2). \tag{59}$$

2. Define the adjustment variables $s_1$ and $s_2$ as:

$$\begin{cases} s_1 = \text{Skew}(z), \\ s_2 = \exp(2\epsilon_1) + \epsilon_2, \end{cases} \tag{60}$$

where $\epsilon_1, \epsilon_2 \sim \mathcal{N}(0, 1)$. For the skew function $\text{Skew}(\cdot)$, we define:

$$y_s \sim \text{Skew}(\xi(x), \omega(x), \alpha(x)), \quad \text{where} \quad x = 2.5 \cdot z, \tag{61}$$

with:

$$\xi(x) = 0.1 \cdot x, \quad \omega(x) = 0.1 \cdot |x| + 0.05, \quad \alpha(x) = -8 + 8 \cdot \left( \frac{1}{1 + \exp(-x)} \right). \tag{62}$$

Next, to generate the treatment variable, we focus on the continuous case (excluding the binary case). First, we concatenate the conditional variable with additional features to form the random vector **x**, and the weight vector $\beta$ is defined as:

$$\mathbf{x} = \begin{bmatrix} z & s_1 \end{bmatrix}, \quad \beta = \begin{bmatrix} \frac{1}{1^2} \\ \frac{1}{2^2} \end{bmatrix} = \begin{bmatrix} 1 \\ 0.25 \end{bmatrix}. \tag{63}$$

The intermediate variable $a_{\text{org}}$ is then generated as:

$$a_{\text{org}} = \Phi\left(3 \cdot (\mathbf{x} \cdot \beta)\right) + 1.5 \cdot \epsilon - 0.5, \quad \epsilon \sim \mathcal{N}(0, 1), \tag{64}$$

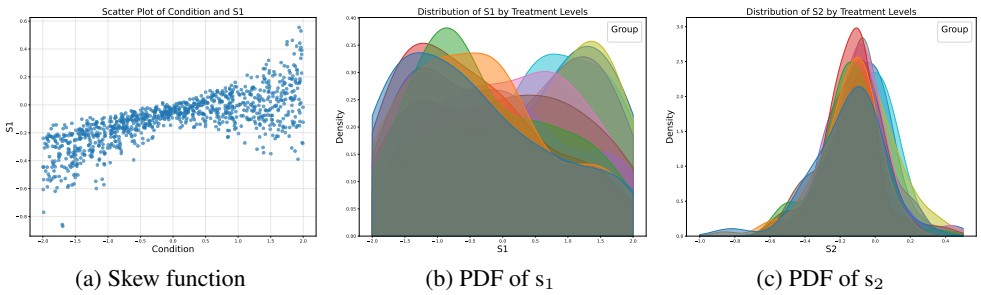

(a) Skew function          (b) PDF of $s_1$          (c) PDF of $s_2$

Figure 7: Visualization of the visualization dataset. (a) Scatter plot of the conditioning variable and the adjustment variable $s_1$; (b) the pdf of the first adjustment variable $s_1$ for different treatments; (c) the pdf of the second adjustment variable $s_2$ for different treatments.

Then, we generate the corresponding treatment variable by

$$a = \frac{1.0}{1.0 + \exp(-a_{\text{org}})}. \tag{65}$$

We observe that the values of the treatment variable lie within the range $(0, 1)$. If discretization is required, we use a step size of $0.1$ for the treatment variable. Finally, we generate the outcome variable:

$$y = a \cdot z \cdot s_1 + 2 \cdot z + s_1 + \epsilon_y, \quad \text{where} \quad \epsilon_y \sim \mathcal{N}(0, 0.16). \tag{66}$$

### F.1.2 SIMULATIONS

For the simulation, which we run mainly our most of different scenarios, including the cate, ate, att and distribution shift. We mainly follow the similar data generation way from (Abrevaya et al., 2015; Singh et al., 2024). For the experimental results in Sec. 5, we make $s \in \mathbb{R}^4$. A single observation is generated as follow. Draw unobserved noise as $\epsilon_i \overset{\text{i.i.d.}}{\sim} \mathcal{N}(0, 1)$, where $i = 1, 2, 3, 4, 5$. Then set

$$z \sim \text{Uniform}(-2, 2), \quad \begin{cases} s_1 = \cos(z) + z + \epsilon_1 \\ s_2 = -1 + \frac{1}{4}z^2 + \epsilon_2 \\ s_3 = \sin z^2 + \epsilon_3 \\ s_4 = \exp(2\epsilon_4) + \epsilon_5, \end{cases} \tag{67}$$

Next, to generate the treatment variable, in this case, we have three different settings, including the binary treatment, the continuous treatment and the discrete treatment. First, we concatenate the conditional variable with the first three adjustment variable to form the random vector $\mathbf{x}$, and the weight vector $\beta$ is defined as:

$$\mathbf{x} = [z, s_1, s_2, s_3], \quad \beta_i = \frac{1}{j^2}, \text{for } j = 1, \cdots, 4. \tag{68}$$

The intermediate variable $a_{\text{org}}$ is then generated as:

$$a_{\text{org}} = \Phi\left(3 \cdot (\mathbf{x} \cdot \beta)\right) + 1.5 \cdot \epsilon - 0.5, \quad \epsilon \sim \mathcal{N}(0, 1), \tag{69}$$

Then, if the treatment is binary, we use the following decision rule to get the treatment

$$a = \begin{cases} 1, & \text{if } a_{\text{org}} > 0; \\ 0, & \text{else.} \end{cases} \tag{70}$$

If we need the continuous treatment, we make the following decision rule to get the treatment

$$a = \frac{1.0}{1.0 + \exp(-a_{\text{org}})}. \tag{71}$$

We observe that the values of the treatment variable lie within the range $(0, 1)$. If discretization is required, we use a step size of $0.1$ for the treatment variable. Note that for the ATE and ATT cases, we

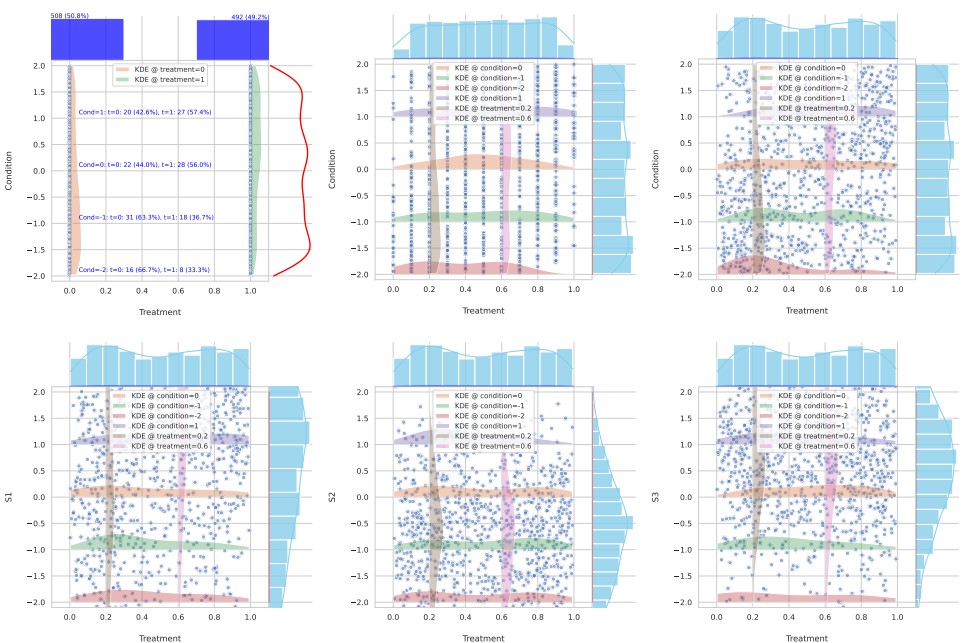

Figure 8: Scatter Plot with Conditional Distribution Overlay

retain only the first three adjustment variables and incorporate the conditioning variable z directly into the adjustment variable set s. For the ATE under distribution shift, we define the target distribution as

$$z \sim \text{Uniform}(-1, 1), \qquad \begin{cases} s_1 = \text{Uniform}(-1, 1) \\ s_2 = \text{Uniform}(-0.5, 0) \\ s_3 = \text{Uniform}(0, 0.5) \end{cases} \tag{72}$$

### F.1.3 SEMI-SYNTHETIC DATASETS

**IHDP.** The IHDP dataset, derived from a randomized controlled trial, is widely used in causal inference to benchmark methods for estimating treatment effects. It contains 747 samples, with 139 in the treatment group and 608 in the control group, and 25 covariates: age, sex, race, married, education, income, maternal age, birth weight, gestation, mother smoking, mother drinking, mother health, father age, father education, father income, father smoking, father drinking, father health, num children, urban, prenatal care, previous pregnancy, low birth weight, premature birth, and health intervention. Among these, 6 covariates are continuous: age, maternal age, birth weight, gestation, father age, and father income, while 19 covariates are binary. Treatment selection bias is introduced by removing a subset of treated individuals, and synthetic outcomes are generated to enable ground-truth calculations of ATE and CATE. The dataset's high-dimensional covariate space and realistic challenges, such as treatment heterogeneity, make it a valuable resource for evaluating causal inference algorithms.

**Binary case.** For the binary case, we do not need to generate the corresponding treatment and directly obtain them from the dataset. To generate the potential outcomes, we first extract the continuous covariates to form the random vector $\mathbf{x}$, and the weight vector $\beta$ is defined as:

$$\mathbf{x} = [\text{birth weight, birth head circumference, preterm birth, \textbf{birth order}, neonatal health, mother's age}],$$
$$\beta_i = \frac{1}{j}, \text{for } j = 1, \cdots, 6. \tag{73}$$

$$y = \begin{cases} 1.2 * (\mathbf{x} \cdot \beta) + 1 + \epsilon_0, & \epsilon_0 \sim \mathcal{N}(0, 0.16) \quad \text{if t} = 0; \\ 1.2 * (\mathbf{x} \cdot \beta) + \exp(\mathbf{x} + 0.5) \cdot \beta + 3 \cdot \text{b.w} + 1 + \epsilon_0, & \epsilon_1 \sim \mathcal{N}(0, 0.16) \quad \text{else} \end{cases} \tag{74}$$

where b.w represents the birth weight.

**Continuous and discrete case.** For the continuous and discrete case, we need to generate the corresponding treatment. To generate the treatment, we first extract the continuous covariates to form the random vector $\mathbf{x}$, and the weight vector $\beta$ is defined as: The intermediate variable $a_{\mathrm{org}}$ is then generated as:

$$a_{\mathrm{org}} = \Phi\left(3 \cdot (\mathbf{x} \cdot \beta)\right) + 1.5 \cdot \epsilon - 0.5, \quad \epsilon \sim \mathcal{N}(0, 1), \tag{75}$$

Then, if the treatment is binary, we use the following decision rule to get the treatment

$$a = \begin{cases} 1, & \text{if } a_{\mathrm{org}} > 0; \\ 0, & \text{else.} \end{cases} \tag{76}$$

If we need the continuous treatment, we make the following decision rule to get the treatment

$$a = \frac{1.0}{1.0 + \exp(-a_{\mathrm{org}})}. \tag{77}$$

We observe that the values of the treatment variable lie within the range $(0, 1)$. If discretization is required, we use a step size of $0.1$ for the treatment variable.

$$y = 1.2 * (\mathbf{x} \cdot \beta) + 1.2 \cdot t + b.w^2 + t * b.w + \epsilon_0, \quad \epsilon_1 \sim \mathcal{N}(0, 0.16). \tag{78}$$

For the ATE with distribution shift case, we generate the six continuous covariates as follows:

$$s_j = \mathrm{Uniform}(0, 0.5) \quad j = 1, \cdots, 6. \tag{79}$$

When evaluating the performance of the estimator in the DS case, we generate new data by combining them with the original data. For both the training and testing sets, we sample the required number of points from the new distribution and replace the corresponding covariate observations of the original data with these new values. These modified data points, now incorporating the new covariates, are treated as being drawn from the shifted distribution. From the visualization of the IHDP dataset, we observe a relatively strong correlation among the first three variables: birth weight, birth head circumference, and preterm birth. In all CATE setups, we use birth weight as the conditioning variable to estimate the causal effect across different subpopulations, stratified by birth weight.

**Lalonde.** This is a widely used benchmark in causal inference for evaluating methods that estimate treatment effects from observational data (LaLonde, 1986). It contains $2,675$ samples, with $151$ in the treatment group and $2,524$ in the control group, and $8$ covariates: age, education, black, hispanic, married, nodegree, re74 (earnings in 1974), and re75 (earnings in 1975). Among these, $4$ covariates are continuous: age, education, re74, and re75, while $4$ covariates are binary: black, hispanic, married, and nodegree. The dataset represents a real-world observational study where treatment assignment is not randomized, creating natural selection bias. The potential outcomes are pre-computed using established econometric methods, enabling ground-truth calculations of ATE and CATE. The dataset's realistic covariate structure and natural treatment heterogeneity make it a valuable resource for evaluating causal inference algorithms in observational settings.

**Binary case.** For the binary case, we do not need to generate the corresponding treatment and directly obtain them from the dataset. The potential outcomes are pre-computed and stored in the dataset. To generate the potential outcomes, we first extract the continuous covariates to form the random vector $\mathbf{x}$, and the weight vector $\beta$ is defined as:

$$\mathbf{x} = \left[\text{age}, \text{education}, \text{re74}, \text{re75}\right],$$
$$\beta_i = \frac{1}{j}, \text{for } j = 1, \cdots, 4. \tag{80}$$

$$y = \begin{cases} 1.2 * (\mathbf{x} \cdot \beta) + 1 + \epsilon_0, & \epsilon_0 \sim \mathcal{N}(0, 0.16) \quad \text{if } t = 0; \\ 1.2 * (\mathbf{x} \cdot \beta) + \exp(\mathbf{x} + 0.5) \cdot \beta + 3 \cdot \text{age} + 1 + \epsilon_0, & \epsilon_1 \sim \mathcal{N}(0, 0.16) \quad \text{else} \end{cases} \tag{81}$$

where age represents the age covariate.

For the ATE with distribution shift case, we generate the four continuous covariates as follows:

$$s_j = \mathrm{Uniform}(0, 0.5) \quad j = 1, \cdots, 4. \tag{82}$$

When evaluating the performance of the estimator in the DS case, we generate new data by combining them with the original data. For both the training and testing sets, we sample the required number of points from the new distribution and replace the corresponding covariate observations of the original data with these new values. These modified data points, now incorporating the new covariates, are treated as being drawn from the shifted distribution.

### F.2 Implementation Details

#### F.2.1 General Setting

**Validation dataset.** In the simulation dataset, we change the seed to sample a validation set from the same data generation process as the training dataset, ensuring that the observations in the validation set share the same distribution as the training data. In the real-world dataset IHDP, we directly randomly segment the training dataset into a training set and a validation set. This validation set is used to control early stopping during the training of the GP. However, this approach is not optimal because our goal is to minimize the uncertainty of the target CQ estimator. Ideally, the GP should perform better on the newly constructed distribution rather than directly on the observed data. This highlights a mismatch between the current validation setup and the optimal validation strategy. In principle, one could construct a validation set that mimics the new target distribution. However, given the limited size of the validation set, such an approach could lead to insufficient data for validation, posing a challenge for reliable model evaluation.

**Trial Setup.** For each trial, we initiated the process with a warm-start size, and the data for each experiment were randomly sampled from the pool dataset. This random sampling was done to ensure variability in the experiments and to evaluate the robustness of our methods under different initial conditions.

**Simulations.** In the binary treatment case, we used the Delta kernel, which is suitable for modeling binary outcomes. For future work, it is possible to explore the use of a multi-task kernel (Alaa & Van Der Schaar, 2017), which could help model the inner correlation between different groups. However, in our current implementation, we did not explore this possibility. In the CATE case, we employed an RBF kernel for the conditioning variable $\mathbf{z}$ and used another RBF kernel for other adjustment variables. The RBF kernel is chosen because of its flexibility in modeling smooth functions. For continuous treatment variables, we also used the RBF kernel. Additionally, we performed kernel ablation experiments by using alternative kernels, such as the Matern kernel and the Rational Quadratic (RQ) kernel, to explore the impact of different kernel choices on model performance. However, the default setup uses the RBF kernel for all conditioning variables, and kernel bandwidths for the adjustment variables are treated as hyperparameters to be optimized during training.

**Visualization.** In the visualization experiment, the warm-start size was set to 20, and the acquisition step was set to 5, meaning that data acquisition occurred in steps of 5 data points. This setup allowed us to monitor the effect of incremental data acquisition on model performance. The total number of training data points for this experiment was set to 200, with the CATE model being the sole model used in this case. All other experimental settings for the visualization were identical to those used in the simulation setup.

**IHDP.** In the IHDP experiment, we worked with a fixed number of observations. The warm-start size was set to 20, and the acquisition step was set to 10, meaning that data acquisition proceeded in increments of 10 data points. This setup was chosen to evaluate the model's performance with a real-world dataset under controlled acquisition conditions.

#### F.2.2 Hyperparameter Settings

In the active learning framework, the overall process is divided into multiple iterative rounds. Each round consists of two sequential stages: `Training` and `Acquisition`. During the training stage, we update the model parameters using the currently labeled dataset. In the acquisition stage, the trained model, together with the proposed acquisition criterion, is used to evaluate the utility of unlabeled points in the pool dataset.

**GP.** We perform GP regression without neural feature extractors in all experiments, including both synthetic simulations and the semi-synthetic IHDP dataset. Instead, we directly apply kernel-based models implemented via `GPyTorch` (v1.12)[3]. All models are trained using the Adam optimizer with a fixed learning rate of 0.05 for 500 epochs. Full-batch training is used throughout. No early stopping is applied in the first 200 epochs, and validation performance is monitored every 10 epochs. Hyperparameters $\sigma^2$ and any kernel parameters (e.g., bandwidths of the RBF kernel) are optimized

---

[3]https://github.com/cornellius-gp/gpytorch/releases/tag/v1.12

by maximizing the exact marginal log-likelihood, defined as:

$$\log p(\boldsymbol{y}_T \mid \boldsymbol{X}_T) = -\frac{1}{2}\boldsymbol{y}_T^\top(\boldsymbol{K}_{\boldsymbol{X}_T\boldsymbol{X}_T} + \sigma^2\boldsymbol{I})^{-1}\boldsymbol{y}_T - \frac{1}{2}\log\det(\boldsymbol{K}_{\boldsymbol{X}_T\boldsymbol{X}_T} + \sigma^2\boldsymbol{I}) - \frac{n_T}{2}\log(2\pi), \quad (83)$$

where $\boldsymbol{X}_T \in \mathbb{R}^{n_T \times (d+1)}$ are the inputs (including the covariates and treatment), $\boldsymbol{y}_T \in \mathbb{R}^{n_T}$ are the observed outcomes, $\boldsymbol{K}_{\boldsymbol{X}_T\boldsymbol{X}_T}$ is the kernel matrix computed via the RBF kernel (note that our kernel construction is detailed in Sec. 4), $\sigma^2$ is the noise variance, and $n_T$ is the number of training samples. Additionally, hyperparameters are updated following each acquisition of new labels. Unless otherwise specified, we use the standard radial basis function (RBF) kernel. We have also evaluated alternative kernel choices, including Matern and RQ kernels, in Fig. 16.

**CME.** We use an RBF kernel for the CME, with the bandwidth selected via the median heuristic. The regularization parameter is a learnable parameter, initialized as $\lambda = 0.01$. Optimization is performed using the Adam optimizer with a learning rate of 0.05, and training is run for 100 epochs without early stopping or validation. To reduce computational cost in the active learning setting, we avoid retraining the CME model after each acquisition round by leveraging the closed-form conditional mean operator. In each round of CATE or ATT estimation, the kernel features of the adjustment variables **s** must be updated. Specifically, the CMEs $\mu_{\mathbf{s}|\mathbf{z}}$ and $\mu_{\mathbf{s}|\boldsymbol{a}'}$ need to be recomputed. This update can be done efficiently using the conditional embedding operator (details are in Sec. 4.2): instead of retraining, we only update the feature map $\Phi_{\boldsymbol{S}}$ within the operator. This update requires only a matrix multiplication and is thus computationally lightweight.

### F.2.3 BASELINE METHODS

We use existing implementations for most of the baseline methods:

1. **Random:** In this case, during each round, we randomly choose the required $n_b$ data points uniformly from the pool dataset.

2. **Query-based Heterogeneous Treatment Effect estimation (QHTE)[4]:** QHTE is a core-set method that uses a distance-based acquisition function to strategically select data points, enhancing the estimation of heterogeneous treatment effects (Qin et al., 2021). We adopt this method as a baseline for the semi-synthetic dataset. To ensure a fair comparison, we also use a Gaussian Process (GP) as the regression function. Similar to QHTE, we select the core sets in the feature space. In the GP case, we define the distance between two points as the posterior covariance, where a smaller distance indicates stronger relationships. We greedily select $n_b$ data points from the pool dataset during each round to ensure they serve as an effective core-set.

3. **CausalBALD[5]:** For CausalBALD (Jesson et al., 2021), we directly take the code of $\mu$-BALD as the information gain method applied to the pool dataset. While CausalBALD is originally designed for the binary case, it can be extended to the continuous case by treating each subgroup independently. As reported in their paper, $\mu$-BALD generally performs well, often achieving results comparable to the best methods.

4. **Total Variance Reduction (TVR):** For this method, we directly compute the posterior variance for all data points in the pool dataset (MacKay, 1992). We then calculate each data point's utility by measuring the total variance reduction, which is based on the sum of the posterior variances of all data points in the pool dataset.

For all baseline methods, to ensure a fair comparison, we use the same training configuration for the GP regression function as in our proposed method, including the learning rate, number of epochs, and validation dataset, as detailed in the implementation section above.

### F.3 HARDWARE

For the running time analysis and visualization experiments, we leverage a machine equipped with a 32-core Intel CPU and 128 GB of RAM. This setup provides sufficient resources for processing and handling the large-scale data required for our experiments. For running all the numerical results, we utilize Spartan, a high-performance computing platform. Each trial is typically run on an NVIDIA A100 (80GB) GPU, paired with an Intel CPU featuring 8 cores and 40 GB of memory.

---

[4] https://github.com/Qcer17/QHTE
[5] https://github.com/OATML/causal-bald

---

**Algorithm 1** Active Learning Framework for CATE Estimation

---

**Require:** Initial labeled dataset $\mathcal{D}_T = \{(\boldsymbol{x}_i, y_i)\}_{i=1}^{n_T}$, Pool of unlabeled data $\mathcal{D}_P = \{\boldsymbol{x}_i\}_{i=n_T+1}^{N}$, Query budget $n_B$, Batch size $n_b$, Utility function $U(\cdot)$

**Ensure:** Final trained CATE model $\hat{\tau}$

1: Train initial CATE model $\hat{\tau}_0$ on $\mathcal{D}_T$.                          # Train a baseline model
2: **for** $r = 1$ to $n_B/n_b$ **do**                          # Main active learning loop for each round
3:      Let $\hat{\tau}_{\text{current}} = \hat{\tau}_{r-1}$.                          # Use the model from the previous round
4:      Compute utility scores $u_i \leftarrow U(\boldsymbol{x}_i; \hat{\tau}_{\text{current}})$ for all $\boldsymbol{x}_i \in \mathcal{D}_P$.          # Evaluate pool points
5:      Select a batch of indices $\mathcal{I}_b$ of size $n_b$ from $\mathcal{D}_P$ based on scores $\{u_i\}$.          # e.g., Top-$k$ or greedy selection
6:      Let the selected batch be $\boldsymbol{X}_b = \{\boldsymbol{x}_i \mid i \in \mathcal{I}_b\}$.
7:      Query outcomes $\boldsymbol{Y}_b = \{y_i \mid i \in \mathcal{I}_b\}$ for the instances in $\boldsymbol{X}_b$.          # Oracle queries for labels
8:      Augment the labeled set: $\mathcal{D}_T \leftarrow \mathcal{D}_T \cup \{(\boldsymbol{x}_i, y_i) \mid \boldsymbol{x}_i \in \boldsymbol{X}_b, y_i \in \boldsymbol{Y}_b\}$.          # Add new data to training set
9:      Update the pool: $\mathcal{D}_P \leftarrow \mathcal{D}_P \setminus \boldsymbol{X}_b$.                          # Remove queried points from pool
10:     Retrain CATE model $\hat{\tau}_r$ on the updated labeled set $\mathcal{D}_T$.  # Improve the model with new data
11: **end for**
12: **return** The final trained model $\hat{\tau}_{n_B/n_b}$.

---

### F.4 METRICS

To evaluate the accuracy of our estimators, we use the Average Mean Squared Error (AMSE). This metric is computed over a predefined set of "points of interest," which represent the specific subpopulations and treatment scenarios relevant to the research question. Since calculating the AMSE requires access to the ground-truth causal effect $\tau$, this evaluation approach is suitable for synthetic data settings where the true data generating process is known. The general principle is to compute the mean squared error between the estimated effect $\hat{\tau}$ and the true effect $\tau$ across a set of $n_I$ interest points. The structure of these points varies depending on the causal quantity being estimated.

**AMSE for CATE.** For the Conditional Average Treatment Effect, the estimator $\hat{\tau}_{\text{CATE}}(a, \boldsymbol{z})$ is a function of both the treatment $a$ and the conditioning covariates $\boldsymbol{z}$. The set of interest points is therefore a collection of treatment-covariate pairs $(\boldsymbol{a}_I, \boldsymbol{Z}_I) = \{(a_i, \boldsymbol{z}_i)\}_{i=1}^{n_I}$. The AMSE is defined as:

$$\text{AMSE}_{\text{CATE}} \stackrel{\text{def}}{=} \frac{1}{n_I} \sum_{i=1}^{n_I} \left(\hat{\tau}_{\text{CATE}}(a_i, \boldsymbol{z}_i) - \tau_{\text{CATE}}(a_i, \boldsymbol{z}_i)\right)^2. \tag{84}$$

**AMSE for ATE and ATEDS.** For the Average Treatment Effect (ATE) and ATE under Distribution Shift (ATEDS), the estimators $\hat{\tau}_{\text{ATE}}(a)$ and $\hat{\tau}_{\text{ATEDS}}(a)$ are functions of the treatment $a$ only. Consequently, the set of interest points is a collection of different treatment values $\boldsymbol{a}_I = \{a_i\}_{i=1}^{n_I}$. The AMSE is defined as:

$$\text{AMSE}_{\text{ATE}} \stackrel{\text{def}}{=} \frac{1}{n_I} \sum_{i=1}^{n_I} \left(\hat{\tau}_{\text{ATE}}(a_i) - \tau_{\text{ATE}}(a_i)\right)^2. \tag{85}$$

The formula for ATEDS is analogous, using the respective estimator and true value.

**AMSE for ATT.** The Average Treatment Effect on the Treated, $\hat{\tau}_{\text{ATT}}(a, \tilde{a})$, is a function of both the target treatment $a$ and the conditioning treatment $\tilde{a}$. The set of interest points is therefore a collection of pairs $(\boldsymbol{a}_I, \tilde{\boldsymbol{a}}_I) = \{(a_i, \tilde{a}_i)\}_{i=1}^{n_I}$. The AMSE is defined as:

$$\text{AMSE}_{\text{ATT}} \stackrel{\text{def}}{=} \frac{1}{n_I} \sum_{i=1}^{n_I} \left(\hat{\tau}_{\text{ATT}}(a_i, \tilde{a}_i) - \tau_{\text{ATT}}(a_i, \tilde{a}_i)\right)^2. \tag{86}$$

#### F.4.1 UNCERTAINTY REDUCTION AS AN EVALUATION CRITERION

While our acquisition strategy is designed to reduce the uncertainty of the causal estimator, directly tracking this uncertainty reduction over acquisition rounds is an unsuitable primary metric for comparing different active learning methods. This is due to several fundamental challenges. First, the

model is retrained after each batch acquisition. This process, which often includes hyperparameter re-optimization, recalibrates the model's posterior distribution. Consequently, the scale and meaning of uncertainty are not consistent from one round to the next, making direct numerical comparisons of uncertainty values across rounds unreliable. Second, different acquisition strategies are designed to reduce different forms of uncertainty. For instance, BALD-based methods target the uncertainty in model parameters ($\boldsymbol{\theta}$), whereas our method targets the posterior variance of the causal quantity ($\tau$) itself. Evaluating all methods using any single, internal uncertainty measure would unfairly favor the method that directly optimizes that specific measure. Therefore, following established practice in active learning, we evaluate performance using an external, task-based metric that is independent of any specific acquisition function. Since the ultimate goal of all compared methods is to produce the most accurate causal estimator possible, the AMSE of the causal estimate is the most appropriate and equitable metric. It directly measures how effectively each strategy uses its budget to achieve the shared final objective.

### F.5 COMPUTATIONAL COMPLEXITY

We analyze the computational complexity for a single round of active learning. This process involves two main stages for each of the $n_P$ points in the pool set $\mathcal{D}_P$: (1) computing the posterior distribution of the causal quantity (e.g., CATE), and (2) evaluating the acquisition function based on this posterior. We denote the training set size as $n_T$. The complexity of any neural network components is omitted from this analysis.

**Stage 1: Posterior Computation Cost.** The cost of computing the posterior is dominated by the number of "query points" required to estimate the causal quantity. Let $m$ be the number of such query points. For example, for a CATE $\tau(a, \boldsymbol{z})$, $m$ would be the number of discrete values of $a$ and $\boldsymbol{z}$ we are interested in. The initial, one-time cost for inverting the training kernel matrix is $\mathcal{O}(n_T^3)$. After this, the cost for predicting the posterior for $m$ query points is:

- **CDE with Monte Carlo Sampling:** To estimate an integral (e.g., over $\mathbf{s}$), we sample $n_s$ points for each of the initial $n_q$ query points (e.g., $n_q$ different $\boldsymbol{z}$ values). This results in a total of $m_{\mathrm{MC}} = n_q \times n_s$ effective query points. The cost to obtain their joint posterior covariance is dominated by:

$$\mathcal{O}(n_T^2 \cdot m_{\mathrm{MC}} + m_{\mathrm{MC}}^2). \tag{87}$$

- **CME-based Method:** The CME method analytically handles the integral, avoiding the need for sampling. The number of effective query points remains $m_{\mathrm{CME}} = n_q$. The CME estimator itself requires a one-time kernel matrix inversion, typically on the training data, with a cost of $\mathcal{O}(n_T^3)$, which is subsumed by the main GP inversion cost. The posterior computation cost is then dominated by:

$$\mathcal{O}(n_T^2 \cdot m_{\mathrm{CME}} + m_{\mathrm{CME}}^2). \tag{88}$$

The key advantage of the CME method is that $m_{\mathrm{CME}} \ll m_{\mathrm{MC}}$, leading to a significant reduction in complexity, especially when a large number of Monte Carlo samples $n_s$ is required for accuracy.

**Stage 2: Acquisition Function Evaluation.** After obtaining the $m \times m$ posterior covariance matrix $\boldsymbol{\Sigma}_\tau$ for the causal quantity, we evaluate the acquisition function for each of the $n_P$ pool points.

- **IG:** This acquisition function is often related to the determinant of the posterior covariance matrix, representing the volume of the uncertainty ellipsoid. The cost of computing the determinant is $\mathcal{O}(m^3)$. To find the best point, this must be done for all pool points, leading to a total complexity for one round of:

$$\mathcal{O}(n_P \cdot (n_T^2 m + m^2 + m^3)). \tag{89}$$

- **TVR:** This acquisition function corresponds to the trace of the posterior covariance matrix, $\mathrm{Tr}(\boldsymbol{\Sigma}_\tau)$. The cost of computing the trace is only $\mathcal{O}(m)$. The total complexity for one round is therefore much lower:

$$\mathcal{O}(n_P \cdot (n_T^2 m + m^2)). \tag{90}$$

**Batch Selection.** When selecting a batch of $n_b$ points:

- **Top-$k$ Selection:** The simplest method is to compute the acquisition score for all $n_P$ points and select the top $n_b$. The cost is determined by the total complexity of the acquisition function evaluation, plus a negligible sorting cost of $\mathcal{O}(n_P \log n_P)$.
- **Greedy Selection:** A greedy approach selects points sequentially, updating the selection criteria at each step. A naive implementation would re-evaluate the acquisition function for all remaining pool points at each of the $n_b$ steps. For a complex, set-based acquisition function like batch IG, this can be prohibitively expensive, with complexity scaling with $n_b \cdot n_P$. More efficient greedy methods exist but are beyond the scope of this analysis (Holzmüller et al., 2023).

## F.6 ADDITIONAL EXPERIMENTAL RESULTS

In this section, we present additional ablation results, including an analysis of the robustness of our proposed methods with respect to different kernel choices, starting points, step sizes, and pool dataset sizes. All results are based on the visualization datasets. Additionally, we conduct various simulation setups and provide further results on both the simulations and the semi-synthetic IHDP data.

### F.6.1 VISUALIZATION

Here, we present the full set of results, including the AMSE metrics and additional visualizations of the acquired data points, complementing the results presented in the main paper. Specifically, for the acquired data visualizations, we focus on the CATE case, where the condition variable is fixed at a specific value, and we evaluate the performance across all possible treatments. The figure illustrates the joint distribution of the treatment and the adjustment variable $S1$. The blue points represent the scatter of the pool dataset, while the red points correspond to the scatter of the data points acquired by different methods. The red rectangle highlights the region of interest, which encompasses all possible treatments and the range of $S1$ conditioned on the fixed condition variable. From this figure, we can clearly observe the distribution shift between the target distribution, as indicated by the CATE estimator, and the distribution of the pool dataset. In this scenario, our eight proposed methods consistently approximate the target distribution more accurately than all baseline methods, resulting in improved estimation performance.

### F.6.2 ABLATION RESULTS

In this part, we present various ablation studies to evaluate the robustness of our data acquisition strategies. For active learning, the experimental setups we control include: (1) the size of the pool dataset, in Fig. 15 (2) the number of warm-start data points, in Fig. 14, (3) different batch acquisition sizes, in Fig. 12, (4) using soft acquisition trick for batch acquisition, in Fig. 18 and (5) different noises 13. Regarding our model, since we use GPs as the regression function, we provide two more kernel choices, the Rational Quadratic (RQ) kernel and the Matérn kernel, to assess the model's reliability and sensitivity to different kernel selections, in Fig. 16.

Fig. 12 shows that smaller batch sizes improve performance by enabling frequent retraining, refining uncertainty estimates, and enhancing adaptive sampling. This supports our conclusion that greedy acquisition approximates sequential single-point estimation. However, smaller batches also increase training iterations, balancing performance and efficiency.

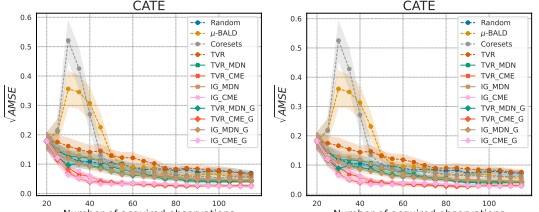

**Robustness to Heteroscedastic Noise.** To further evaluate the robustness of our framework against likelihood misspecification, specifically conditionally heteroscedastic errors, we conducted an additional experiment

Figure 10: Full results of the $\sqrt{\text{AMSE}}$ performance (with shaded standard error) for all methods on the Visualization dataset. From left to right: (1) In-distribution results; (2) Testing-distribution results.

using the visualization dataset setup. We modified the data generation process to introduce heteroscedasticity, where the noise variance explicitly depends on the covariate $s_1$. Specifically, the

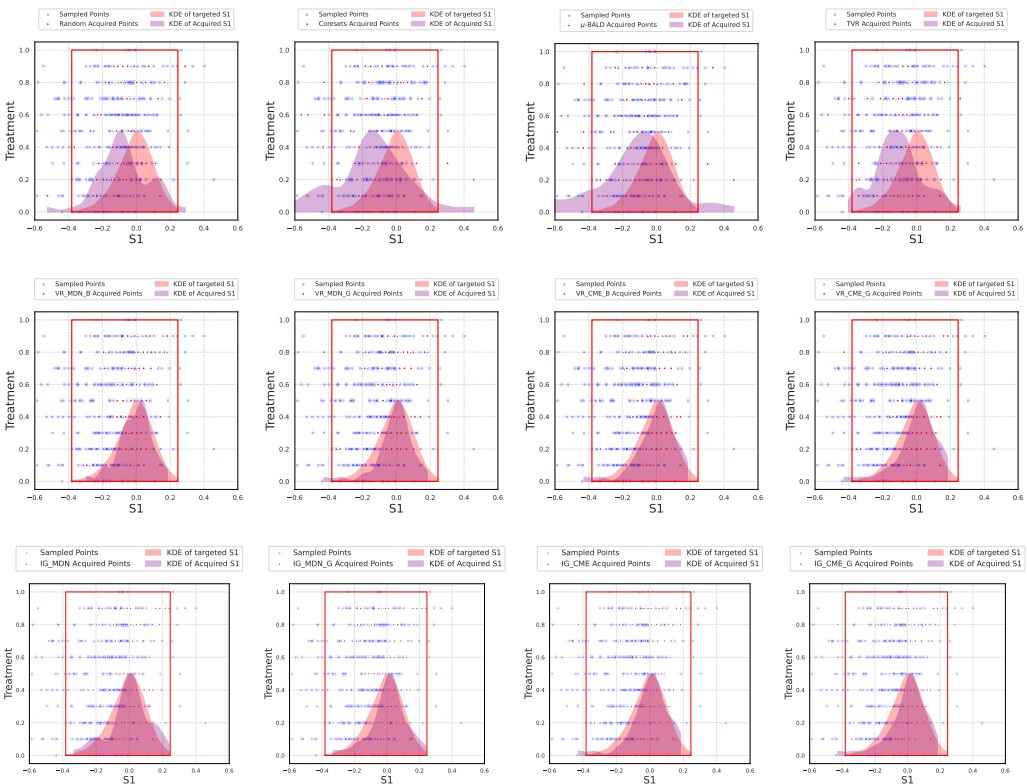

Figure 9: Acquired data points visualization.

outcome is generated as $y = \mu(x) + \epsilon$, with $\epsilon \sim \mathcal{N}(0, 0.5|s_1|)$. The results are presented in Fig. 11. We observe that our proposed methods, particularly the CME-based strategies, consistently outperform the baselines. This demonstrates that our framework effectively handles scenarios where uncertainty varies across the input space. Even under this deviation from the fixed-variance assumption of the standard GP likelihood, the ActiveCQ strategies successfully identify and query informative regions, maintaining superior sample efficiency.

Across all results, our CME-based greedy data acquisition strategies, including the IG-based and TVR-based methods, consistently demonstrate the best performance. In most cases, the naive method also performs comparably to the greedy approach, with the added benefit of significantly faster computation. In practice, when computational resources are limited, the naive method can be a practical alternative that delivers strong performance.

The CDE MC-sampling methods generally achieve second-tier performance across various settings. This is because they aim to reduce the uncertainty of the target estimator, focusing on the constructed distribution rather than the observational data represented by the pool dataset.

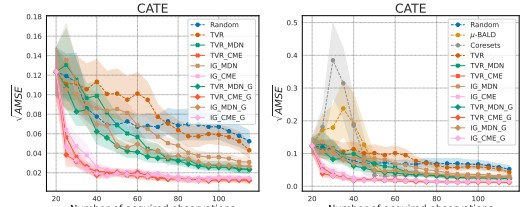

Figure 11: Full results of the $\sqrt{\text{AMSE}}$ performance (with shaded standard error) for all methods on the Visualization dataset. From left to right: (1) results for some methods for good visualization; (2) results for all methods.

However, as previously discussed, the CDE method struggles in high-dimensional scenarios and is sensitive to noisy covariates that do not influence treatment assignment or outcome values.

In contrast, the CME-based method iteratively refines the learned representation by adjusting the bandwidth hyperparameter of the RBF kernel, enabling it to better capture the influence of relevant variables. Building on this insight, leveraging neural network-based methods could further improve

performance by filtering out irrelevant covariates more effectively (Xu et al., 2021). Alternatively, one could apply independence tests between covariates, treatment, and outcomes to remove irrelevant variables (Zhang et al., 2018). However, this approach heavily depends on the reliability of the independence testing tool. If it incorrectly excludes important variables, it could adversely affect subsequent procedures. We leave these explorations for future work.

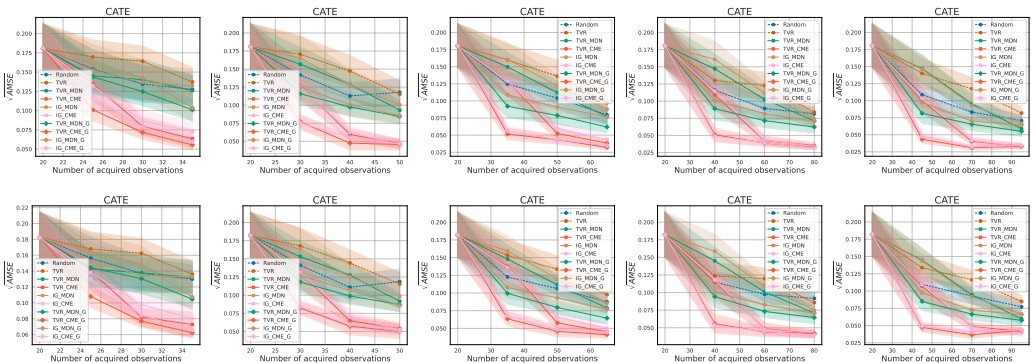

Figure 12: **One step acquisition / Different batch sizes.** The $\sqrt{\text{AMSE}}$ performance (with shaded standard error) for different batch sizes is presented. The first row illustrates the in-distribution performance, while the second row depicts the out-of-distribution performance. From left to right, the batch sizes are set to 5, 10, 15, 20, and 25.

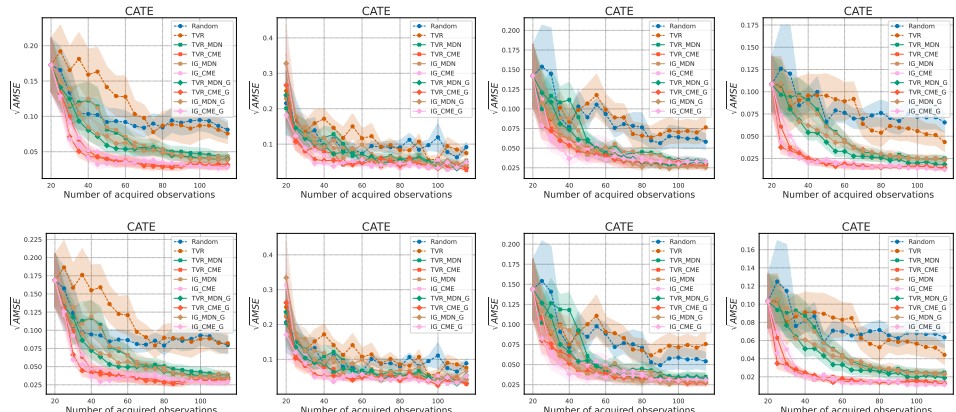

Figure 13: **Different noises.** The $\sqrt{\text{AMSE}}$ performance (with shaded standard error) for different batch sizes is presented. The first row illustrates the in-distribution performance, while the second row depicts the out-of-distribution performance. From left to right, the noises are normal, student, Laplace and uniform.

### F.6.3 SIMULATION RESULTS

In this part, we present additional experimental results on the simulation datasets, covering the CATE, ATE, ATT, and DS cases. Across all scenarios, we explore two primary treatment settings.

The first setting involves fixing the target treatment value, meaning we focus solely on estimating the causal effect for specific subgroups. This approach is particularly practical in real-world applications. For example, in personalized medicine, we might estimate the effect of a fixed dosage of a drug on patients with a specific genetic profile. Similarly, in educational interventions, we might evaluate the impact of a fixed number of tutoring sessions on students with a certain performance baseline.

The second setting, as discussed in the main paper, examines all possible causal effects across different treatment levels within specific subgroups. This broader perspective allows us to assess the variability of treatment effects. For instance, we could evaluate how varying dosages of a medication

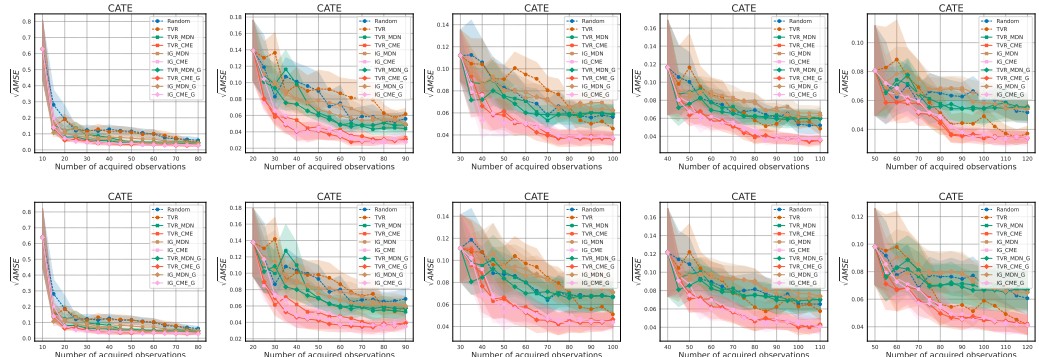

Figure 14: **Different starting points.** The $\sqrt{\text{AMSE}}$ performance (with shaded standard error) for different warm starting points is presented. The first row illustrates the in-distribution performance, while the second row depicts the out-of-distribution performance. From left to right, the starting poitns are set to 10, 20, 30, 40, and 50.

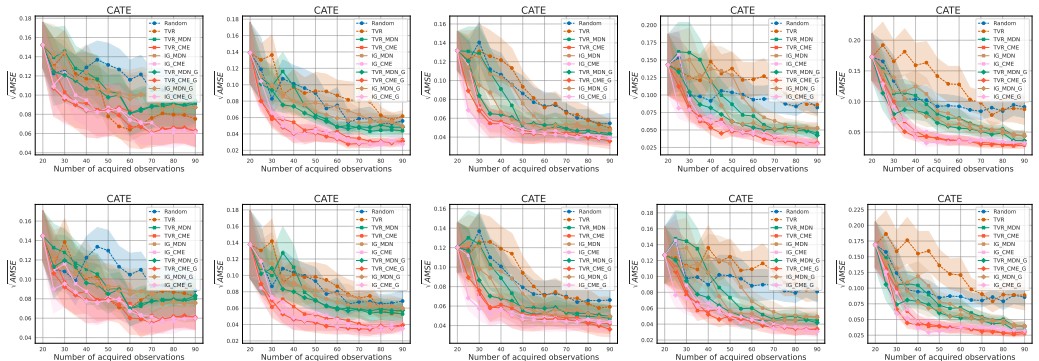

Figure 15: **Different sizes of pool datasets.** The $\sqrt{\text{AMSE}}$ performance (with shaded standard error) for different Pool dataset sizes is presented. The first row illustrates the in-distribution performance, while the second row depicts the out-of-distribution performance. From left to right, the Pool dataset sizes are set to 100, 150, 200, 300, 400, and 500.)

influence recovery rates for patients with a particular condition. Alternatively, we might analyze how different levels of government subsidies affect the economic outcomes of households in a given income bracket.

We would like to clarify why we use discrete treatments in all cases, which is driven by two main reasons. The first reason pertains to the IG-based method, which requires calculating the determinant of the posterior covariance matrix. This determinant is influenced by the size of the pool dataset and the number of possible treatments, where the size is the product of these two factors. In our setup, we use 600 observations in the training dataset, and the starting point is set to 100, meaning the pool dataset consists of 500 samples. If we were to use a continuous treatment, the worst-case scenario would involve 500 different treatment levels. This would result in a covariance matrix of size $250,000 \times 250,000$, which is computationally prohibitive for calculating the determinant. While it is possible to approximate this by sampling covariates and treatments, which is a reasonable approach in practice, it falls outside the scope of this paper. Therefore, we discretize the continuous treatment in most cases, excluding IG-based methods in the process.

The second reason relates to the simulation mechanisms in our dataset. When we focus on a fixed treatment value, the binary case is straightforward, as we can easily segment the data into treated and untreated groups. However, in the continuous treatment case, the treatment is determined by the covariates with added noise. To evaluate the performance of our estimator for a specific treatment, generating observations with a fixed treatment value is challenging. To avoid this issue, we discretize

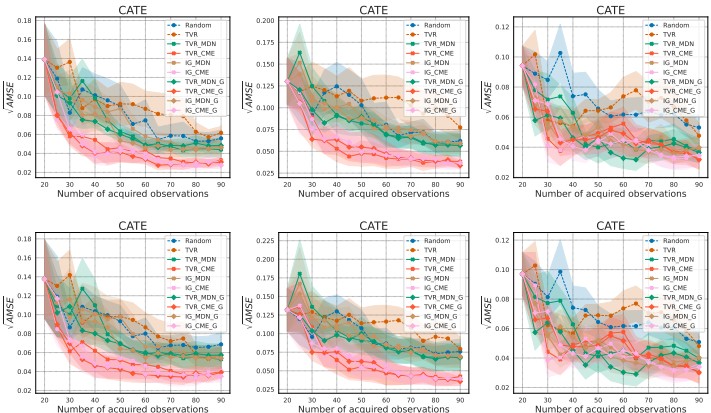

Figure 16: **Different choices of kernels.** The $\sqrt{\text{AMSE}}$ performance (with shaded standard error) for different kernel functions is presented. The first row illustrates the in-distribution performance, while the second row depicts the out-of-distribution performance. From left to right, the kernel functions are set to RBF, Matern and RQ.

the treatment in the fixed-treatment case and also use discrete treatments for all possible treatment scenarios.

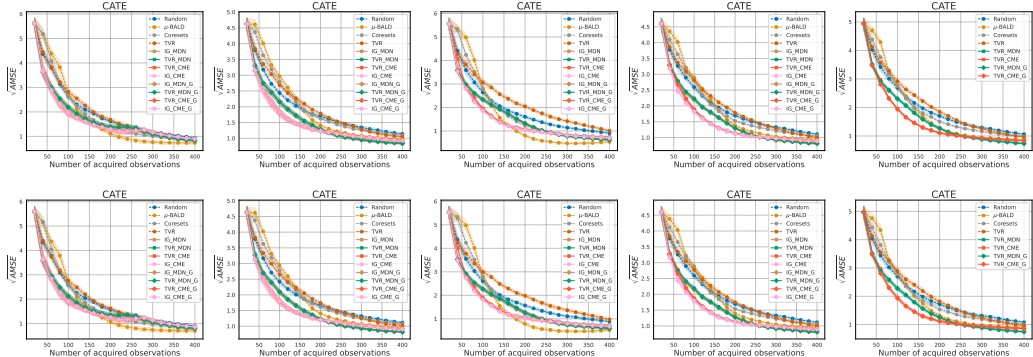

Figure 17: The $\sqrt{\text{AMSE}}$ performance (with shaded standard error) for the CATE case on simulation data is presented. The first row shows the in-distribution performance, while the second row illustrates the out-of-distribution performance. From left to right, the settings include: fixed and binary treatment, fixed and discrete treatment, all with binary treatment, all with discrete treatment, and all with continuous treatment.

**CATE.** In the CATE case, the results are shown in Fig. 17. We can see that our CME-based methods consistently show the best performance, and the CDE with MC sampling methods usually show the second-tier best performance. The benefit of our proposed methods over the baseline methods is substantial, since we fixed value of the conditioning variable, which means that the gap between the interested distribution and the distribution of the pool dataset is quite large. Moreover, we would like to explain why we need to learn the conditional distribution $\mathbb{P}_{\mathbf{s}|\mathbf{z}}$.

Even for the CATE case, to construct the interested distribution, the joint distribution $\mathbb{P}_{\mathbf{s},\mathbf{z}}$ is decomposed as $\mathbb{P}_{\mathbf{s},\mathbf{z}} = \mathbb{P}_{\mathbf{s}|\mathbf{z}} \times \mathbb{P}_{\mathbf{z}}$, meaning that we first sample the marginal distribution of $\mathbb{P}_{\mathbf{z}}$ to get the observations of the conditioning variable. This could be from a delta distribution if we only care about a fixed value of the conditioning variable, which implies that we only care about the causal effect over a specific subgroup, or from the observational distribution, which means that we care about all possible subgroups given different values of the conditioning variable. Then, we can sample from the conditional distribution $\mathbb{P}_{\mathbf{s}|\mathbf{z}}$ to obtain the observations of the adjustment variables.

Notice that the conditional distribution remains unchanged in these cases, meaning that we can directly take observations from the dataset as a surrogate for sampling from the conditional distribution. However, there are two main reasons why this is not sufficient. The first reason is that if we want to evaluate a subgroup, say $\mathbf{z} = \mathbf{z}^*$, and this value does not appear in the dataset, it is impossible to get any samples from the dataset. In this case, we need the flexible conditional distribution to sample from $\mathbb{P}_{\mathbf{s}|\mathbf{z}^*}$. The second reason is that we can sample more observations from the learned distribution, which helps reduce the variance and improve accuracy

Also, note that if one cares about all possible subgroups, it may not be strictly necessary to learn the conditional distribution. We believe that the primary benefit of our methods comes from the distribution shift caused by the treatment variable, from $\mathbb{P}_{\mathbf{a}}$ to the interested distribution $\mathbb{P}_{\mathbf{a}}^*$. This shift is determined by whether one cares about the performance over a fixed treatment or all possible treatments.

Here, we also incorporate the softmax-based batch acquisition strategy proposed by (Kirsch et al., 2023), which reweights the acquisition scores of candidate observations based on their sample importance within the pool dataset. This approach can accelerate the acquisition process and improve performance. We find that BALD-based methods benefit from this strategy, achieving stable performance and demonstrating compatibility. However, our method consistently outperforms BALD, whether using Top-b selection or greedy acquisition, as it directly targets uncertainty reduction in the target estimator. Notably, because our approach prioritizes predictive performance over reducing parameter uncertainty, the softmax-based reweighting appears to negatively affect its effectiveness.

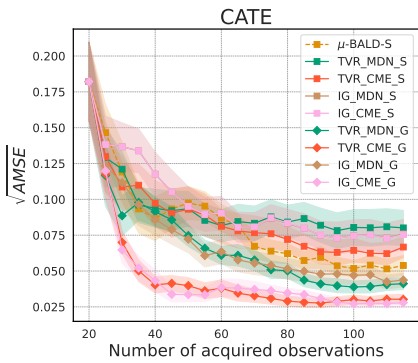

Figure 18: The $\sqrt{\text{AMSE}}$ performance (with shaded standard error) for different kernel functions. Methods with "S" use the soft-max acquisition trick.

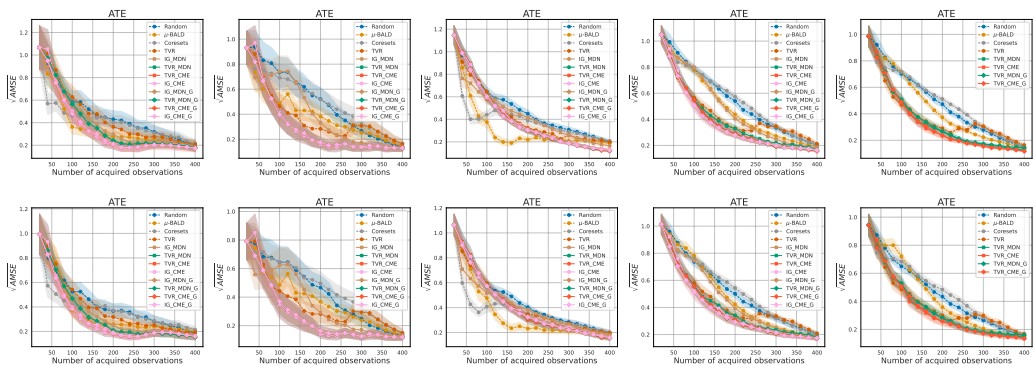

Figure 19: The $\sqrt{\text{AMSE}}$ performance (with shaded standard error) for the ATE case on simulation data is presented. The first row shows the in-distribution performance, while the second row illustrates the out-of-distribution performance. From left to right, the settings include: fixed and binary treatment, fixed and discrete treatment, all with binary treatment, all with discrete treatment, and all with continuous treatment.

**ATE.** For the ATE case, the results are shown in Fig. 19. We can observe that our methods consistently outperform the baseline methods across all different setups. As with the CATE case, we can conclude that when the interested distribution significantly differs from the observational distribution in the pool dataset, our methods perform much better than the baseline methods. Consequently, for all fixed treatment cases, our methods yield superior results. However, for the all-possible-treatment case in binary treatment scenarios, our methods show comparable performance to the baseline methods. This is because, to ensure that the overlap assumption holds, we do not modify $\mathbb{P}_{\mathbf{a}|\mathbf{s}}$ significantly.

Moreover, in the ATE case, the distribution shift arises solely from the shift in the treatment variables $\mathbb{P}_a$, which is why all methods show similar results in this case.

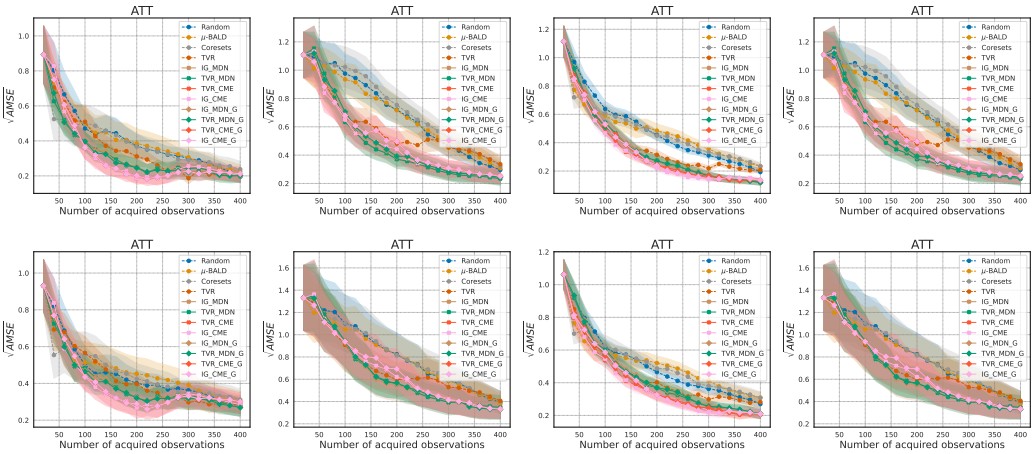

Figure 20: The $\sqrt{\text{AMSE}}$ performance (with shaded standard error) for the ATT case on simulation data is presented. The first row shows the in-distribution performance, while the second row illustrates the out-of-distribution performance. From left to right, the settings include: fixed and binary treatment, fixed and discrete treatment, all with binary treatment, all with discrete treatment.

**ATT.** For the ATT case, the results are shown in Fig. 20. We draw a similar conclusion as before: our proposed methods consistently outperform the baseline methods, although the performance improvement is not as substantial. We believe this is due to the data simulation, where the covariates are high-dimensional, and to ensure the overlap assumption, we avoid making significant distribution shifts in the treatment selection. This is further supported by the visualizations of the datasets shown in Fig. 8 and Fig. 7, which illustrate that $\mathbb{P}_{s|a}$ does not change significantly when constructing the interested distribution.

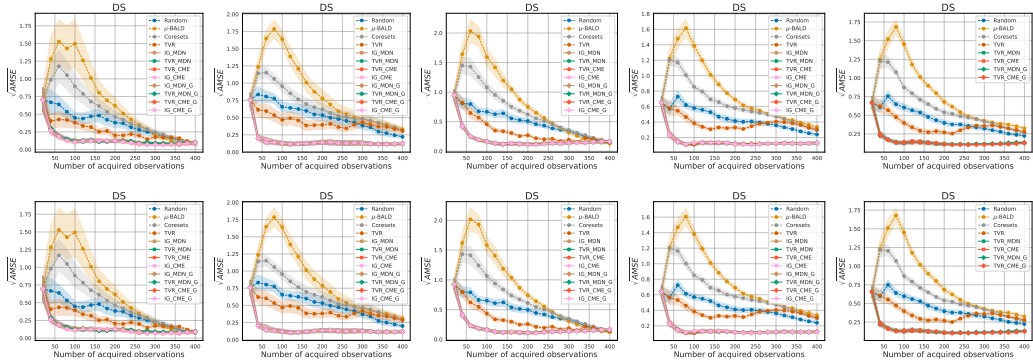

Figure 21: The $\sqrt{\text{AMSE}}$ performance (with shaded standard error) for the DS case on simulation data is presented. The first row shows the in-distribution performance, while the second row illustrates the out-of-distribution performance. From left to right, the settings include: fixed and binary treatment, fixed and discrete treatment, all with binary treatment, all with discrete treatment, and all with continuous treatment.

**ATEDS.** For the ATEDS case (DS for short), the results are shown in Fig. 21. We believe this case provides the most compelling evidence to support the benefits of our proposed methods. Across all setups, our methods consistently show significantly better results compared to the baseline methods. The performance improvement can be attributed not only to the changes in the treatment variable distribution $\mathbb{P}_a$ but also to the distribution shifts in the covariates $\mathbb{P}_s$, which we have intentionally

manipulated in this setting. Consequently, we observe a substantial performance improvement, which further highlights the importance of targeting the desired distribution when acquiring data points.

Also, note that for the ATE and DS cases, in the setup with fixed value treatment, the IG-based methods will perform identically to the TVR-based methods. This is because the target estimator collapses to the same formulation, where the target becomes a point estimate and the posterior of this estimator is a normal distribution. In this case, minimizing entropy and minimizing variance become equivalent.

In conclusion, from the above results, we can derive the following key takeaway:

> 🔑 **Takeaway Message:** The benefits of CQ-specific data acquisition strategies are more pronounced when there is a significant distribution shift between the target distribution and the pool dataset distribution..

### F.6.4 IHDP RESULTS

**CATE.**    In this part, we present additional experimental results on the IHDP datasets, covering the CATE, ATE, ATT, and DS cases. Across all scenarios, we explore two primary treatment settings as shown before in the simulations. When studying the IHDP dataset, since all covariates in this dataset are fixed and the treatment assignment for the binary case is deterministic, it results in highly imbalanced data. However, for continuous treatment, we can design the specific form of the treatment assignment ourselves and must determine the outcome regression function.

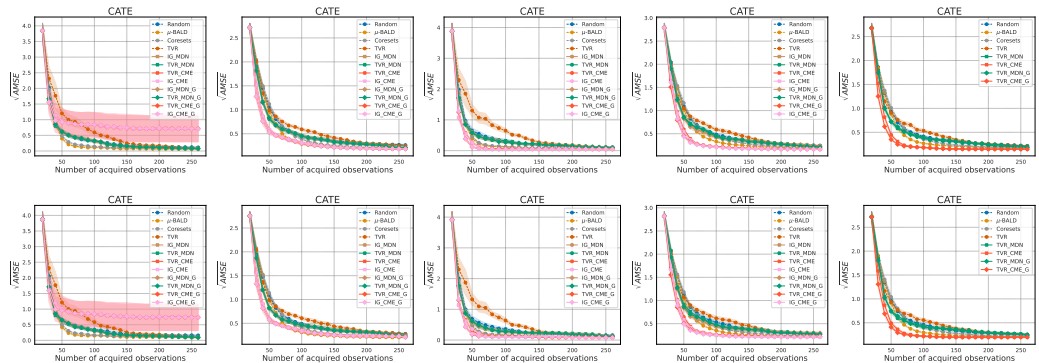

Figure 22: The $\sqrt{\text{AMSE}}$ performance (with shaded standard error) for the CATE case on IHDP data is presented. The first row shows the in-distribution performance, while the second row illustrates the out-of-distribution performance. From left to right, the settings include: fixed and binary treatment, fixed and discrete treatment, all with binary treatment, all with discrete treatment, and all with continuous treatment.

In the CATE case, the results are shown in Fig. 22. We observe that, in most cases, our CME-based method consistently outperforms the baseline methods. However, for the fixed-value treatment case, our method performs poorly due to extreme data imbalance and the use of a binary kernel, which inherently prevents the exploration of shared correlations between the two subgroups.

**ATE and ATT.**    In the ATE and ATT cases, the results are presented in Fig. 23 and Fig. 24. An interesting observation is that, across all scenarios, none of the methods consistently outperform random acquisition. This outcome can be attributed to the minimal distribution shift in these cases. Additionally, the combination of high-dimensional features and a limited number of observations hinders effective feature learning, which adversely impacts the precise representation of uncertainty in data points and, consequently, the overall performance.

**ATEDS.**    For the ATE with distribution shift case, the results are shown in Fig. 25. Once again, our proposed methods demonstrate consistently better performance compared to all baseline methods. This improvement can be attributed to the large distribution shift, which amplifies the advantage of our methods over the baseline approaches.

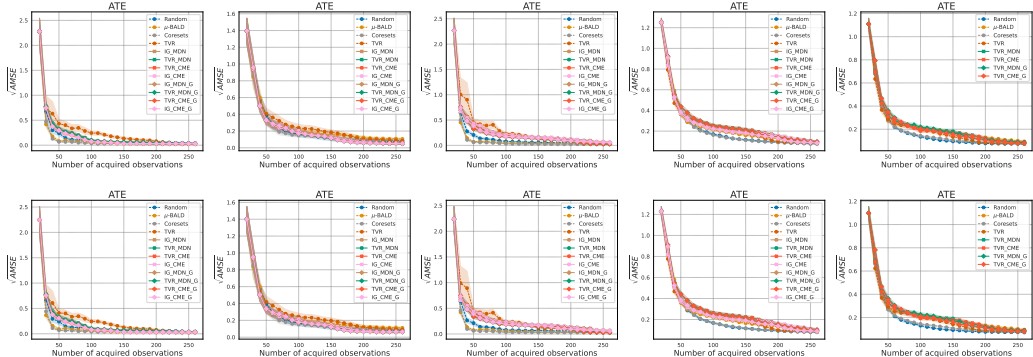

Figure 23: The $\sqrt{\text{AMSE}}$ performance (with shaded standard error) for the ATE case on IHDP data is presented. The first row shows the in-distribution performance, while the second row illustrates the out-of-distribution performance. From left to right, the settings include: fixed and binary treatment, fixed and discrete treatment, all with binary treatment, all with discrete treatment, and all with continuous treatment.

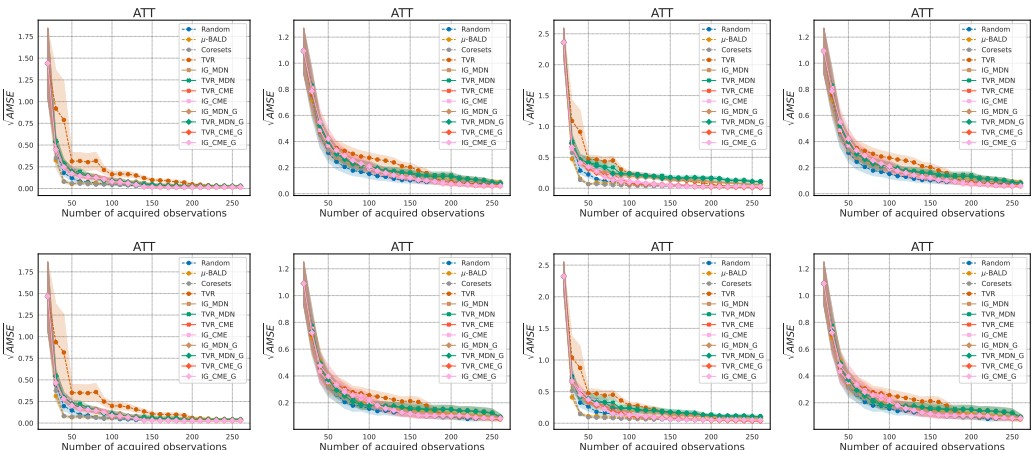

Figure 24: The $\sqrt{\text{AMSE}}$ performance (with shaded standard error) for the ATT case on IHDP data is presented. The first row shows the in-distribution performance, while the second row illustrates the out-of-distribution performance. From left to right, the settings include: fixed and binary treatment, fixed and discrete treatment, all with binary treatment, all with discrete treatment.

### F.6.5 EXPERIMENTAL RESULTS ON LALONDE

Our first task was to actively estimate the CATE. Tab. 1 presents the performance comparison between our proposed methods and several strong baselines. The results demonstrate that our methods, particularly the CME-based variants (IG-CME and TVR-CME), consistently and significantly outperform the baselines across all acquisition budgets. For instance, after acquiring only 60 labels, our TVR-CME method achieves an AMSE of $8.2 \pm 0.5$, whereas the best-performing baseline, Coresets, only reaches a similar error level after acquiring nearly 200 labels. This highlights our framework's superior sample efficiency in selecting informative individuals that are directly relevant to the CATE estimator.

The second task evaluated the framework's ability to handle a distribution shift when estimating the ATE. As shown in Tab. 2, our methods again demonstrate a clear advantage. They reduce the estimation error much more rapidly than the baselines in the initial stages of acquisition and maintain superior performance throughout the process. Interestingly, in this setting, both the MDN-based and CME-based variants of our approach perform comparably well after the first 60 samples, and both are significantly better than the baselines. This result validates our framework's ability to effectively target the relevant data distribution, even when it differs from the source population.

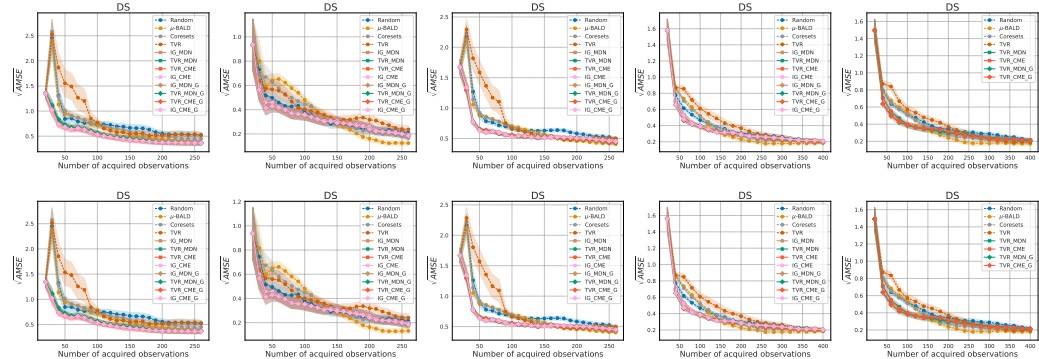

Figure 25: The $\sqrt{\text{AMSE}}$ performance (with shaded standard error) for the DS case on IHDP data is presented. The first row shows the in-distribution performance, while the second row illustrates the out-of-distribution performance. From left to right, the settings include: fixed and binary treatment, fixed and discrete treatment, all with binary treatment, all with discrete treatment, and all with continuous treatment.

Table 1: Performance comparison on the LaLonde dataset for active CATE estimation. Lower AMSE is better.

| Method | 20 | 60 | 100 | 140 | 180 | 200 |
|---|---|---|---|---|---|---|
| Random | $21.7 \pm 0.7$ | $16.1 \pm 0.5$ | $13.6 \pm 0.5$ | $12.1 \pm 0.5$ | $11.0 \pm 0.5$ | $10.8 \pm 0.5$ |
| Variance Reduction | $21.7 \pm 0.7$ | $18.2 \pm 1.0$ | $16.0 \pm 1.2$ | $12.7 \pm 0.9$ | $11.1 \pm 0.3$ | $10.7 \pm 0.4$ |
| Coresets | $21.7 \pm 0.7$ | $14.2 \pm 0.8$ | $12.1 \pm 0.6$ | $10.1 \pm 0.8$ | $8.6 \pm 0.6$ | $8.2 \pm 0.5$ |
| μ-BALD | $21.7 \pm 0.7$ | $15.0 \pm 1.2$ | $9.0 \pm 0.6$ | $8.0 \pm 0.7$ | $7.0 \pm 0.6$ | $6.7 \pm 0.6$ |
| IG-MDN (Ours) | $21.7 \pm 0.7$ | $11.9 \pm 0.6$ | $10.0 \pm 0.6$ | $8.9 \pm 0.6$ | $8.3 \pm 0.5$ | $8.1 \pm 0.5$ |
| TVR-MDN (Ours) | $21.7 \pm 0.7$ | $11.8 \pm 0.6$ | $9.9 \pm 0.6$ | $8.8 \pm 0.6$ | $8.2 \pm 0.5$ | $7.9 \pm 0.5$ |
| IG-CME (Ours) | $21.7 \pm 0.7$ | $8.3 \pm 0.5$ | $6.2 \pm 0.5$ | $5.9 \pm 0.5$ | $5.9 \pm 0.5$ | $5.9 \pm 0.5$ |
| TVR-CME (Ours) | $21.7 \pm 0.7$ | $8.2 \pm 0.5$ | $6.1 \pm 0.5$ | $5.8 \pm 0.5$ | $5.6 \pm 0.5$ | $5.8 \pm 0.4$ |

Table 2: Performance comparison on the LaLonde dataset for active ATE with distribution shift estimation. Lower AMSE is better.

| Method | 20 | 60 | 100 | 140 | 180 | 200 |
|---|---|---|---|---|---|---|
| Random | $22.6 \pm 0.8$ | $16.1 \pm 1.0$ | $13.9 \pm 1.0$ | $12.8 \pm 0.9$ | $11.3 \pm 1.0$ | $11.0 \pm 1.0$ |
| Variance Reduction | $22.6 \pm 0.8$ | $19.7 \pm 1.0$ | $14.7 \pm 1.5$ | $13.3 \pm 1.4$ | $12.2 \pm 1.2$ | $11.9 \pm 1.2$ |
| Coresets | $22.6 \pm 0.8$ | $15.2 \pm 1.2$ | $12.8 \pm 1.1$ | $10.6 \pm 1.1$ | $9.3 \pm 1.2$ | $8.9 \pm 1.1$ |
| μ-BALD | $22.6 \pm 0.8$ | $16.0 \pm 1.6$ | $8.2 \pm 1.1$ | $7.7 \pm 1.0$ | $7.5 \pm 1.1$ | $7.2 \pm 1.0$ |
| IG-MDN (Ours) | $22.6 \pm 0.8$ | $9.4 \pm 1.0$ | $7.8 \pm 1.0$ | $7.5 \pm 1.0$ | $7.3 \pm 1.0$ | $7.2 \pm 1.0$ |
| TVR-MDN (Ours) | $22.6 \pm 0.8$ | $9.1 \pm 1.0$ | $7.3 \pm 1.0$ | $7.1 \pm 1.0$ | $7.1 \pm 1.0$ | $7.1 \pm 1.0$ |
| IG-CME (Ours) | $22.6 \pm 0.8$ | $9.3 \pm 1.1$ | $7.8 \pm 1.0$ | $7.5 \pm 1.0$ | $7.3 \pm 1.0$ | $7.2 \pm 1.0$ |
| TVR-CME (Ours) | $22.6 \pm 0.8$ | $9.1 \pm 1.0$ | $7.2 \pm 1.1$ | $7.1 \pm 1.0$ | $7.1 \pm 1.0$ | $7.1 \pm 1.0$ |

## G  DISCUSSIONS

### G.1  SUMMARY OF CONTRIBUTIONS AND FUTURE DIRECTIONS

A key challenge in causal inference is that target CQs are often defined by integrating over specific sub-populations that differ from the overall observed population. For instance, CATE is defined by the conditional distribution $p(\mathbf{s}|\mathbf{z})$, while ATT is defined by $p(\mathbf{s}|\mathbf{a})$. Consequently, the central principle of our work is that an effective active learning strategy must target the uncertainty relevant to these specific sub-populations. Instead of selecting data to reduce the global uncertainty of the underlying outcome model f, our framework prioritizes samples that maximally reduce the posterior uncertainty

of the final causal estimator $\hat{\tau}$ itself. While this principle of targeted uncertainty reduction establishes a strong foundation, our framework has several limitations that highlight promising avenues for future research.

**Relaxing Identification Assumptions.** Our current work operates under the standard assumption of no unmeasured confounders. A significant avenue for future work is to extend our active learning framework to more complex scenarios where this assumption is violated. This involves incorporating established identification strategies such as the front-door criterion (Dance et al., 2024; Chau et al., 2021b), instrumental variables (Xu et al., 2021), or proxy variables (Mastouri et al., 2021). Since many Bayesian and GP-based methods already exist for these settings, our framework can be naturally extended.

**Model Architecture and High-Dimensionality.** From a meta-learning perspective, our current model is a T-learner. An important extension would be to explore architectures that share statistical strength across treatment groups, such as S-learners or multi-task GPs (Alaa & Van Der Schaar, 2017). Furthermore, our current CME-based approach can be challenged by high-dimensional covariates. Integrating feature selection techniques (Witte et al., 2020) or neural network-based feature extractors (Johansson et al., 2022) would be a crucial step to enhance the framework's applicability to high-dimensional data.

**Scalability with Bayesian Neural Networks.** While GPs offer theoretical guarantees and closed-form posteriors, their cubic complexity limits scalability. Replacing the GP with a Bayesian Neural Network (BNN) (Jesson et al., 2021) is a compelling direction for future work. BNNs offer superior scalability for large datasets and greater flexibility in handling complex, high-dimensional inputs. They can learn feature representations directly from data, removing the need for kernel engineering, while still providing the necessary uncertainty quantification for active learning.

**From Estimation to Policy Optimization.** While our current focus is on minimizing the estimation error of the Causal Quantity (e.g., $\hat{\tau}_{\text{CATE}}$) globally, a compelling direction for future work is to target downstream decision-making directly. In many high-stakes applications, such as personalized medicine, the ultimate goal is to identify the optimal treatment policy $\pi(z)$ that maximizes patient welfare. This decision-theoretic objective shifts the priority from precise estimation across the entire domain to accurate sign determination near the decision boundary. Our ActiveCQ framework provides a unique advantage for this transition: unlike methods that rely on information-theoretic proxies (e.g., maximizing mutual information with model parameters (Jesson et al., 2021; Fawkes et al., 2025)), our derivation of the analytic posterior distribution for the causal estimator itself enables the direct calculation of the Expected Value of Information (EVOI). This capability allows for the design of acquisition functions that explicitly maximize the expected gain in downstream policy value, aligning with emerging frontiers in regulatory-confirmed adaptive trials (Klein et al., 2025).

## G.2 THE ADVANTAGES OF CME IN THE ACTIVE LEARNING SETUP

The choice of CMEs over traditional CDEs offers several crucial advantages within our active learning framework. Foremost is its *adaptability to learned features*. In each round of the AL loop, as the GP retrains on new data and refines its kernel-induced feature space, the CME can be re-estimated using this updated representation. This ensures the distribution embedding is always expressed in the most relevant, data-informed feature space, becoming progressively more aligned with the regression task.vSecond, CMEs exhibit superior *task alignment*. Our ultimate goal is to compute an expectation (an integral), which is precisely what the CME is designed for, representing the distribution as a mean element in an RKHS. CDEs, in contrast, attempt to model the entire density function, a more general and often unnecessarily complex task, introducing extra overhead and potential for misspecification. The CME provides a more direct and efficient path to the quantity of interest.vFinally, CMEs are particularly well-suited for *high-dimensional settings*. Unlike many density estimation techniques that suffer from the curse of dimensionality, CMEs can gracefully handle higher-dimensional data. By imposing fewer restrictive assumptions on the distribution's form, they offer a more flexible and robust alternative, making them a powerful and appropriate choice for the ActiveCQ task.

