# OpenReview forum: "ActiveCQ: Active Estimation of Causal Quantities"
_ICLR.cc/2026/Conference — ICLR 2026 Poster_

### Official Review · Reviewer_bTeT · 2025-10-30

**Soundness:** 2
**Presentation:** 2
**Contribution:** 2
**Rating:** 4
**Confidence:** 3

**Summary:**

Strengths:This work makes both remarkable conceptual and methodological contributions: it delineates the new task of Active Estimation of Multiple Causal Quantities in a form that is immediately applicable to the real-world challenge of costly outcome measurement and begins closing the theoretical gap in CQ-aware active learning. On a methodological front, it would be difficult to find a more innovative way of marrying GP and CME than this paper, producing a coherent framework for uncertainty modeling in the RKHS, where analytic integration over target distributions replaces Monte Carlo sampling noise. Meanwhile,the paper sets a good basis in theory for consolidation and better portability to further acquisition strategies by summarizing the various causal quantities ATE, CATE, ATT, and DS into one integral formulation.Weaknesses:The paper remains thin on empirical validation; most experiments are run on simulated and, at best, semi-synthetic datasets (IHDP, LaLonde) without any observational reality, so it is challenging to talk about the paper with practical implications. Besides, the "compatibility of the GP kernel space with the conditional embedding" is claimed but not empirically tested, so this essential assumption is not empirically validated. Third, while the author blames the distributional mismatch for the poor performance of the methods reviewed, such a mismatch is not quantified, nor is any evidence provided that the performance boost observed for ActiveCQ results from addressing such an issue. Therefore, there is a logical gap in the causal reasoning

**Strengths:**

na

**Weaknesses:**

na

**Questions:**

na

---

> ### Author Response · Authors · 2025-11-20
>
> Thank you so much for your efforts in evaluating our manuscript. In the following, we will respond to your concerns point by point.
>
> ---
> **[Q1. The paper remains thin on empirical validation; most experiments are run on simulated and, at best, semi-synthetic datasets (IHDP, LaLonde) without any observational reality, so it is challenging to talk about the paper with practical implications.]**
>
> We thank the reviewer for this comment regarding empirical validation. We understand the preference for real-world data, but we respectfully argue that our evaluation methodology is rigorously aligned with the scientific standards of causal inference for the following reasons:
>
> 1.  **The Necessity of Semi-Synthetic Benchmarks:** The fundamental problem of causal inference is the impossibility of observing counterfactual ground truth in purely real-world datasets. Without ground truth, it is scientifically impossible to calculate the estimation error (AMSE) to validate a method's accuracy. This is why datasets such as IHDP and LaLonde have become widely used benchmarks in the community. They offer a useful balance:
>     * **Observational Reality:** They use **real-world covariates** from actual clinical/economic studies, preserving the complex correlations, imbalances, and selection biases of reality.
>     * **Scientific Rigor:** The synthetic outcome process provides the known ground-truth $\tau$ required for quantitative evaluation.
>
> 2.  **Robustness to "Real-World" Complexity (App. Fig. 12):** We acknowledge that real-world data often violates ideal modeling assumptions. To address this "practical implication" concern directly, we conducted extensive ablation studies testing our framework against two distinct types of realistic noise: Non-Gaussianity (Fig. 12): We tested against Heavy-tailed (Student-t, Laplace) distributions to simulate outliers. Heteroscedasticity (New Experiment, Fig. 11): We explicitly introduced conditionally heteroscedastic errors (where noise variance depends on covariates) to simulate varying uncertainty across the population. In both cases, our method maintained superior performance, demonstrating its practical robustness to the messy, non-ideal conditions typical of observational reality.
>
> 3.  **Theoretical Foundation:** Finally, our contribution is not merely empirical. We provide the first formal identification of the transductive nature of Active Causal Estimation. Our framework addresses the root cause of the objective mismatch in prior methods, supported by convergence guarantees (**Thm. 2**).
>
> In summary, by demonstrating superior sample efficiency on realistic benchmarks and robust performance under non-ideal noise conditions, we provide strong evidence of the framework's practical value for high-stakes domains where outcomes are expensive.
>
>
> ---
> **[Q2. Besides, the "compatibility of the GP kernel space with the conditional embedding" is claimed but not empirically tested, so this essential assumption is not empirically validated.]**
>
> We thank the reviewer for this sharp technical observation. We believe the confusion stems from our notation in Eq. (6), where we expressed the CATE as an RKHS inner product involving $f$. We would like to clarify that while Eq. (6) serves as a notational analogue, our framework and estimators are mathematically grounded in Eq. (4), where the causal quantity is defined as an integral of the GP.
>
> 1.  **Mathematical Validity via Integration:** Since the target causal quantity $\hat{\tau}$ is a linear functional (an integral) of the GP $f$, it also remains a GP. The posterior mean and covariance in Eq. (7) (the only quantities used by our acquisition functions) follow directly from standard GP integration properties (e.g., $\mathbb{E}[\int f] = \int \mathbb{E}[f]$). These operations are mathematically well defined for common kernels such as the RBF kernel and do not require the “nuclear dominance’’ condition associated with path-wise inner products.
> 2.  **Clarifying "Compatibility":** In this correct integration-based view, the "compatibility" requirement is straightforward: it simply means that the kernel used to model the GP covariance must be the same as the kernel used to estimate the CME. In our experiments, we satisfy this by construction (using the same RBF kernel for both).
> 3.  **Theoretical Support:** Recent work (e.g., Sec. 4.1 of *Kanagawa et al., 2025* [1]) discusses precisely how this integral formulation serves as a valid analogue to the inner product for GPs.

---

> ### Author Response · Authors · 2025-11-20
>
> **[Q3. Third, while the author blames the distributional mismatch for the poor performance of the methods reviewed, such a mismatch is not quantified, nor is any evidence provided that the performance boost observed for ActiveCQ results from addressing such an issue. Therefore, there is a logical gap in the causal reasoning.]**
>
>
> We thank the reviewer for this critical question regarding the logical link between distributional mismatch and our performance. We argue that our experimental design **directly establishes this link**. The evidence is provided through (1) the correlation between mismatch severity and performance gain, and (2) direct visual verification of the sampling mechanism.
>
> **1. Evidence: Performance Gain Scales with Mismatch Severity**
>
> We designed our experiments to vary the degree of distributional mismatch. The results confirm that our method's advantage grows precisely when the mismatch becomes the dominant bottleneck:
>
> * **Implicit Mismatch (ATE/CATE):** In standard tasks (Fig. 2, Left), the mismatch is "inherent" (generalizing from factual $p_{\text{pool}}$ to counterfactual $\mathbb{P}^*$). Here, baselines struggle but function, and our method provides a consistent, moderate boost.
> * **Explicit & Severe Mismatch (ATEDS):** In the ATEDS experiment (Fig. 2, Right), we explicitly constructed a severe distributional shift between the source and target populations. This effectively "quantifies" the mismatch as maximal.
>     * **Result:** Under this severe mismatch, standard baselines (TVR, BALD) failing completely (high error), while our target-aware method maintains low error.
>     * **Logical Conclusion:** The fact that our performance advantage explodes exactly when the distributional mismatch is maximized provides the direct causal evidence that our method works by addressing this specific issue.
>
> **2. Mechanistic Evidence: Visualization (Fig. 3)**
>
> We provide direct visual evidence of the mechanism in Fig. 3.
> * **The Quantification:** The plot explicitly shows the divergence between the pool distribution and the target distribution, the acquired points and the target areas.
> * **The Solution:** It visually confirms that baseline strategies continue to sample from the pool density, whereas our strategy (ActiveCQ) shifts sampling to the target density. This visualizes the "addressing" of the issue in action.
>
> **3. Theoretical Foundation: The Inherent Nature of the Problem**
>
> Finally, we clarify that this mismatch is foundational to the problem definition, not an artifact. Unlike standard AL, which targets the pool distribution $p_{\text{pool}}$, Active Causal Estimation inherently targets a different distribution $\mathbb{P}^*_{\text{CQ}}$ (due to the unobserved nature of counterfactuals or specific subgroup definitions like ATT). Our framework is mathematically derived to close this specific gap, which standard AL objectives (like TVR or BALD) theoretically cannot address.
>
> [1] Kanagawa et al., Gaussian Processes and Reproducing Kernels: Connections and Equivalences, arxiv2025.

---

### Official Review · Reviewer_TTvj · 2025-10-31

**Soundness:** 4
**Presentation:** 4
**Contribution:** 3
**Rating:** 8
**Confidence:** 4

**Summary:**

The paper considers the problem of active learning when the goal is estimation of causal quantities like the average treatment effect (ATE) or conditional average treatment effect (CATE). More precisely, the authors consider settings in which there is available a (generally small) dataset containing outcomes $y$, treatments $a$, confounders $s$, and effect modifiers $z$, and also a larger dataset containing the same observables other than outcomes. One can obtain outcomes for selected observations in the second larger data set but at some cost. The problem is then to choose the data-point (or set of points) in the large sample, so that obtaining outcome information for that points leads to the greatest reduction in uncertainty in the target causal quantity of interest.

The approach of the authors is Bayesian, with Guassian process priors on the relevant features of the joint distribution of the observables and the outcomes assumed to be conditionally Gaussian with fixed conditional variance. The use of RKHS leads to tractable formulations for the posterior distribution of the causal quantities of interest. Points can be selected to achieve the greatest expected reduction in posterior uncertainty, as measured by either information gain or a total variance reduction criterion. A greedy algorithm is proposed for tractable selection of batches of points (selecting the optimal batch of $m$ points out of a large sample is computationally infeasible when $m$ is anything other than very small, for obvious combinatoric reasons). The performance of the method is assessed against benchmarks on semi-synthetic and simulated data and appears to perform very favourably.

In my reading, the key technical contribution here is as follows. Causal quantities depend on both the regression function ($E[y|a,z,s]$ in the authors' notation) and often also on the joint/conditional distribution of $z$, $s$ and (possibly) $a$. Two examples of causal quantities that depend on the latter are the ATE and CATE conditional only on $z$ (i.e., $E[y|1,z]-E[y|0,z]$). For standard active learning, it generally suffices to target the posterior of the regression function. The authors' approach is premised on the observation that many causal quantities of interest can be written as integrals of the regression function and the joint distribution or related objects. A key insight of the paper is then that, by expressing the joint distribution in terms of conditional kernel mean embeddings (CME) (an approach employed in some other recent papers on causal learning in RKHS), an analytic expression for the posterior of the causal quantities can still be obtained. The authors do consider an alternative based on kernel density estimation and Monte Carlo sampling, but the CME approach appears more tractable and elegant.

**Strengths:**

I enjoyed reading the paper. I found it to be well-written. The discussion of the analysis and approach is thorough. The question is novel and the methods are innovative and practically relevant. The analysis is very elegant. I found myself convinced that the methods proposed by the authors are effective.

**Weaknesses:**

The could be a little more detail given regarding possible real-world applications. The figures are too small. The grammar in some parts of the appendices needs fixing.

**Questions:**

I wondered which if any of the simulated/semi-synthetic datasets violate the assumption that the outcome is conditionally Gaussian with fixed variance? While the likelihood is misspecified if this is the case, the comparison on such potentially less favorable datasets might be quite informative as in the real world conditionally heteroskedastic errors may be very common.

One reason that the CATE may be of interest is that knowledge of the CATE can allow researchers to optimally allocate treatment in some new population on the basis of the covariates $z$. More precisely, one may hope to choose a treatment assignment rule $\pi$, (where $\pi(z)\in \lbrace 0,1 \rbrace $) that achieves high expected utility compared to the baseline of no treatment in the new population. For example, equating the ITE with the utility gain from treatment, one would hope to achieve a large value of the objective  $\ell[\pi]=\int E[y|a=\pi(z),z]-E[y|a=0,z]d \tilde{P}_z (z)$, where $\tilde{P}_z$ is the distribution of $z$ in the new population which one may take as known. While this objective depends on the unknown CATE, given sample data, the posterior distribution of this object can be evaluated, and one could find the treatment allocation rule (perhaps in some restricted class) that maximizes the posterior expectation of this object (this is the approach in Bayesian statistical decision theory). This suggests an alternative (to information gain and the total variance) objective one could use to select the points whose outcomes one should sample, namely the supremum over $\pi$ of the posterior expectation of $\ell[\pi]$. I wondered if the methods in the paper would still apply for this objective?

---

> ### Author Response · Authors · 2025-11-20
>
> We appreciate the reviewer’s thoughtful and detailed comments. Below, we provide point-wise responses and will incorporate these discussions in the revision.
>
> ---
> **[Q1. The could be a little more detail given regarding possible real-world applications. The figures are too small. The grammar in some parts of the appendices needs fixing.]**
>
> > On Real-World Applications
>
> We appreciate the reviewer's desire for more detail. The problem our ActiveCQ framework is designed to solve, estimating causal quantities when outcome labels are expensive and scarce, is itself motivated by numerous high-stakes, practical scenarios.
> - **Healthcare:** Estimating the true effect of a medical treatment often requires costly and time-consuming longitudinal surveys to gather patient outcomes. Our framework provides a principled way to select the most informative subset of patients for this follow-up, maximizing the accuracy of the treatment effect estimate under a fixed budget.
> - **Social Services & Policy:** As highlighted in related work, a street outreach provider, for instance, may have millions of unstructured case notes, but only a limited budget for experts to read and label the true housing outcomes embedded within that text. Our method offers a formal strategy to decide which notes to annotate to best evaluate their program's impact.
> - **Business:** The same principle applies to quantifying the effect of a sales campaign or A/B testing, where determining the true outcome requires significant expertise and evidence.
>
> We will highlight these practical motivations more explicitly in the revision.
>
> > On Figure Size
>
> We will ensure that all figures are enlarged in the revision to improve readability.
>
> >On Appendix Grammar
>
> We also thank the reviewer for their careful reading and for pointing out the grammatical errors in the appendices. We will perform a thorough proofread of the entire manuscript, especially the supplementary materials, to correct these and improve the overall presentation.
>
> ---
> **[Q2. I wondered which if any of the simulated/semi-synthetic datasets violate the assumption that the outcome is conditionally Gaussian with fixed variance? While the likelihood is misspecified if this is the case, the comparison on such potentially less favorable datasets might be quite informative as in the real world conditionally heteroskedastic errors may be very common.]**
>
> We thank the reviewer for this nuanced question. To address your concern regarding real-world conditionally heteroscedastic errors, we have added a new additional experiment and combined it with our existing robustness analysis.
>
> **1. New Experiment: Conditionally Heteroscedastic Noise**
> We introduced heteroscedasticity into the visualization dataset by generating outcomes where the noise variance explicitly depends on the covariates. As shown in the new Fig. 11 in the revised Appendix, our proposed methods (specifically the CME-based strategies) continue to significantly outperform baselines. They achieve lower AMSE faster, demonstrating that our framework effectively handles scenarios where uncertainty varies across the input space.
>
> **2. Robustness to Non-Gaussian Likelihoods**
> This complements our existing ablation study in **App. E.6.2 (Fig. 12)**, where we tested robustness against Heavy-tailed (Student-t, Laplace) and Uniform noise. The fact that our method remains robust under both heteroscedasticity (variable variance) and heavy tails (outliers) provides strong empirical evidence that the ActiveCQ framework is resilient to severe likelihood misspecification, making it well-suited for complex real-world data.

---

> ### Author Response · Authors · 2025-11-20
>
> **[Q3. Extention to adaptive experimental design with treatment assignment.]**
>
> We thank the reviewer for this insightful comment. You are absolutely correct to draw a critical distinction between two different goals:
>
> 1.  **Our Current Goal (Accurate Estimation):** Our chosen objectives, IG and TVR, are estimation-centric. The goal is to minimize the uncertainty of the CATE function $\hat{\tau}(z)$ globally across its entire domain.
> 2.  **Your Proposed Goal (Optimal Decision-Making):** Your proposed objective is decision-centric. It aims to find an optimal policy $\pi(z)$ to maximize downstream utility. As you noted, this objective prioritizes the *sign* of $\hat{\tau}(z)$ near the decision boundary over the exact magnitude of $\hat{\tau}(z)$ in regions where the decision is obvious.
>
> To answer your core question: "I wondered if the methods in the paper would still apply for this objective?" In fact, our framework is not only applicable but serves as the essential foundation required to compute this advanced, decision-theoretic objective. To optimize the decision-theoretic objective, one must calculate the expected gain in policy value from a new observation. This requires two capabilities:
> 1.  Calculating the posterior expectation of the policy value under the current data.
> 2.  Analytically computing how this posterior updates after observing a hypothetical new data point.
>
> This is precisely what our ActiveCQ framework (Prop. 1) provides: an analytic, closed-form posterior distribution (mean $\nu$ and covariance $q$) for the CATE function $\hat{\tau}(z)$. Without such a tractable posterior (e.g., in methods that rely only on variance proxies), directly optimizing this decision objective is computationally intractable. The transition from estimation to policy optimization is a frontier in our field. For instance, the recent work[1] introduces a similar objective, "Post-Trial Mean Benefit", which maximizes the value of the final treatment policy. The authors achieve this by extending a Bayesian acquisition function (Causal-BALD) to target this specific policy goal. This demonstrates that:
> 1.  Targeting downstream policy value is a critical and timely research direction.
> 2.  Success in this direction requires building upon a strong Bayesian framework (like our GP+CME framework) that can support task-specific acquisition functions.
>
> Therefore, while our paper implements general-purpose estimation objectives (IG/TVR), our core technical contribution can be used for this advanced, policy-driven acquisition. This is a promising direction for future work that our framework is uniquely positioned to support. We also added this discussion into the revision in App. G.1.

---

### Official Review · Reviewer_NNrP · 2025-10-31

**Soundness:** 3
**Presentation:** 3
**Contribution:** 3
**Rating:** 6
**Confidence:** 4

**Summary:**

The paper introduces ActiveCQ, a pool-based active learning approach tailored towards estimating causal quantities of interest. The approach is observational, selecting “unlabeled” items from a pool to query their outcomes (labels), as opposed to carrying out interventions.  ActiveCQ follows a Bayesian active learning approach, where uncertainty about the causal quantity of interest is minimized. A key insight of the proposed approach is that if the outcome is modeled as a Gaussian process (GP), the causal quantity of interest is also a GP, with mean and covariance functions derived in equations (4), along with two sample-based estimators based on conditional density estimation and conditional kernel mean embeddings.  This insight allows utilizing acquisition functions from active learning with Gaussian processes, along with a (suitably adapted) theoretical analysis, to this novel setting.  The approach is evaluated in a quite extensive set of experiments.

**Strengths:**

- The specific instance of active learning considered in this paper is interesting, practically relevant and has not been studied yet to the best of my knowledge
- The approach is principled, and accompanied by a theoretical analysis
- The experiments and baselines are quite comprehensive
- The paper is quite well written (except some acronyms, such as SUTVA, should be defined to be more self-contained)

**Weaknesses:**

- The paper is primarily a natural composition of ideas that have individually been fairly well explored, thus perhaps not exceedingly surprising to researchers in the area.  Nevertheless, I find it makes a relevant contribution.

**Questions:**

- The figures are tiny (especially Fig 3)... perhaps it would make sense to move some to the appendix?

---

> ### Author Response · Authors · 2025-11-20
>
> We appreciate the reviewer’s thoughtful and detailed comments. Below, we provide point-wise responses and will incorporate these discussions in the revision.
>
> ---
> **[Q1. The paper is quite well written (except some acronyms, such as SUTVA, should be defined to be more self-contained).]**
>
> We thank the reviewer for the positive assessment of our paper's writing. We agree that defining key acronyms is essential for making the paper self-contained. We have explicitly added the formal definition of SUTVA (Stable Unit Treatment Value Assumption) in App. C.1 of the revised manuscript. We have also conducted a thorough proofread to ensure all other acronyms are properly defined upon their first appearance.
>
> ---
> **[Q2. The paper is primarily a natural composition of ideas that have individually been fairly well explored, thus perhaps not exceedingly surprising to researchers in the area. Nevertheless, I find it makes a relevant contribution.]**
>
> We agree that the core components, GP, active learning, and conditional kernel embeddings, are individually established. However, we respectfully suggest that our primary contribution is the novel and non-trivial synthesis of these ideas to define and solve a new, broader problem. We would like to highlight three aspects of this synthesis that we believe are novel and significant:
>
> 1.  **New, Unified Problem Formulation:** Our work is the first to formalize the general task of ActiveCQ: actively estimating a broad class of Causal Quantities (ATE, ATT, CATE, ATEDS, etc.). This "novel task formalization" and "unified framework" moves the field beyond the literature's narrow focus on just CATE.
> 2.  **Elegant Technical Core:** Our key technical contribution lies in the synergistic integration of GPs with CMEs. This combination is not merely a modular stacking of two components, CME’s ability to represent distributions directly in the GP’s kernel, induced feature space enables us to derive a closed-form posterior for the causal quantity $\hat{\tau}$ itself (Prop. 1). Importantly, this synergy is what makes the method actively adaptive: as the GP is retrained in each AL round and its feature space evolves, the CME is re-estimated in the same space, maintaining perfect task alignment and avoiding density-estimation misspecifications. This tight coupling is the key mathematical mechanism that underpins both the analytic tractability and the efficiency of our framework.
> 3.  **A New, Principled Capability:** This analytic posterior is why our contribution is relevant. It allows us, for the first time, to derive principled acquisition functions (IG/TVR) that are explicitly CQ-aware. This directly solves the fundamental objective mismatch of prior methods, which incorrectly target the regression function's uncertainty over the pool distribution, not the causal quantity's uncertainty over the target distribution.
>
>
> ---
> **[Q3. The figures are tiny (especially Fig 3)... perhaps it would make sense to move some to the appendix?]**
>
> We appreciate the reviewer’s comment and agree that some figures were difficult to read due to size constraints. In the revised version, we have enlarged the key figures (especially Fig. 3) to improve readability while moving some supplementary visualizations to the appendix for better clarity and balance between the main text and supplementary materials.

---

### Official Review · Reviewer_u6vV · 2025-11-01

**Soundness:** 4
**Presentation:** 3
**Contribution:** 4
**Rating:** 8
**Confidence:** 4

**Summary:**

This paper proposes the first unified framework for active estimation of multiple causal estimands, including ATE, ATT, CATE and ATE under distribution shift. The framework is a Bayesian framework with uncertainty quantification, which is supported by solid theoretical grounding.   Empirical validation on several synthetic or semi-synthetic datasets shows the effectiveness of the framework.

**Strengths:**

1. This paper is novel in task formalization, unified modeling of causal estimands, and creative integration of Gaussian Processes and Conditional Mean Embeddings to balance uncertainty quantification with computational efficiency in active learning.
2. The paper demonstrates exceptional quality through theoretical rigor, robust experimental design, and transparent reproducibility practices.
3. The paper is exceptionally clear, making its technical content accessible to both experts in causal inference and researchers new to the field.

**Weaknesses:**

1. This paper acknowledges standard GPs’ cubic complexity and mentions efficient alternatives (sparse variational GPs, Random Fourier Features, Nyström). However, no empirical evaluation of these methods is provided.
2. The CME’s regularization parameter $\lambda=0.01$ is fixed with no sensitivity analysis.
3. If I am right, the greedy strategy for batch selection still requires iteratively re-computing the posterior for all remaining pool points after each selection, which is not efficient enough.

**Questions:**

I would suggest using a alternative term for the word "causal quantity". In potential outcome framework, a more common definition is "causal estimand", indicating that we aim to estimate the value.

---

> ### Author Response · Authors · 2025-11-20
>
> Thank you very much for your positive evaluation of our paper. Below, we hope that our clarification addresses your concerns.
>
> ---
> **[Q1. This paper acknowledges standard GPs’ cubic complexity and mentions efficient alternatives (sparse variational GPs, Random Fourier Features, Nyström). However, no empirical evaluation of these methods is provided..]**
>
> We did not include an empirical evaluation of these scalable alternatives, and we would like to offer two points of context:
>
> 1.  **The Problem's Trade-off:** The ActiveCQ framework is fundamentally designed for settings where data acquisition cost is the primary bottleneck, not computational cost. This is common in high-stakes domains like medicine or public policy, where labeling a single outcome is exceptionally expensive. In this specific (and highly practical) regime, the computational cost of an exact GP is often negligible compared to the cost of acquiring one new data point. Therefore, using an exact GP is the preferable choice as it provides the most accurate posterior uncertainty to guide the expensive selection process.
> 2.  **Modularity of the Framework:** Our core contribution is the ActiveCQ framework itself, saying the principled, CQ-aware acquisition strategy and its theoretical guarantees (e.g., Prop. 1 and Thm. 2). The GP is a modular regressor component within this framework. Our contributions are not dependent on the specific GP implementation. Our framework is fully compatible with plugging in scalable alternatives like RFFs or Sparse VGP; doing so is a standard trade-off between computational efficiency and posterior effectiveness, and does not alter the scientific principle of our method.
>
>
> ---
> **[Q2. The CME’s regularization parameter $\lambda=0.01$ is fixed with no sensitivity analysis.]**
>
> Thank you for pointing this out. We apologize for this error in the text. The statement in App. E.2.2 that "The regularization parameter is fixed at $\lambda = 0.01$" is incorrect. In fact, $\lambda$ is a learnable parameter that can be optimized alongside other model hyperparameters by maximizing the marginal log-likelihood. The value of $0.01$ was only its initialization value to ensure numerical stability at the start of training. We have corrected this sentence in the revision.
>
> ---
> **[Q3. If I am right, the greedy strategy for batch selection still requires iteratively re-computing the posterior for all remaining pool points after each selection, which is not efficient enough.]**
>
> Your understanding is correct. The greedy strategy (denoted with `_G` in our figures) is indeed computationally expensive for precisely the reason you mentioned: it requires iteratively updating the posterior and re-evaluating the utility of all remaining pool points after each single selection within a batch. This is a well-known **trade-off** in batch AL:
>
> 1.  **Naive "Top-k" (Faster):** Selecting the top $n_b$ points based on a single utility calculation is fast but often suboptimal. It tends to select a batch of redundant points that are all clustered in the same high-uncertainty region.
> 2.  **Greedy (Slower):** The iterative, sequential selection process is designed to mitigate this. By re-computing the posterior after adding each point, it ensures that the *next* point selected adds the most marginal utility, leading to a more diverse and informative batch.
>
> We explicitly acknowledge this efficiency cost in the paper. Our runtime analysis confirms that the greedy methods are the slowest due to these "frequent posterior recalculations". However, we also showed that the computationally cheaper, naive "top-k" method often performs comparably to the greedy approach in our experiments. This makes it a very practical and efficient alternative when computational resources are limited.
>
>
> ---
> **[Q4. I would suggest using a alternative term for the word "causal quantity". In potential outcome framework, a more common definition is "causal estimand", indicating that we aim to estimate the value.]**
>
> We are aware that causal estimand is the more common term in the potential outcomes literature to denote the target of estimation. However, we intentionally adopt the term causal quantity to emphasize a slightly broader conceptual scope. Specifically, our framework aims not only to estimate identifiable estimands (e.g., ATE, CATE, ATT) but also to unify a class of causal objectives that share a common integral form—some of which may be abstract causal functionals defined prior to identification. The term causal quantity thus reflects our goal of covering both identifiable estimands and their theoretical counterparts under a unified notation. That said, we will clarify this distinction in the revised version by explicitly noting that CQs correspond to the causal estimands under the standard identifiability assumptions. We hope this explanation clarifies our choice of terminology.

---

### Meta-Review · Area_Chair_qDtM · 2026-01-10

**Summary:**

The paper uses the active learning for causal estimation involving ATE and CATE. The paper uses a Bayesian active learning setup. They show that if the outcomes are modeled as GP, the causal quantity follows as GP. The paper theoretically analyses the setup and provide empirical validation of their approach.  As rev TTvj wrote, the authors do also consider methods like  kernel density estimation and Monte Carlo sampling, but their approach works quite well. Reviewers appreciated the novelty, theoretical analysis and insights presented the paper. One reviewer acknowledged that there is no surprise in the approach and results, but that does not reduce the value being added.  Hence, I recommend acceptance.

In response to some reviewer concerns, that their method is fully compatible with Random Fourier Features. But, I would request that if the authors could, they should address them in more details in the final revision.

**Reviewer Concerns:**

To the best of my understanding, all concerns are addressed by the rebuttals.

**Reviewer Scores:**

Reviewers were already positive.  Reviewer bTeT was slightly negative, but their concerns are addressed.

---

### Decision · Program_Chairs · 2026-01-26

Accept (Poster)